

# Advances in Air Quality Research – Current and Emerging Challenges

Ranjeet S Sokhi[1], Nicolas Moussiopoulos[2], Alexander Baklanov[3], John Bartzis[4], Isabelle Coll[5], Sandro Finardi[6], Rainer Friedrich[7], Camilla Geels[8], Tiia Grönholm[9], Tomas Halenka[10], Matthias Ketzel[8], Androniki Maragkidou[9], Volker Matthias[11], Jana Moldanova[12], Leonidas Ntziachristos[2], Klaus Schäfer[13], Peter Suppan[14], George Tsegas[2], Greg Carmichael[15], Vicente Franco[16], Steve Hanna[17], Jukka-Pekka Jalkanen[9], Guus J. M. Velders[18,19] and Jaakko Kukkonen[9,1]

[1]Centre for Atmospheric and Climate Physics Research, and Centre for Climate Change Research, University of Hertfordshire; College Lane, Hatfield, AL10 9AB, UK

[2]Laboratory of Heat Transfer and Environmental Engineering, School of Mechanical Engineering, Aristotle University, Thessaloniki, GR-54124, Greece

[3]Science and Innovation Department, World Meteorological Organization (WMO), 7 bis, Avenue de la Paix, BP2300, CH-1211 Geneva 2, Switzerland

[4]Department of Mechanical Engineering, University of Western Macedonia, 50100 Kozanim Greece

[5] Univ Paris Est Creteil and Université de Paris, CNRS, LISA, F-94010 Créteil, France

[6]ARIANET, via Gilino 9, 20128 Milano, Italy

[7]Institute for Energy Economics and Rational Energy Use, University of Stuttgart, Stuttgart, D-70180 Germany

[8]Department of Environmental Science, Aarhus University, Roskilde, DK-4000, Denmark

[9]Finnish Meteorological Institute, Erik Palmenin aukio 1, P.O.Box 503, FI-00101 Helsinki

[10]Charles University, Faculty of Mathematics and Physics, Department of Atmospheric Physics, V Holešovičkách 2, 182 00 Prague, Czech Republic

[11]Institute of Coastal Environmental Chemistry, Helmholtz-Zentrum Hereon, Max-Planck-Straße 1, D-21502 Geesthacht, Germany

[12]IVL Swedish Environmental Research Institute, P.O. Box 530 21, SE-400 14 Göteborg, Sweden

[13]Aerosol Akademie, 83404 Ainring, Germany

[14]Institute of Meteorology and Climate Research (IMK-IFU), Karlsruhe Institute of Technology (KIT), 82467 Garmisch-Partenkirchen, Germany

[15]Department of Chemical and Biochemical Engineering, University of Iowa, Iowa City, IA 52242, USA[16] Vicente Franco,

[16]European Commission, DG Environment, Brussels, Belgium

[17]Harvard School of Public Health, Boston, MA, USA

[18]National Institute for Public Health and the Environment (RIVM), PO Box 1, 3720 BA Bilthoven, The Netherlands

[19]Institute for Marine and Atmospheric Research Utrecht, Utrecht University, Heidelbergerlaan 8, 3584 CS Utrecht, The Netherlands

*Correspondence to*: Ranjeet S Sokhi (r.s.sokhi@herts.ac.uk)

**Abstract.** This review provides a community's perspective on air quality research focussing mainly on developments over the past decade. The article provides perspectives on current and future challenges as well as research needs for selected key topics. While this paper is not an exhaustive review of all research areas in the field of air quality, we have selected key topics that we feel are important from air quality research and policy perspectives. After providing a short historical overview, this review focuses on improvements in characterising sources and emissions of air pollution, new air quality observations and instrumentation, advances in air quality prediction and forecasting, understanding interactions of air quality with meteorology





and climate, exposure and health assessment, and air quality management and policy. In conducting the review, specific objectives were (i) to address current developments that push the boundaries of air quality research forward, (ii) to highlight

the emerging prominent gaps of knowledge in air quality research and (iii) and to make recommendations to guide the direction for future research within the wider community. This review also identifies areas of particular importance for air quality policy. The original concept of this review was borne at the International Conference on Air Quality 2020 (held online due to the COVID 19 restrictions during 18-26 May 2020), but the article incorporates a wider landscape of research literature within the field of air quality science. On air pollution emissions the review highlights, in particular, the need to reduce uncertainties in

emissions from diffuse sources, particulate matter chemical components, shipping emissions and the importance of considering both indoor and outdoor sources. There is a growing need to have integrated air pollution and related observations from both ground based and remote sensing instruments, including especially those on satellites. The research should also capitalize on the growing area of lower cost sensors, while ensuring a quality of the measurements which are regulated by guidelines. Connecting various physical scales in air quality modelling is still a continual issue, with cities being affected by air pollution

gradients at local scales and by long range transport. At the same time, one should allow for the impacts from climate change on a longer timescale. Earth system modelling offers considerable potential by providing a consistent framework for treating scales and processes, especially where there are significant feedbacks, such as those related to aerosols, chemistry and meteorology. Assessment of exposure to air pollution should consider both the impacts of indoor and outdoor emissions, as well as application of more sophisticated, dynamic modelling approaches to predict concentrations of air pollutants in both

environments. With particulate matter being one of the most important pollutants for health, research is indicating the urgent need to understand, in particular, the role of particle number and chemical components in terms of health impact, which in turn requires improved emission inventories and models for predicting high resolution distributions of these metrics over cities. The review also examines, how air pollution management needs to adapt to the above-mentioned new challenges and briefly considers the implications from the COVID-19 pandemic for air quality. Finally, we provide recommendations for air quality

research and support for policy.

**Dedication**

We wish to dedicate this article to the following eminent scientists who made immense contributions to the science of air quality and its impacts:

Paul J. Crutzen (1933-2021), atmospheric chemist, awarded the Nobel Prize in Chemistry 1995

Mario Molina (1943-2020), atmospheric chemist, awarded the Nobel Prize in Chemistry 1995

Kirk Smith (1947-2020), global environmental health

Martin Williams (1947-2020), air quality science and policy

Sergej Zilitinkevich (1936-2021), atmospheric turbulence, awarded the IMO Prize 2019



## 1. Introduction

Air pollution remains one of the greatest environmental risks facing humanity. WHO (2016) estimated that over 90% of the global population is exposed to air quality that does not meet WHO guidelines and Shaddick et al., (2020) reporting that 55% of the world's population were exposed to $PM_{2.5}$ concentrations that were increasing between 2010 and 2016. Shaddick et al., (2020) also highlighted marked inequalities between global regions with decreasing trends in annual average population-weighted concentrations in North America and Europe but increasing trends in Central and Southern Asia. WHO (2016) has evaluated that approximately seven million people died prematurely in 2012 throughout the world as a result of air pollution exposure originating from outdoor and indoor anthropogenic emissions.

Over the past decade there have been significant developments in the field of air quality research spanning, e.g., improvements in characterising sources and emissions of air pollution, new measurement technologies offering the possibility of lower cost sensors, advances in air quality prediction and forecasting, understanding interactions with meteorology and climate and exposure assessment and management. However, there has not been a broader and comprehensive review of recent developments that push the boundaries of air quality research forward. This was recognised as a major gap in the literature at the last International Conference on Air Quality – Science and Application held online due to the COVID 19 restrictions during 18-26 May 2020. While the concept of this review originated at the International Conference on Air Quality and was stimulated by the presentations and discussions at the conference, this article has been extended to incorporate a wider landscape of research literature in the field of air quality, spanning in particular the developments occurring over the last decade. It is hoped that such a review will help to pave the path for further research in key areas where significant gaps of knowledge still exist, and also to make recommendations to guide the direction for future research within the wider community. Although this paper has been written to be accessible to readers from a wide scientific and policy background, it does not seek to provide an introduction to the topic of air quality science. For readers less familiar with the research area, an introductory lecture with a focus on air quality in megacities has been published by Molina (2021). There are also other recent specific reviews, e.g., Manisalidis et al. (2020) on health impacts and Fowler et al., (2020) on air quality developments. This section begins with a short historical perspective on air quality research, before providing the underlying rationale for the key areas considered in this paper.

### 1.1 A brief historical perspective

In order to provide context to the topics considered in this review, this section briefly touches upon developments of air quality research since the last century. For a more thorough historical survey of air quality issues, the reader is referred to Fowler et al. (2020). Over the previous century there have been a number of landmark events of elevated air pollution that have brought



air quality increasingly to prominence, especially in relation to the adverse health impacts. It has been well known since the early 1900's that cold weather in winter can lead to increased mortality (e.g., Russell 1926).

The perception that air pollution can have severe health impacts significantly changed, when a high air pollution episode
occurred during 1-5 December 1930 over an industrial town in the Meuse Valley in Belgium (Firket 1936). The atmospheric conditions were foggy and stagnant. A large proportion of the population experienced acute respiratory symptoms; in addition, health conditions of people with pre-existing cardiorespiratory problems worsened (e.g., Nemery et al., 2001; Anderson 2009). A similar event was recorded in Donora, Pennsylvania, USA during October 1948, reported by Schrenk et al. (1949). Although air pollution was generally treated as a nuisance, this 'unusual episode' along with that over the Meuse valley raised awareness
and acceptance of the seriousness of air pollution for human health. Both air pollution events, Meuse valley and Donora, were associated with air pollution from industrial emissions, which accumulated during cold winter periods exhibiting atmospheric stagnation caused by thermal inversions.

The so-called 'Great London Smog' occurred during 5-9 December 1952, when similar stagnant atmospheric conditions were
prevalent. However, in this case cause of severe air pollution was mainly the burning of low-grade, sulphur rich coal for home heating (e.g., Anderson 2009). Estimates of deaths resulting from this smog episode ranges from 4000 to 12000 (e.g., Stone 2002).

Since these historical events, the prominence of air pollution sources has changed from industrial and heating to road traffic
and become a global threat to health. Trends of air pollution emissions over the past decades have been markedly different for different regions of the world, which has led to similar disparities in air quality concentrations (e.g., Sokhi 2011). These disparities still exist, as shown in Figure 1. Spatial distributions in this figure are based on recent analysis showing the large variations in population weighted annual mean $PM_{2.5}$ concentrations across the globe. Commonly, now some of the highest concentrations occur in parts of Asia, Africa and Latin America as reported by Health Effects Institute (HEI, 2020).



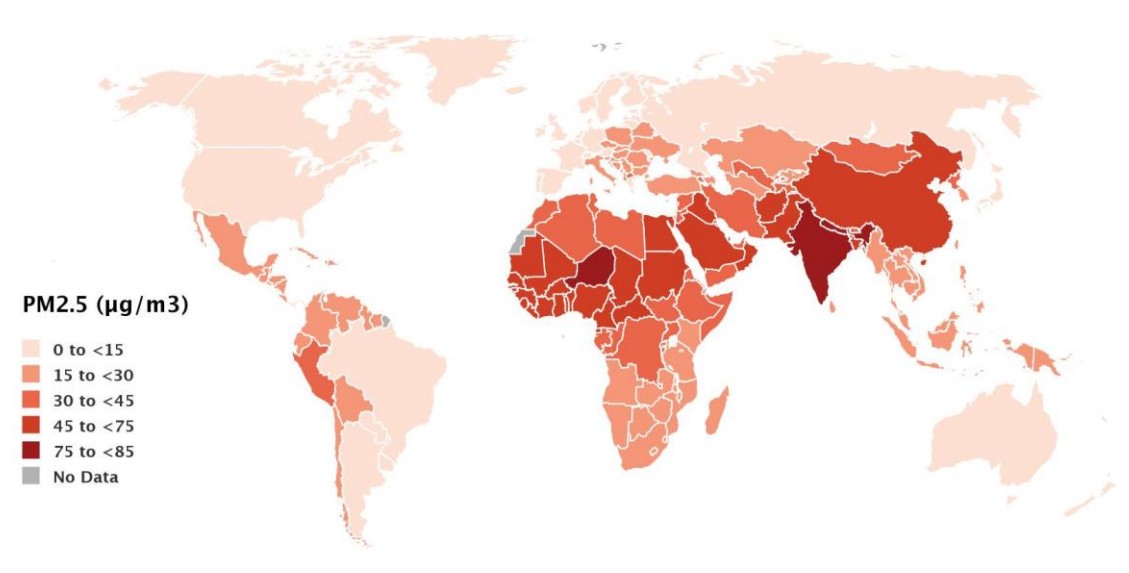

Figure 1: Global distribution of population weighted annual PM$_{2.5}$ concentrations for 2019 (HEI, 2020). Figure produced from https://www.stateofglobalair.org/data/#/air/map.

As the recognition of poor air quality has increased, so has the need for capability to assess levels of key air pollutants not only though monitoring but also through modelling. Historically, although air pollution was obviously poor prior to the first World War (WWI), the primary impetus for development of transport and dispersion (T&D) models during and after WWI was the widespread use of chemical weapons. Fundamental theoretical advances were made by Richardson, Batchelor, and many other famous fluid dynamicists. The earliest models were analytical (e.g., Gaussian and K-theory) models used for surface boundary

layer releases. With the advent of nuclear weapons in WWII, new emphasis was placed on plume rise and dispersion of large thermal radiological explosions. Thus, the full troposphere and stratosphere had to be modelled.

Later in the 1980s the first investigations came up about the atmospheric consequences of a hypothetical nuclear war initiated by Paul Crutzen (Crutzen and Birks, 1982) and others (Aleksandrov and Stenchikov, 1983; Turco et al., 1983). The concept

of a nuclear winter was created. It is one of the first examples that enormous emissions of dust into the atmosphere cause global effects and catastrophic long-term climate change. Also, the nuclear winter scenario was examined in recent years with current model tools for certain nuclear war scenarios (Robock et al., 2007; Toon et al., 2019).





Deposition (wet and dry) was a main concern for many radiological substances, especially for accidental plume dispersion
monitoring and modelling of nuclear power plants. In the US, a major change was the introduction of the Clean Air Act in the
1970s. A similar legislation was also issued in other countries. This effort initially focused on T&D models for industrial
sources, such as the stacks of fossil power plants. The first applied models were analytical plume rise and Gaussian T&D
models. Soon computer codes were written to solve these equations and produce outputs at many spatial locations and for
every hour of the year.

## 1.2 Sources and emissions of air pollutants

From a human health perspective, the key emission sources are those affecting concentration of particulate matter and its size
fractions ($PM_{2.5}$ and $PM_{10}$), but also sources affecting other air pollutants, such as ozone and nitrogen dioxide ($NO_2$), especially
in highly populated urban areas. Sources in the direct vicinity of urban areas could also be considered as especially important,
including vehicular traffic and shipping, local industrial sources, various abrasive processes, and residential and commercial
heating.

An important component of PM is secondary; regional sources of the precursors of secondary PM are therefore of major
importance. These include volatile organic compounds (VOCs), nitrogen dioxide ($NO_2$), sulphur dioxide ($SO_2$), and ammonia
($NH_3$); the first two also being precursors of ozone ($O_3$). Important regional precursor sources are biogenic and industrial
emissions of VOCs, agriculture ($NH_3$), road traffic (nitrogen oxides, NOx = NO+ $NO_2$), shipping (NOx and $SO_2$), and industrial
and power generation sources, along with biomass burning and forest fires (VOC, NOx, also primary PM). An important
source of PM is the resuspension of dust, especially in arid regions and seasonally also in areas with intensive agriculture.

While Europe and many other parts of the world have experienced decreasing anthropogenic emissions, climate change and
its associated impacts can lead to an increase of dust and wildfire emissions, as a result of increased drought and desertification.
Climate change is also expected to lead to significantly higher biogenic VOC emissions in different regions, e.g., Arctics and
China (Kramshöj et al., 2016; Liu et al., 2019), also from urban vegetation (Churkina el al., 2017).

The emission inventory work in Europe is harmonised through the official reporting of EU member states of their emissions
to the Commission through an e-reporting scheme (Implementing Provisions for Reporting, IPR of EU Air Quality Directive,
2008/50/EC). The methodologies applied by the individual member states can, however, differ, which can sometimes bring
inconsistencies into the reported national emissions. Within the last decade the EU-funded MACC project and the on-going
Copernicus service have been developing consistent European-wide and global gridded emission inventories, which are
suitable for air quality modelling. The access to the different inventories and analysis of differences have been facilitated by





centralized databases like Emissions of atmospheric Compounds and Compilation of Ancillary Data (ECCAD, eccad.aeris-data.fr).

Developing innovative methods to refine the emission inventories feeding the models and conducting studies to discriminate
the role of different sources in local air quality has become essential to reduce uncertainties in predictions of urban air quality, and help target effective abatement measures (Borge et al., 2014). The emission compilation that needs to be carried out also requires (i) the involvement of all stakeholders (e.g., citizens, decision-makers, service providers and industrialists) and (ii) the implementation of dedicated and specific tools for assessing the quality of the urban environment. This type of research can be used for quantifying the impacts of different emission control scenarios, and supporting incentive policies (Fulton et
al., 2014).

One area that has been receiving increased attention recently is ship emissions, which are an important source of air pollution, especially in coastal areas and harbour cities. Detailed bottom-up emission inventories based on ship position data have been established for $SO_2$, $NO_x$, PM, carbon monoxide (CO) and VOCs for various marine regions and also globally (Jalkanen et al.,
2009, 2012, 2016, Aulinger et al., 2016, Johansson et al., 2017). Despite these advances, the evaluation of the shipping emissions for products of incomplete combustion, such as black carbons (BC), CO and VOCs, is uncertain, as these may depend on characteristics, which are not known accurately, such as the service history of ships. Regional model applications have quantified the contribution of shipping to air pollution to be of the order of up to 30 %, depending on pollutant and region (e.g. Matthias et al., 2010, Jonson et al., 2015, Aulinger et al., 2016, Karl et al., 2019a, Kukkonen et al, 2018 and 2020a). More
recent studies focus on the harbour and city scale, where relative contributions from ships to $NO_2$ concentrations may be even higher (Ramacher et al., 2019, Tang et al., 2020). Effects of in-plume chemistry, e.g., regarding the $NO_x$ removal and secondary aerosol formation, are not sufficiently well considered in larger scale dispersion models (e.g., Prank et al., 2016).

**1.3 Air quality in cities**
Extensive and growing urban sprawl in different cities of the world is leading to environmental degradation and the depletion of natural resources, including the availability of arable land, thereby resulting in per capita increases of resource use and greenhouse gas emissions as well as air pollution, with significant impacts on health (WHO, 2016). Urban features have a profound influence on air quality in cities due to e.g. diurnal changes in urban air temperature, urban heat island, which is developed especially during heat waves (Halenka et al., 2019), stable stratification and air stagnations, wind flow and
turbulence near and around streets and buildings affecting air pollution hotspots. Climate change will modify urban meteorology patterns which will affect air quality in cities and may even affect atmospheric chemistry reaction rates. The relative role of urban meteorology and climate compared to local emissions and chemistry is complex, non-linear and is subject to continued research especially with boundary layer feedback (Baklanov et al., 2016).

With air quality standards being regularly exceeded in many urban areas across the globe, air quality issues are today strongly centred on the phenomena of proximity to emitters such as traffic - or certain industrial activities present in urban areas - but they also call for better understanding of contributions from regional, diffuse, or specific local sources (e.g., residential wood combustion and maritime traffic) to the daily exposure of city dwellers. In particular, the prevalent issue of individual exposure calls for a better understanding of the variability of concentrations at street level and the dispersion of emissions in the built

environment. However, the approach implemented should not only be local, since urban air quality management involves a set of scales going beyond the city limits, both in terms of the economic, societal, or logistical levers involved but also the interplay of pollutant sources and transport extending to regional and even global scales.

Beyond the scales of governance and urban functioning, it becomes essential to take into account of the fact that scale

interactions also exist in a geophysical context. The urban dweller has become especially exposed and vulnerable to the impacts of natural disasters, weather and climate extreme events and their environmental consequences. These events often result in domino effects in the densely populated, complex urban environment in which system and services have become interdependent. There has never been a bigger need for user-focused urban weather, climate, water, and related environmental services in support of safe, healthy, and resilient cities (Baklanov et al., 2018; Grimmond et al., 2020). The eighteenth World

Meteorological Congress (2019) noted the current rapid urbanization and recognized the need for an integrated approach providing weather, climate, water, and related environmental services tailored to the urban needs (WMO, 2019).

### 1.4 Measuring air pollution

Measurements in the atmosphere are not only necessary for air quality monitoring but also for different purposes in weather

forecast and climate change study, energy production, agriculture, traffic, industry, health protection or tourism (e.g., Foken, 2021). Additional areas of application include the detection of emissions into the atmosphere, disaster monitoring and the initialisation and evaluation of modelling. Depending on the different objectives, in situ measuring as well as ground-based, aircraft-based, and space-based remote sensing techniques and integrated measuring techniques are available. Satellite observations are a growing field of development due to increasingly small and thus cost-effective platforms (down to

nanosatellites). Another area of growth is the use of unmanned aerial vehicles (UAV) for air pollution measurements (Gu et al., 2018).

Networks of ground level measurements with continuous monitoring stations remain a major effort but the coverage is starkly regionally dependent and with scarce measurements in the continent of Africa (Rees et al., 2019).


Over the past decade, there has been increasing recognition that measuring air pollution at outdoor locations may not necessarily reflect the health impact on individuals or populations. The research should therefore be directed to the evaluation of both personal exposure and dynamic population exposure (Kousa et al., 2002, Soares et al., 2014). Temporal concentration




and location information is needed on air pollution concentrations at all the relevant outdoor and indoor microenvironments.
The actual exposure of individuals and populations cannot realistically be represented by selected concentrations at fixed outdoor locations, due to the fine-resolution spatial variability of concentrations in urban areas, and the mobility of people (Kukkonen et al., 2016b, Singh et al., 2020b).

Further development is ongoing of the installation of a larger number of cheap measurement devices especially for $PM_{2.5}$, that
are    operated    by    people    interested    in    air    quality    in    so-called    citizen    science    projects (https://www.eea.europa.eu/publications/assessing-air-quality-through-citizen-science). Examples of such projects are the Open Knowledge Foundation Germany, OK Labs (http://luftdaten.info/), Opensense (open air quality, meteorological and noise data platform), connected with OK Labs (https://opensensemap.org/), or AirSensEUR: open framework for air quality monitoring, (https://airsenseur.org/website/airsenseur-air-quality-monitoring-open-framework/). However, the accuracy of
these measurements is still debated (Duvall et al., 2020, Concas et al., 2021), although the development of more accurate, but still lower cost devices is ongoing for denser measurement networks, 3-D-measurements, and new modelling. Measurements are not only required for compliance and for monitoring long-term trends. Observations are used more and more for evaluating models and where measurements might also be used to nudge the model results, for example through data assimilation (see for example Campbell et al. (2015), Wang et al. (2015a)).


## 1.5 Air quality modelling from local to regional scales

Air pollution models have and continue to play a pivotal role in furthering scientific understanding and to support policy. Additionally, to air quality assessments by regulatory methods, it is also important to predict or even forecast peak pollutant concentrations to prevent or reduce health impacts from acute episodes. Both complex and simple models have also been
developed for dispersion on urban and local scales. A review has been provided by Thunis et al., (2016) that examines local and regional scale models especially from an air quality policy perspective. Briefly, the spectrum of finer and urban scale air quality models applied for urban areas is very broad and includes urbanised chemistry transport models (CTMs) coupled with high-resolution meso-scale numerical weather prediction (NWP) models, Computational Fluid Dynamics (CFD) type of obstacle-resolved models in Reynolds-averaged Navier-Stokes (RANS) and large eddy simulation (LES) formulations (the
latest mostly only for research studies), as well as statistical and land use regression (LUR) models. Developments in local scale air quality models continues. For example, the dispersion on local or urban scales that also considers obstacle effects has recently been investigated using wind tunnels and CFD models (e.g. Badeke et al., 2020).

During the last decades many countries have established real-time air quality forecasting (AQF) programs to forecast
concentrations of pollutants of special health concerns. The history of AQF can be traced back to 1960s, when the U.S. Weather Bureau provided the first forecasts of air stagnation or pollution potential using numerical weather prediction (NWP) models to forecast conditions conducive to poor air quality (e.g., Niemeyer, 1960). Accurate AQF can offer tremendous societal and





economic benefits by enabling advanced planning for individuals, organizations, and communities in order to avoid exposure, and reduce and adverse health impacts resulting from air pollution. Forecasts can also assist urban authorities, for example, in

changing and managing traffic and hence reduce road emissions in a particular area. Air quality modelling however, can provide a more holistic assessment of air pollution for policy makers and decision makers to develop strategies that do not compromise benefits in one area while worsening air pollution in another.

Two main approaches can be generally distinguished in AQF: empirical/statistical methods and chemical transport modelling.

Until the mid-1990s, AQF has been mainly performed using empirical approaches and statistical models trained with or fitted to historical air quality and meteorological data (e.g., Aron, 1980). The empirical/statistical approaches have several common drawbacks for AQF which are reviewed and discussed by Zhang et al., (2012a) and Zhang and Baklanov (2020).

The chemical transport models (CTMs) are now more commonly used nowadays for air quality assessment and forecasting.

Over the last decade AQF systems based on CTMs have been developed rapidly and are currently in operation in many countries. Progress in CTM development and computing technologies has allowed daily AQFs using simplified or more comprehensive 3D CTMs, such as offline-coupled and online-coupled meteorology-chemistry models. There are several comprehensive review papers, e.g., Menut and Bessagnet (2010), Kukkonen et al. (2012), Zhang et al. (2012a,b), Baklanov et al. (2014), Ryan (2016), Bai et al. (2018) and Baklanov and Zhang (2020) which have more thoroughly examined the

development and principles of 3-D global and regional AQF models and identified areas of improvement in meteorological forecasts, chemical inputs, and model treatments of atmospheric physical, dynamic, and chemical processes.

Interest in regional pollution interests arose in the 1980s, initially spurred by the acid rain problem (Sokhi, 2012; Fowler et al., 2020). In the past few years, these regional air pollution models have become routinely linked with outputs of NWP models

such as WRF and ECMWF. Models such as WRF coupled with CTMs are often run in a nested mode down to an inner domain with grid size of 1 km. As computer speed and storage continually improve with developments in parameterisation, in the future, these nested models may potentially take over most applied T&D analyses on local scales. Another development over the last decade is the increasing use of ensemble techniques which have also progressed and make it possible to cover an increasing range of pollutants and physical parameters, using a multiplicity of observations (e.g. ground, airborne, satellite)

that enable the different dimensions of models to be investigated. At the same time, that the use of regional Eulerian models has grown, the puff, particle and plume T&D models for small scales and mesoscales have been improved. Several agencies and countries now have Lagrangian particle or puff models that are linked with an NWP model and are applied at all scales (Ngan et al., 2019).






### 1.6 Interactions of air quality, meteorology and climate

Meteorological processes are the main driver for atmospheric pollutants dispersion, transformation and removal. However, as many observational and modelling studies have shown (e.g. Rosenfeld et al., 2008; Zhang, 2008), the chemistry dynamics feedbacks exist among the Earth system components including the atmosphere. Potential impacts of aerosol feedbacks can be broadly explained in terms of four types of effects: direct, semidirect, first indirect and second indirect (e.g. Kong et al., 2015; Fan et al., 2016). Such feedbacks, forcing mechanisms and two-way interactions of atmospheric composition and meteorology can be important not only for air pollution modelling, but also for NWP and climate change prediction (CCMM, 2016).

There is a strong scientific need to increase interfacing or even coupling of prediction capabilities for weather, air quality and climate. The first driver for improvement is the fact that information from predictions is needed at higher spatial resolutions (and longer lead times) to address societal needs. Secondly, there is the need to estimate the changes in air quality in the future driven by climate change. Thirdly, there is continued improvements in prediction skill requires advances in observing systems, models and assimilation systems. In addition, there is also growing awareness of the benefits of more closely integrating atmospheric composition, weather and climate predictions, because of the important feedbacks resulting from the role that aerosols (and atmospheric composition in general) play in these systems. Recently, this trend for further integration is leading to greater coupling of atmospheric dynamics and composition models to deliver seamless Earth System Modelling (ESM) systems.

### 1.7 Air quality and health perspectives

Air pollution has serious impacts on our health by reducing our life span and exacerbating numerous illnesses. One of the most hazardous air pollutants is particulate matter. Primary particles are directly released into the atmosphere and originate from natural and anthropogenic sources. Secondary particles are formed in the atmosphere by chemical reactions involving, in particular, gas-to-particle conversion. Primary particles tend to be larger than secondary particles. Ultrafine and fine particles, on the other hand, deposit into the respiratory system; these may reach human lungs and blood circulation and may therefore cause severe adverse health effects (e.g., Maragkidou, 2018; Stone et al., 2017).

When considering numbers of particles, most of these in the atmosphere are smaller than 0.1 μm in diameter (e.g., de Jesus et al., 2019). On the other hand, the majority of the particle volume and mass is found in particles larger than 0.1 μm (e.g., Filella, 2012). The particle number concentrations are therefore in most cases dominated by the ultrafine aerosols, whereas the mass or volume concentrations are dominated by the coarse and accumulation mode aerosols (e.g., Seinfeld and Pandis, 2006). Other characteristics of PM have also been shown to be important in relation to health impact. The characteristics of atmospheric particles in addition to the size include mass, surface area, chemical composition, and shape and morphology (Gwaze, 2006).





It has been convincingly shown in previous literature that the exposure to particulate matter (PM) in ambient air can be associated with negative health impacts (e.g., Hime et al., 2018; Thurston et al., 2017). It is also known that PM can cause health effects combined with other environmental stressors, such as heat waves and cold spells, allergenic pollen or airborne microorganisms. For understanding such associations, reliable methods are needed to evaluate the exposure of human populations to air pollution.


The strong association between the exposure to mass-based concentrations of ambient PM air pollution and severe health effects has been found by numerous epidemiological studies (e.g., Pope et al., 2020) . In particular, there is extensive scientific evidence to suggest that exposure to PM air pollution can have acute effects on the human health resulting in, e.g., respiratory, cardiovascular and lung problems, chronic obstructive pulmonary diseases (COPD), asthma, oxidative stress, immune response

and even lung cancer (e.g., Chen and Chen, 2018; Hime et al., 2018; Falcon-Rodriguez et al., 2016; Thurston et al., 2017). For instance, a cohort study conducted across Montreal and Toronto (Canada) on 1.9 million adults during four cycles (1991, 1996, 2001, and 2006) resulted in a possible connection between ambient ultra-fine particles and incident brain tumors in adults (Weichenthal et al., 2020).

**1.8 Air quality management and legislative and policy responses**

Air quality management and policy is an important, but also a complex task for the political decision makers. It started in the middle of the last century when concerns about smoke and London smog arose. The national authorities at that time reacted by stipulating efficient dust filters and high stacks for large firings. In the 1980s, forest dieback led to a shift in focus on other important air pollutants, especially $SO_2$, $NO_x$ and later ozone, and so also on the ozone precursors including VOCs. In the

1990s studies showed a relation between $PM_{10}$ and 'chronic' mortality, thus drawing particular attention to the health effects of fine particles (WHO, 2013b). Also, in the 1990s, the European Commission (EC) increasingly took over the responsibility for air pollution control from the authorities of the member states, on the basis that there is free trade of goods in the European Union and also transboundary air pollutants.

The EC launched the first Air Quality Framework Directive 96/62/EC and its daughter Directives, which regulated the concentrations for a range of pollutants including ozone, $PM_{10}$, $NO_2$ and $SO_2$. The first standard for vehicles (Euro 1) was established in 1991. The sulphur content in many oil products was reduced starting in the late 1990s. Some of the problems with air pollution in the EU, e.g. the acidification of lakes, were caused by the transport of air pollutants from Eastern Europe to the EU. This problem was discussed in the United Nations Economic Commission for Europe (UNECE), as all countries

involved were members of this Commission. The Convention on Long-range Transboundary Air Pollution within the UNECE agreed on 8 protocols, that set aims for reducing emissions, starting 1985 with reducing national $SO_2$ emissions, with the latest protocol being the revised Protocol to Abate Acidification, Eutrophication and Ground-level Ozone (Gothenburg Protocol), which limits national $SO_2$, NOx, VOC, $NH_3$ and $PM_{2.5}$ emissions.



Over time, regulation of air pollution has become more stringent and thus more complex and more costly. To achieve acceptance, it had to be demonstrated that the measures would achieve the environmental and climate protection goals safely and efficiently, i.e. with the lowest possible costs and other disadvantages, and that the advantages of environmental protection outweigh the disadvantages (Friedrich et al., 2020). It is a scientific task to support this demonstration, mainly by developing and applying integrated assessments of air pollution control strategies, e.g. by carrying out cost-effectiveness– and cost-benefit-

analyses. With a cost-effectiveness analysis (CEA) the net costs (costs minus monetizable benefits) for improving an indicator used in an environmental aim with a certain measure are calculated, e.g., the costs of reducing the emission of 1 t of $CO_{2,eq}$. The lower the unit costs, the higher the effectiveness of a policy or measure. The CEA is mostly used for assessing the effects associated with climatically active species, as the effects are global. The situation is different for air pollution, where the avoided damage of emitting 1 tonne of a pollutant varies widely depending on time and place of the emission.


The more general methodology is cost-benefit analysis (CBA). In a CBA, the benefits, i.e., the avoided damage and risks due to an air pollution control measure or bundle of measures are quantified and monetized. Then, costs including the monetized negative impacts of the measures are estimated. If the net present value of benefits minus costs is positive, benefits outweigh the costs, thus the measure is beneficial for society, i.e., it increases welfare. Dividing the benefits minus the nonmonetary

costs by the monetary costs will result in the net benefit per € spent which can be used for ranking policies and measures. Of course, for performing mathematical operations like summing or dividing costs and benefits, they have first to be quantified and then converted into a common unit, for which a monetary unit, i.e., Euros, is usually chosen.

The term 'integrated' in the context of integrated assessment means that – as far as possible – all relevant aspects

(disadvantages, benefits) should be considered, i.e., all aspects that might have a non-negligible influence on the result of the assessment. Given the high complexity of answering questions related to managing the impacts of air quality, a scientific approach is required to conduct an integrated assessment, which is defined here as 'a multidisciplinary process of synthesizing knowledge across scientific disciplines with the purpose of providing all relevant information to decision makers to help to make decisions' (Friedrich, 2016).


## 2. Scope and structure of the review

The focus of this review is on research developments that have emerged over approximately the past decade. Where needed, older references are given but these provide either a historical perspective or support emerging work or where no recent references were available. The following areas of air quality research have been examined in this review:

(i)   Air pollution sources and emissions

        (ii)  Air quality observations and instrumentation

        (iii)  Air quality modelling from local to regional scales



(iv) Interactions between air quality, meteorology, and climate

(v) Air quality exposure and health

(vi) Air quality management and policy development

Each section begins with a brief overview and then examines the current status and challenges before proceeding to highlight emerging challenges and priorities in air quality research. In terms of climate research, the focus is more on the interactions between air quality and meteorology with climate and not on climate change *per se*.


The section on air quality observations focuses on new technological developments that have led to remote sensing, lower cost sensors, crowd sourcing and modern methods of datamining rather than attempting to cover the more traditional instrumentations and measurements which are dealt with e.g., in Foken (2021). After considering these themes of research, the section on Discussion pulls together common strands on science and implications for policy makers.


## 3. Air Pollution Sources and Emissions

### 3.1 Brief overview

A fundamental prerequisite of successful abatement strategies for reduction of air pollution is understanding the role of emission sources in ambient concentration levels of different air pollutants. This requires a good knowledge of air pollution 440    sources regarding their strength, chemical characterisation, spatial distribution and temporal variation along with knowledge on their atmospheric transport and processing. In observations of ambient air pollution, typically a complex mixture of contributions from different pollution sources is observed. These source contributions have to be disentangled before efficient reduction strategies targeting specific sources can be set up. Consequently, our discussion below is divided into two main topics: í) Emission inventories and emission pre-processing for model applications and ii) Source apportionment methods and 445    studies.

This paper cannot give a full overview of the status of and the emerging challenges in all emissions sectors. For example, we do not deal with aviation as the impact on air quality in cities is generally rather small or concentrated around the major airports, or with the working and construction machinery or industrial sources which make significant contributions to air 450    pollution in some areas. Instead, we put emphasis on two emission sectors that have experienced important methodology developments in recent years in terms of emission inventories and that are of major concern for health effects: exhaust emissions from road traffic and shipping. We also touch other anthropogenic emissions, e.g., from agriculture and wood burning, As later in this paper we will explain, that individual exposure including the exposure to indoor pollution should gain importance in assessing air pollution, emissions from indoor sources will be addressed in a subchapter. Natural and biogenic 455    emissions encompass VOC emissions from vegetation, NO emissions from soil, primary biological aerosol particles, wind-blown dust, methane from wetlands and geological seepages, various pollutants from forest fires and volcanoes, these are





described in a series of papers edited by Friedrich (2009). As natural and biogenic emissions depend on meteorological data, which is input data for the atmospheric model, they are usually estimated in a submodule of the atmospheric model. They are not further discussed here.


## 3.2 Current status and challenges

### 3.2.1 Emissions inventories

In the European Union, emissions of the most important gaseous air pollutants have decreased during the last thirty years (see Figure 2). $SO_2$ and CO show reductions of at least 60% (CO) or almost 90% ($SO_2$). Also, $NO_x$ and NMVOC emissions

decreased by approx. 50% while $NH_3$ shows much lower reductions of 20% only. Likely to $NH_3$, PM emissions also stay at similar levels compared to 2000 (Figure 2b). Only black carbon shows considerably larger reductions. While traffic is the most important sector for $NO_X$ emissions and an important source for BC, PM emissions stem mainly from numerous small emission units like households and commercial applications (Figure 2c).

In parallel, research came on the path of accompanying and evaluating local emission control measures in a more

comprehensive and systemic approach to urban space. The main technical advances of this research field have consisted in producing a more reliable assessment of the predominant emissions on the scale of an agglomeration / region. This has been done in order to feed the models with activity-based emission data such as population energy consuming practices or local characteristics of road traffic, with the concern to better include their temporal variability or weather condition dependency. The originality of these approaches has been to develop the emissions inventories and modelling efforts in collaboration with

stakeholders, for better data reliability and greater realism in policy support.

a)



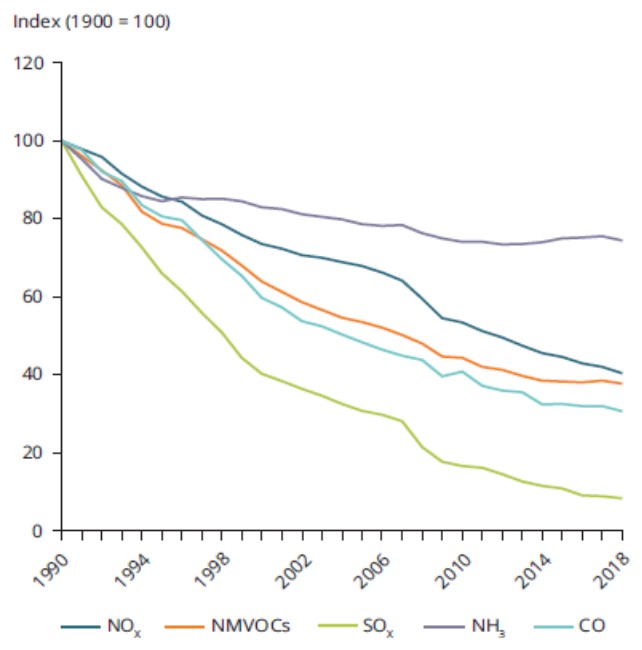

500

b)

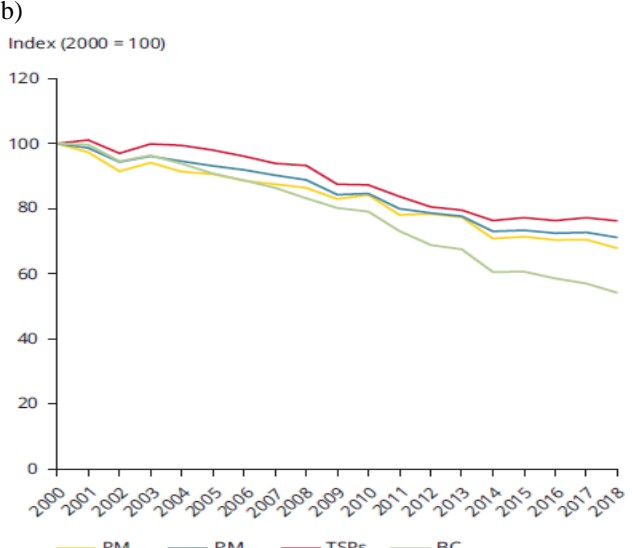



c)

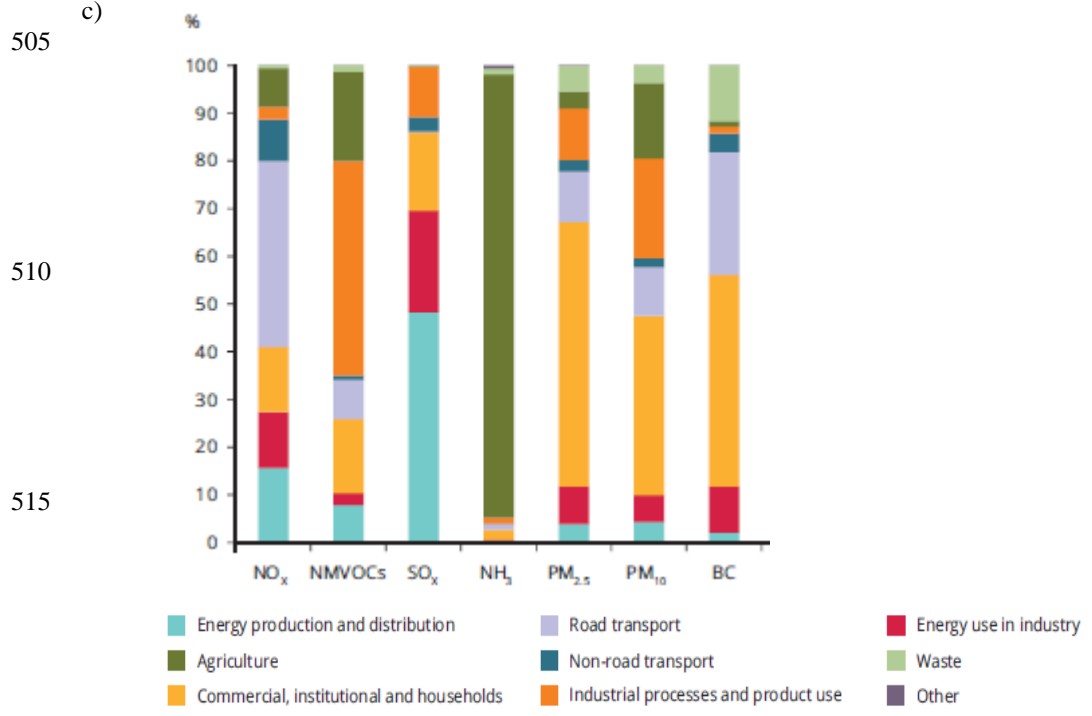

Figure 2. EU-28 emission trends for a) the main gaseous air pollutants, b) particulate matter and c) share of EU emissions of the main pollutants by sector in 2018 (EEA, 2020c)

Improved and innovative representation of emissions, such as real configuration of residential combustion emission sources (location of domestic households using biomass combustion and surveys regarding the characteristics and use of wood stoves, boilers and other relevant appliances) allow more realistic diagnoses (e.g. Grythe et. al., 2019, Savolahti et al., 2019, Plejdrup et al., 2016, Kukkonen et al., 2020b). Also, increased use of traffic flow models for the representation of mobile emissions have provided refined traffic and emission estimates in cities and on national levels, as a path for improved scenarios (e.g. Matthias et al., 2020a). Kukkonen et al. (2016a) presented an emission inventory for particulate matter numbers (PN) in the whole of Europe, and in more detail in five target cities. The accuracy of the modelled PN concentrations (PNC) was evaluated against experimental data on regional and urban scales. They concluded that it is feasible to model PNCs in major cities within a reasonable accuracy, although major challenges remained in the evaluation of both the emissions and atmospheric transformation of PNCs.

For shipping, and in most recent development also aviation, inventories based on position data from transponders on individual vessels are becoming more widely used and provide refined emission inventories with high spatial resolution for use in harbour-city and airport studies (e.g. Johansson et al., 2017, Ramacher et al., 2019, 2020). Refined emission inventory and





emission modelling are in many cases integrated into a complete regional-to-local modelling chain, which allows these refined
data to be taken into account and ensures the consistency of the final results. This links to the subsequent chapters on air quality
and exposure modelling.

*3.2.2 Preprocessing emission data for use in atmospheric models*

Emission inventories usually contain annual data for administrative units apart from data for large point sources and line
sources. Atmospheric models, however, need hourly emission data for the grid cells of the model domain, furthermore the
height of the emissions (above ground), and for NMVOC, PM, and NOx a breakdown into species or classes of species
according to the chemical scheme of the atmospheric model is necessary. For PM, information is also required on the size
distribution. Thus, a transformation of the available data into structure and resolution as needed by the models has to be made
(Matthias et al., 2018).

For the spatial resolution, standard procedures for several emission sectors are described in chapter 7 of the EMEP/EEA air
pollutant emission inventory guidebook 2019 (EMEP/EEA, 2019). In principle, proxy data that are available in high spatial
resolution and that are correlated to the activity data of the emission sources are used. For point sources (larger sources like
power plants) these are coordinates of the stack. For road transport shape files with coordinates at least for the main road
network are used together with traffic counts (for past times) or traffic flow modelling for scenarios for future years. Figure 3
shows as an example the result of a distribution of road transport emissions to grid elements for the EU countries, Norway and
Switzerland. As well the major roads as the urban areas can be identified as sites for the NOx emitters. For households, land
use data (e.g., residential area with a certain density) combined with statistical data (number of inhabitants, use of heating
technologies) is used. Especially for heating with wood specific algorithms using data on forest density and specific residential
wood combustion emission inventories and models have been developed (Aulinger et al., 2011, Bieser et al, 2011a, Mues et
al., 2014, Paunu et al., 2020, Kukkonen et al., 2020b). Thiruchittampalam (2014) contains a comprehensive description of the
methodology for the spatial resolution of emissions for Europe for all emission source categories.

The algorithms for disaggregating annual emission data into hourly data follow a similar scheme. All kinds of available data
containing information about the temporal course of activities leading to emissions are used for temporal disaggregation. For
road transport, data from continuously monitoring the traffic volume are available, statistical data provide the electricity
production from power plants. The activity of firings for heating depends on the outside temperature or more precisely on the
degree days, an indicator for the daily heating demand, together with an empirical daily course of the use of the heating(
(Aulinger et al., 2011, Bieser et al, 2011a, Mues et al., 2014).





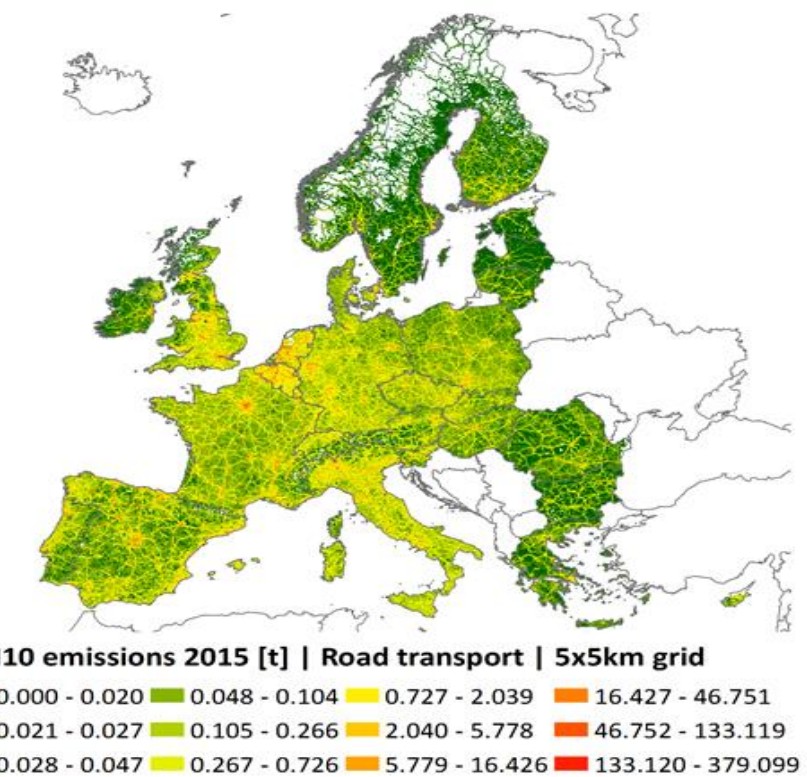

Figure 3. spatial distribution of national PM$_{10}$ emissions from road transport in the EU28 on a 5km*5km grid (Schmid, 2018)

A detailed description of the methodology for the temporal resolution of emission data for all source sectors in Europe is contained in Thiruchittampalam (2014). A compilation of temporal profiles for disaggregating annual into hourly data is published by Denier van der Gon (2011) and in Matthias et al. (2018). New sets of global time profiles for numerous emission sectors have recently been provided by Crippa et al. (2020) and Guevara et al. (2020a). Crippa et al. provide high resolution temporal profiles for all parts of the world including Europe, Guevara et al. developed temporal profiles as part of the Copernicus Atmosphere Monitoring Service and also includes higher resolution European profiles designed for e.g. regional air pollution forecasting. The temporal profiles include time dependent yearly profiles for sources with inter-annual variability of their seasonal pattern, country-specific weekly and daily profiles and a flexible system to compute hourly emissions. Thus, a harmonized temporal distribution of emission is given, which can be applied to any emission database as input for atmospheric models up to the global scale.

For the temporal and spatial distribution of agricultural emissions a number of approaches have been established; these are based on information on farmer practice, available proxy data as well as meteorological data, e.g. farmland and animal densities and the consideration of temperature and wind speed for agricultural emissions (e.g. Skjøth et al., 2011, Backes et al., 2016, Hendriks et al., 2016, see Figure 4).





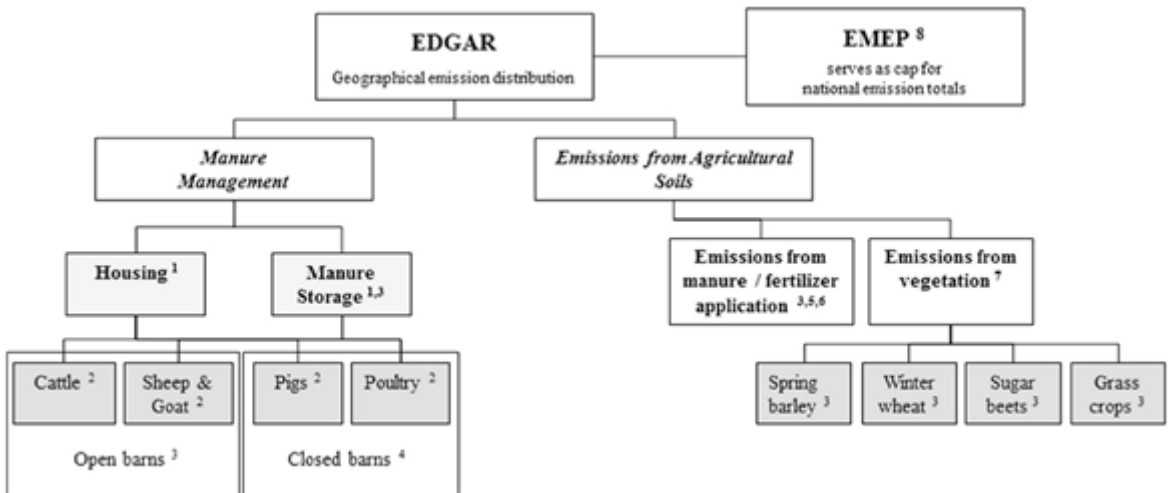

Figure 4. Break-down of agricultural emissions into sub-sectors in order to improve the spatial and temporal distribution (from Backes et al., 2016).

Comprehensive VOC split vectors are provided by Theloke et al. 2007 and more recently by Huang et al. 2017. Region- and source-specific speciation profiles of NMVOC species or species groups are compiled and provided, with corresponding quality codes specifying the quality of the mapping. They can then be allocated to the reduced number of VOC species used in the chemical reaction schemes implemented in atmospheric chemistry transport models. Typical heights for the release of emissions, e.g., typical stack heights, are given by Pregger et al. (2009) and Bieser et al. (2011b).

Model systems have been developed that perform the entire temporal and spatial emission distribution and the NMVOC and PM speciation in order to provide hourly gridded emission data for use in different chemistry transport models. Recent examples are the HERMES model (Guevara et al., 2019, 2020b), FUME (Benešová et al., 2018), and the CEDS model system (Hoesly et al., 2018). Because natural emissions, e.g. biogenic emissions, sea spray and dust depend strongly on the meteorological conditions, these emissions are frequently calculated within the chemistry transport model systems (CTMs).

Other established CTMs like the EMEP model (Simpson et al., 2012) or LOTOS-EUROS (Manders et al., 2017) don't use emissions preprocessors, but distribute gridded emissions in time based on standard temporal and speciation profiles alongside with the chemistry transport calculations in order to avoid storing and reading large emission data sets.

### 3.2.3 Road transport emissions

Exhaust emissions from road transport have been a significant source of primarily $NO_X$ and ultrafine particles (UFP) in urban areas around the world. In the EU, road transport is the single most important source of $NO_X$, producing 28.1% of total $NO_X$ emissions (EEA, 2020a). In terms of $PM_{10}$, its contribution is 7.7% when both exhaust and non-exhaust sources are counted and 2.9% when only exhaust emissions are considered (EEA, 2020a). Road transport contributes 32-97% of total UFP in urban





areas (Kumar et al., 2014). The difference between $PM_{10}$ and UFP contributions from road transport is a direct outcome of the
small size of exhaust particles that mostly reside in the UFP range (Vouitsis et al., 2017).

The proximity of people to the emission source (vehicles) significantly increases exposure to traffic-induced pollution (Zak et al., 2017). Consequently, traffic exhaust emissions have been extensively studied and comprehensive sets of emission factors have been available since long. The two most widespread methods to estimate emissions in Europe include COPERT
(emisia.com/utilities/copert) and HBEFA (www.hbefa.net). These methods share the same experimental database of vehicular emissions - the so-called ERMES database (ermes-group.eu) - but express emission factors in different modelling terms. COPERT is as well part of the EMEP/CORINAIR Emission Inventory Guidebook (EMEP/EEA, 2019).

These models define the emissions for several pollutant species, for a wide range of vehicles and operating conditions.
Emission factors are regularly being updated in an effort to reflect the best knowledge of on-road vehicle emission levels. Despite this, there are still some uncertainties in estimating emissions from road transport, in particular when these are to be used as input to air quality models. More attention is therefore needed in the following directions:

(i)    Emission factors for latest vehicle technologies always come with some delay. This is the result of the time lag
between placement of a new vehicle technology on the road and the organisation of measurement campaigns to collect the experimental information required to develop the emission factors. The latest regulation (Reg. (EU) 2018/858) - mandating a minimum number of market surveillance tests in the different member states - may help to reduce this lag and to extend the availability of vehicle tests on which to base emission factors.

(ii)   The availability of measurements of pollutants which are currently not included in emissions regulations ($NH_3$, $N_2O$,
$CH_4$, PAHs, etc.) is limited compared to regulated pollutants. Moreover, any available measurements have been mostly collected in the laboratory, due to instrumentation limitations for on-road measurements. Therefore, emission models may miss on-road operation conditions that potentially lead to high emissions rates of non-regulated pollutants.

(iii)  The increase of emissions with vehicle age is still subject to high uncertainty. Emission increases with age may be
due to normal system degradation, the presence of high emitters on the road (Murena and Prati, 2020) or vehicle tampering to improve performance or decrease operational costs. Current models use degradation functions based on remote sensing data (e.g. Borken-Kleefeld and Chen, 2015). This is a useful source of information, but remote sensing data need to be collected in additional locations in the EU, covering a range of climatic and operation conditions.

(iv)   Emission models may be conservative in their approach of estimating emissions in extreme conditions of temperature
(Lozhkina et al., 2020), altitude, road gradient or creeping speeds. Although such conditions may not be substantial for estimating the total emissions of most countries, they can potentially lead to a significant underestimation of emissions that have to be locally calculated for high-resolution air quality modelling.





Despite uncertainties in modelling emissions, there is a high level of confidence that exhaust gas emissions of mobile sources
will continue to decrease in the years to come. For example, Matthias et al. (2020b) projected that the contribution of road
traffic to ambient $NO_2$ concentrations will decrease from 40-60% in 2010 to 10-30% in 2040. This is the result of relevant
technological development driven by demanding $CO_2$ reduction targets and air pollutant emission standards applicable to new
vehicles. An example of such technological development is the increase in the availability of plug-in hybrid vehicles, which
have exhibited great potential in reducing both pollutant emissions and $CO_2$ emissions from traffic (Doulgeris et al., 2020).

Technological improvement in decreasing emissions from internal combustion engines will be accelerated in the EU market
due to the current Euro 6d emission standard and the upcoming Euro 7 regulation but also the proliferation of electric
powertrains to meet $CO_2$ targets. The only road transport pollutant not significantly affected by the introduction of electric
vehicles is non-exhaust PM coming from tyre, brake and road wear with estimates suggesting both increases due to heavier
vehicles and reductions due to wider exploitation of regenerating braking systems (Beddows and Harrison, 2021).

New techniques are also being developed with the capacity to monitor emissions of vehicles in operation. This can verify that
emissions remain below limits in actual use and not just in type approval testing conditions. A current example of such on-
board monitoring systems is the on-board fuel consumption measurement (OBFCM) device which is already mandatory for
new light duty vehicles and is being extended for heavy-duty vehicles (Zacharof et al., 2020). Information from such systems,
together with new computation methods (big data) can provide very useful information for improving the reliability, and
temporal and spatial resolution of current emissions inventories.

*3.2.4 Shipping emissions*
Ships consume high amounts of fossil fuels, on the global scale they emit comparable amounts of $CO_2$ as big industrialized
countries like Germany and Japan. Because ships use high sulphur fuels, regardless of the global introduction of 0.5% sulphur
cap in 2020, and typically are not equipped with advanced exhaust gas cleaning systems, their share from global $CO_2$ is 2.9%,
but corresponding shares of $NO_X$ and $SO_X$ are considerably higher, 13% and 12%, respectively (IPCC, 2014, Smith et al.,
2014, Faber et al., 2020). Ship routes are frequently located in the vicinity of the coast, this may go along with significant
contributions to air pollution in coastal areas. Effects on ozone formation and secondary aerosol formation also need to be
considered.

The environmental regulation concerning the sulphur emissions from ships has been in place in the Baltic Sea since 2006, with
the North Sea following in 2007. Currently, also North America and some Chinese coastal areas have stringent sulphur limits
for ship fuels. Everywhere else the use of high sulphur fuel in ships was allowed until the start of 2020, when sulphur reductions
of maximum 0.5% S were extended to all ships (IMO, 2016). This has been estimated to reduce the premature deaths by 137

000 each year (Sofiev et al., 2018). Nitrogen oxide emissions from ships are regulated by $NO_X$ ECAs, which currently exist only in the coastlines of Canada and the US. The Baltic Sea and the North Sea areas will quickly follow, because in 2021 all new ships sailing these areas must comply with 80% $NO_X$ reduction.

The introduction of Automatic Identification System (AIS), Long-Range Identification and Tracking (LRIT) and Vessel Monitoring Systems (VMS) have enabled tracking of individual ships at unprecedented detail. These navigational aids offer an excellent description of vessel activities on both local and global scales.

Currently, ship emission models using AIS data as an activity source are most popular. They can have accurate information about quantity, location and time of the emissions. Most of the model systems applied nowadays use a bottom-up approaches to calculate shipping emissions (e.g. Jalkanen et al., 2009, 2012, 2016, Johansson et al., 2017, Aulinger et al., 2016). The

combination of vessel activity, technical description and an emission model allows for prediction of emissions for individual ships. This also facilitates comparisons to fuel reports, like those of the EU Monitoring, Reporting, Verification (MRV) scheme or IMO Data Collection System (DCS). Emission models may also include external contributions, like wind, waves, ice or sea currents in vessel performance prediction, which brings them closer to realistic conditions experienced by ships than the assumptions applied for ideal conditions (Jalkanen et al., 2009; Yang et al., 2020). Vessel level modelling approach allows for

very high spatio-temporal resolution and flexible 4D-grids (lat, lon, height, time) on which the data can be given. New information about modified or new emission factors for certain chemical species can easily be adopted in the models. Ship emission data is available on a global grid on 0.1° x 0.1° and in higher resolution for regional domains in Europe (see Figure 5), North America and East Asia (e.g., Johansson et al., 2017).

Emissions from ships in ports can be quantified for arrival and departure following the same AIS based approach as for regional

and global shipping emissions. Emissions for ships at berth are estimated based on ship type and size, but with large uncertainties.

Introduction of emission limits gives shipowners a choice to comply with through at least three options. The first of these is the use of low sulphur fuels, the second option involves the use of aftertreatment devices ($SO_X$ scrubbers) which remove air pollutants by spraying the exhaust with seawater. The third option probably applies only to new ships, because it involves the

use of Liquid Natural Gas (LNG) as a marine fuel.

Exhaust aftertreatment systems, which are commonly used to remove $NO_X$, $SO_X$ or PM often involve chemical additives (urea, caustic soda) or large amounts of seawater. Use of so-called open loop $SO_X$ scrubbers, which use seawater spray to wash the ship exhaust releases the effluent back to the sea. This may lead to a creation of a new water quality problem, especially in areas where water volumes are small (estuaries, ports) or water exchange is slow (e.g., the Baltic Sea) (Teuchies et al., 2020).





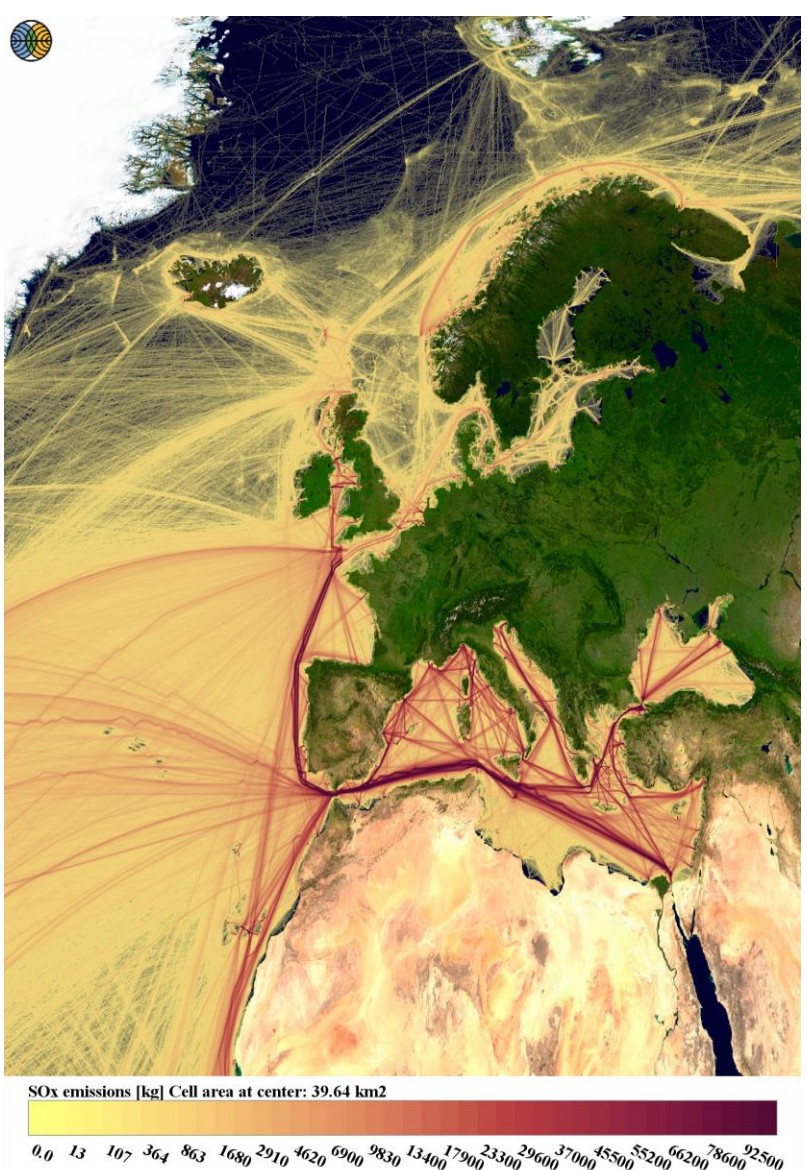


Figure 5: The predicted SO$_X$ emissions from ships in Europe in 2018, computed using the STEAM model (e.g., Johansson et al., 2017). Use of low sulphur fuels and SO$_X$ scrubbers is concentrated to the North Sea and Baltic Sea ECAs. Background map © US Geological Survey, Landsat8 imagery.

The use of low sulphur or LNG as fuel are fossil-based solutions, unless the fuel was made using renewable or fully synthetic

sources. However, emissions of NO$_X$, SO$_X$ and PM from LNG engines can be very low, but this depends very much on the engine type selected.



Methane, methanol and ammonia are three fuels which can be produced by fossil, bio and synthetic pathways. These three fuels are also suitable for use in internal combustion engines as well as fuel cells. All three are hydrogen carriers and processes, which lead to synthesis of these three fuels have hydrogen production as an intermediary step. This could offer a viable pathway
towards hydrogen-based shipping, but also allows the use of current engine setups and existing fuel infrastructure (DNV-GL, 2019).

*3.2.5 Emissions of indoor sources*

The shift in focus from regulating the outdoor concentration of pollutants to putting more emphasis on reducing the individual exposure to pollutants, which is described later in chapters 7.4 and 8 of this paper, makes it necessary to analyse not only
possibilities for reducing emissions from outdoor sources, but also those from indoor sources. Thus, detailed knowledge about emission factors from indoor sources is needed.

Smoking, combustion appliances and cooking are important sources of $PM_{2.5}$, NO, $NO_2$ and PAHs (Hu, 2012, Li, 2020, Weschler and Carslaw, 2018). Particularly important indoor sources of NO and $NO_2$ are gas appliances such as stoves and boilers (Farmer et al., 2019). For $PM_{2.5}$, apart from diffuse abrasion processes, passive smoking is still the most important
source, although the awareness that passive smoking is unhealthy has been increasing with the EU ban of smoking in public buildings. Schripp et al. (2013) report, that not only smoking, but also consuming e-cigarettes leads to a high emission of VOCs and fine and ultrafine particles. Frying and baking lead to the evaporation and later condensation of fat and is a large source for $PM_{2.5}$, especially if no kitchen hood is used; a larger number of studies on frying is available and listed in Li (2020) and Hu et al. (2012). Hu et al. (2012) reviewed emissions of $PM_{2.5}$ from the use of candles and incense sticks and found that
incense sticks have much higher emission rates than candles. Zhao et al. (2020a) measured simultaneously indoor and outdoor concentrations of PM in homes in Germany and report abrasion and resuspension processes as major contributors of coarser particles ($PM_{2.5-10}$) and toasting, frying, baking and burning of candles and incense sticks as important sources for ultrafine particles. Also, the use of open chimneys and older wood stoves in the living area is an important source. For wood stoves, mostly measured indoor concentrations of $PM_{2.5}$ are used to characterize the pressure coming from indoor emissions, or
emissions are estimated as a fraction of the overall emissions of a stove. As only few studies measuring emissions from wood stoves into the interior exist (Li et al., 2019b, Salthammer et al., 2014), more measurements are necessary. Schripp et al. (2014) report very high emission factors of ethanol burning fireplaces, as these have no chimney.

Laser printers are emitting ultra-fine particles, especially longer-chained alkanes (C21-C45) and siloxanes (Morowska, 2009). Also the new 3D-printers are a source of nanoparticles, as Gu et al. (2019) found out. Schripp et al. (2011) analysed the
emissions from electric household appliances and reported high emission rates especially from toasters, raclette grills, flat irons and hair blowers.

New furniture is often a source of formaldehyde. The use of chemicals such as cleaning agents and personal care products leads to VOC and SVOC emissions, that partly are oxidized and condensate and thus transformed into fine particles. McDonald et al. (2018) point out that with rapidly decreasing emissions of VOC from transportation, emissions from the use of volatile





chemical products indoors are becoming the dominant sources in the urban VOC emission inventory, so that VOC concentrations often are higher indoors than outdoors (Kristensen et al., 2019).

Excreta of house dust mites, use of fan heaters, vacuum cleaning, especially without HEPA filters and pets are further indoor emission sources. Furthermore, all kinds of human activities produce abrasion. As there are numerous different processes causing these emissions, instead of estimating emissions measured concentrations, that are typical for stemming from abrasion

processes are used.

Apart from reducing emissions, the concentration of pollutants indoors can also be reduced by ventilation, i.e. by opening windows or using mechanical ventilation, or by filtering the air, e.g. with HEPA filters for the removal of fine particles.

*3.2.6. Source apportionment methods and studies*

The question of how much are the different sources contributing to the ambient levels of different air pollutants is critical for the design of effective strategies for urban air quality planning. Different methods are used for source apportionment of ambient concentrations, each including certain limitations given by the intrinsic assumptions underpinning the individual methods and by availability and robustness of data underpinning the source apportionment. In many cases these methods are complementary to each other and implementation of a combination of different methods decreases the uncertainties (Thunis et al., 2019). There

are two principally different source apportionment models: the receptor models apportioning the measured mass of an atmospheric pollutant at a given site to its emission sources, and the source-oriented models based on sensitivity analyses performed with different types of air quality models (Gaussian, Lagrangian, or Eulerian chemistry transport models) (Viana et al., 2008, Hopke, 2016, Mircea et al., 2020). Another method addressing the source-receptor relation of air pollution is inverse modelling used for improvement of emission inventories from global scale to individual industrial sources (e.g. Stohl et al.,

2010, Henne et al., 2016, Bergamaschi et al., 2018).

The main receptor models are the incremental (Lenschow) method, the chemical mass balance (CMB) method and the positive matrix factorisation (PMF) (Mircea et al., 2020). The Lenschow method is based on the assumption that source contributions can be derived from the differences in measured concentrations at specific locations not affected and affected by the emission

sources. This approach is based on the assumptions that the regional contribution is constant at both locations and that the sources do not contribute to the regional background. The CBM is based on known source composition profiles and measured receptor species concentrations. The result depends strongly on the availability of source profiles, which ideally are from the region where the receptor is located and that should be contemporary with the underpinning ambient air measurements. PMF is the most commonly used analytical technique operating linear transformation of the original variables to create a new set of

variables, which better explain cause-effect patterns. Hopke (2016) provides a complete review of receptor models.

The source-oriented apportionment methods utilising source-specific gridded emission inventories and air pollution models include two in principal different methods, the wide used Sensitivity Analysis, also called Brute Force Method or Emission



Reduction Potential (Mircea et al., 2020) or Emission Reduction Impact (ERI) method (Thunis et al., 2019) and the Tagged
Species methodology which involves computational algorithms solving reactive tracer concentrations within the chemistry
transport models. ERI and Tagged Species methods are conceptually different and address different questions. Generally, the
ERI method analyses how the concentrations predicted by an air quality model respond to variations in input emissions and
their uncertainties. An important aspect to consider when using this method is that the relationship between precursor emissions
and concentrations of secondary air pollutants may include non–linear effects. In non-linear situations, the sum of the
concentrations of each source is different from the total concentration obtained in the base case. The magnitude of the emission
variations considered in ERI may vary from small perturbations, studying the model response in the same chemical and
physical regime as the base case, to removing 100% of the studied emissions (the zero-out method) which may include non-
linear effects present in the model response (Mircea et al., 2020). The Tagged Species method is based on CTM simulations
with tagging/labelling technique which keeps track of the origin of air pollutants trough the model simulation. This
accountability makes it possible to quantify the mass contributed by every source or area to the pollutant concentration (Thunis
et al., 2019).

The principal differences between the different source-apportionment methods and implications of these differences on
apportionment of sources to the observed or modelled ambient concentration levels are in detail explained and discussed in
Clappier et al. (2017) and Thunis et al. (2019). Belis et al. (2020) evaluated 49 independent source apportionment results
produced by 40 different research groups deploying both receptor and source-oriented models in the framework of the
FAIRMODE intercomparison study of PM$_{10}$ source apportionment. The results have shown good performance and
intercomparability of the receptor models for the overall dataset while results for the time series were more diverse. The source
contributions of the source-oriented models to PM$_{10}$ were less than the measured concentrations.


In this section we further focus on new developments in source characterisation with help of receptor-oriented models and in
construction of emission inventories while the air quality models and emission sensitivity studies are subject of Section 5 of
this paper. Several new studies reported on characterisation of local composition of particulate matter as well as of NMVOCs
and PAHs tracking contribution of main emission sources (Christodolou et al. 2020; Diémoz et al., 2020; Saraga et al., 2020;
Liakakou et al., 2020; Kermenidou et al., 2020). The particulate matter has been characterised in terms of carbonaceous matter
– elemental or black and organic carbon, organic matter, metals, ionic species as well as its elemental composition.
Aethalometer model to identify BC related to fossil fuel combustion and biomass burning has been applied in several studies
(Grange et.al., 2020; Christodolou et al., 2020; Diémoz et al., 2020). Combination of the different analytical methods and
analysis of temporal and spatial variation of the data allowed for identification of chemical fingerprints of different emission
sources. Belis et al. (2019) present a multistep PMF approach where high time-resolution dataset from Italy of aerosol organic
and inorganic species measured with several online and offline techniques gave internally consistent results and could identify
additional emission sources comparing to earlier studies.



The local studies characterising the local composition of PM, as well as NMVOCs and PAHs revealed the important roles of
road traffic and residential combustion for concentration levels of air pollutants in both urban and rural areas. Wood burning
has an important share in many residential areas, especially those outside the city centres and in the countryside (Saraga et al.,
2020; Fameli et al., 2020). Fuel oil is another important fuel in residential combustion, in some cities as e.g. Athens the
dominating one (Fameli et al., 2020). The studies show important differences in the diurnal and seasonal patterns of these two
emission sources. While road traffic emissions have maxima in the morning and afternoon hours, contributions from residential
combustion dominate at night-time and in the cold season. Important contributions of traffic are found in all studies. Maggos
et al. (2020) shows, as results from the ICARUS study performed in 6 European cities, that the main contribution to road traffic
related $PM_{2.5}$ is the tyre and brake wear and resuspension of the particles. Fuel oil combustion source is, apart from residential
heating, also associated with industrial emissions and shipping emissions. Contributions from these sources become important
at specific locations, as e.g. in cities with certain industrial plants or in harbour cities.

Analyses of data from longer time-series show a decreasing trend for exhaust gas emissions in road traffic. Its contributions to
BC in the last decade decreased while the residential combustion, especially the wood burning contribution does not show any
clear trend (Grange et. al. 2020). Efficient abatement measures for improvement of the local air quality need to address the
important sources. In most cases these are the local traffic and residential combustion, but in many cases these also include
industrial sources and in some cases shipping. Targeting these different sources requires a different approach for each.

Inverse modelling is mainly used for improvement of emission inventories with help of measurements. Different inversion
methods applied in Lagrangian dispersion models (e.g. Stohl et al., 2010, Manning et al., 2011, Henne et al., 2016) and global
and regional Eulerian models have been widely used for improvements of emission inventories of greenhouse gases on a wide
range of geographical scales from global over national to urban and local. An overview of different inverse modelling
approaches applied on European $CH_4$ emission inventory is presented by Bergamaschi et al. (2018). The inverse modelling
has potential to reduce uncertainties of emission inventories comparable to other approaches, e.g. incremental method
combining aircraft measurements and high-resolution emission inventory (Gurney et al., 2017).

**3.3 Emerging challenges**

3.3.1 *Emission inventories and preprocessors:*

Emission inventories have still large uncertainties. In particular, PM emissions stemming from all kinds of diffuse processes,
especially from abrasion processes in industry, in households, agriculture, and in traffic show a large variability and
uncertainty. For example, abrasion processes of trains may cause very large PM concentrations in underground train stations,
but emission factors and total emissions are not well known. With the ongoing reduction of exhaust gas emissions and the
continuing introduction of electric vehicles, abrasion will become the most important process for traffic emissions.




For residential wood combustion many uncertainties relate to the quality and refinement of information about the use of wood and the heating device technologies, tree species, wood storage conditions or combustion procedures implemented. Their impact on emission inventories is not well evaluated, but new research underlines how national characteristics need to be taken
into account and also shows what type of data that can be used in order to improve the spatial representation of these emissions. Despite the activities to improve temporal profiles of agricultural emissions, more detailed information about the amount of $NH_3$ and PM emissions are still needed for many regions of the world.

Chemical composition of NMVOC emissions from combustion processes remains highly uncertain, especially when new fuels enter the market like low sulphur residual fuels in shipping or when new exhaust gas cleaning technologies are introduced that
modify the chemical composition of the exhaust gas. Advanced instrumentation for the characterisation of new emission profiles are needed, here. Measurement techniques employed in the characterization of emissions impact the results, for example, the dilution methods used have a large impact on the measured gas to particle partitioning. Better understanding of these impacts and a robust assessment of the uncertainties and variabilities remains a challenge. Emission inventories should include air pollutants and greenhouse gases at the same time. Integrated assessments analyse measures and policies targeting
air pollution control as well as climate protection at the same time and potential and their co-benefits need to be investigated.

Emissions preprocessors aim at increasing the level of detail they take into account for calculating the spatial and temporal resolution of emissions. However, the availability of input data sets (e.g. traffic data from mobile phone positions, AIS ship position data), the huge size of these data sets and also data protection rules currently hinder their use. Still, there is big potential
in extending the data sources used for emissions preprocessing towards big data, e.g. from mobile phone positions, traffic counts, or online emission reporting, in order to reach real-time emission data and improved dynamic emission inventories to be used in air quality forecast systems. Monitoring data from numerous air quality sensors at multiple locations might help in advancing these inventories.

*3.3.2 Road emissions:*

The accuracy and relevance of our current emission estimation and modelling approaches may in the future be challenged by relevant developments, the most important ones being the following:

    (i)   The exhaust emissions from road transport are continuously decreasing, as exhaust filters become increasingly efficient and are used in a wider range of vehicle technologies, including gasoline vehicles, while the market share of
electric cars is also increasing. However, $PM_{2.5}$, $PM_{10}$ and heavy metal emissions from wear and abrasion processes increase with increasing traffic volume as they are not regulated and electric cars also produce emissions from tyre wear road abrasion. For instance, the emissions of $PM_{2.5}$ reported by Germany to the EEA for 2018 show 9.9 kt/a for exhaust gases of cars, trucks, and motorcycles, 7.6 kt for tyre and brake wear and 4.3 kt from road abrasion. A scenario reported by Germany for 2030 shows only 2.0 kt $PM_{2.5}$ for exhaust emissions, but 7.9 kt from tyre and brake wear
and 4.4 kt from road abrasion (EIONET, 2019). Emissions from wear of tyres and brakes and abrasion of road surfaces



are less studied than exhaust emissions. Wear emissions depend on a range of parameters including driving behaviour (acceleration and braking pattern), vehicle weight and loading, structure and material of brakes and tyres, road surface material and weather conditions (e.g. road water coverage) (e.g., Denby et al. 2013, Stojiljkovic et al., 2019, Beddows and Harrison, 2021). Capturing the effect of technological developments in this area would be therefore important for
relevant air quality estimates.

(ii) The profile of non-methane organic gases (NMOG) is important to estimate the contribution of exhaust to secondary organic aerosol formation. NMOG depend on fuel and lube oil use, combustion, aftertreatment, and operation conditions. The profile of emission species may be differentiated as new fuels, including renewable, oxygenated and other organic components are being increasingly used to decarbonize fuels. Hence, although total hydrocarbon
emissions are still controlled by emission standards, the speciation of these emissions may vary in the future. Monitoring those changes is cumbersome as the study of the chemistry and/or volatility of organic species is a tedious and expensive procedure. Hence any changes may escape relevant experimental campaigns.

(iii) Questions remain about the suitability of widespread emission factors and models to capture the effects of lane layouts, vehicle interactions and driving behaviour, while lane-wide average traffic parameters are a structural
limitation to emission modelling. As urban policies are advancing in an effort to decrease the usage of private vehicles in cities, the impact of traffic calming and banning measures may not be satisfactorily captured by today's available emission models. In order to take driving behaviours into account, it is necessary to improve so-called microscopic models such as the "Passenger car and Heavy duty Emission Model" (PHEM) (Hausberger et al. 2003) that calculate emissions from high temporal frequency information on network configuration as well as traffic and driving
conditions (see review by Franco et al. 2013). Their use calls for the development of new methodologies to provide the simulation with individual speed profiles, taking into account the actual road usage and the specificities of the emissions of the most recent vehicles.

### 3.3.3 Shipping emissions

The efforts of decarbonising shipping have thus far concentrated on minimising the energy need of ships, but a shift to carbon-neutral or non-carbon fuels is necessary. Methane, methanol and ammonia are three fuels that could offer a viable pathway towards hydrogen-based shipping, but also allow for the use of current engine setups and existing fuel infrastructure (DNV-GL, 2019). Regardless of the fuel or aftertreatment technique used, detailed emission factor measurements for various combinations of fuels and engines are needed (Anderson et al., 2015, Winnes et al., 2020) to reliably model the emissions.

Little is known about emissions of VOCs from ships and how much they contribute to particle formation and ozone formation. VOC emissions from ships aren't included in most ship emission models, because emission factors are not available or stem from comparably old observations. In addition, VOC emissions are expected to vary considerably with the type of fuel burned and the lubricants used on board, both of which have changed considerably with the introduction of low sulphur fuels in 2015 (in ECAs) and in 2020 (on a global level). The most recent greenhouse gas emission report from IMO (2020) states that





evaporation might be the most important source for VOCs from shipping, which is not considered in any emission inventory, yet.

Current exhaust gas cleaning technologies, in particular scrubbers applied for removing $SO_2$ from the ship exhaust, dump large parts of the scrubbed pollutants into the sea. More comprehensive research is therefore urgently needed on the combined effects of shipping, which will treat both the impacts via the atmosphere and those on the marine environments. The impacts via the

atmosphere include the health effects on humans, the deposition of pollutants to the sea and climatic forcing. The impacts on the marine environment include acidification, eutrophication, accumulation of pollution in the seas and the impacts on marine biota. Recently, there have been attempts to combine the expertise of oceanic and atmospheric researchers for resolving these issues (Kukkonen et al., 2020a).

Ships have high emissions when they arrive in ports and also when they depart short time later. In addition, they need electricity

and heat when they stay at berth leading to additional emissions in ports stemming from their auxiliary engines and boilers. The impact of these emissions on urban air quality in port areas is of high interest because of their large impact on human exposure.

### 3.3.4 *Indoor sources*

Even though people in industrialised countries spend more than 80% of their time indoors, systematic knowledge on indoor air quality, source strength of the indoor air pollution sources and physico-chemical transformation of indoor air pollutants is still limited. Therefore, systematic quantification of different indoor air pollution sources, such as building material, consumer products and human activities is needed, including exploitation of the already existing test-chamber and other relevant laboratory data is needed. Special attention is also needed to the outdoor source component. Besides obtaining new data on

indoor to outdoor (I/O) ratios, the existing data need to be systematically analysed. One of the key challenges here is how to translate such data into outdoor contribution into a real indoor environment with considerable heterogeneity in terms of ventilation, volume, microclimatic characteristics and multiple indoor sources (Bartzis et al, 2015).

Development of indoor air quality models with accurate description of the key chemical and physical processes involved in outdoor - indoor air interaction as well as processing and transport of indoor air pollution inside the buildings is needed to

properly address connection between the outdoor air quality and indoor air pollution sources. Additional advanced modelling is needed for air-surface interactions targeting emissions/sinks on different surfaces including those in the ventilation set-up (Liu et al, 2013) along with verification of the indoor-air models with measurements in a variety of indoor air environments.

### 3.3.5 *Source apportionment*

Continuous improvement of emission inventories with help of verification with source- and receptor-oriented source apportionment methods is needed, especially as large changes in emissions, both in terms of the emission totals and profiles of emission species from individual sources are expected as a result of upcoming new technologies, fuels and changes in lifestyle emerging mainly from the Paris agreement climate change targets.



Currently, apportionments of the overall measurement datasets usually give consistent results while source apportionment of data with high temporal resolution still remain challenging. With rapid development of both advanced online measurement instruments and low-cost measurement sensors, development of source apportionment methods towards high temporal resolution data and increasing number of parameters is necessary. This also requires improvements in characterisation of sources both in terms of speciation and temporal profiles. This in particular concerns emission profiles for NMVOCs, PAHs and particulate organic matter (e.g., most existing profiles for PAH emission from vehicles are quite old and do not follow vehicle technologies evolution, Cecinato et al., 2014; Finardi et al., 2016). Inverse modelling methods are very powerful and promising tools for source estimation and improvement of emission inventories, but the currents models provide large spreading in results and need to be further improved and intercompared.

## 4. Air quality observations and instrumentation

Here we concentrate on another growing field of development: low-cost sensor (LCS) networks, crowdsourcing and citizen science together with small-scale air quality model simulations to provide personal air pollution exposure. Modern satellite and remote sensing techniques are not in focus here.

### 4.1 Brief overview

Europe's air quality has been improved over the past decade. This has led to a significant reduction in premature deaths over the same period in Europe, but all Europeans still suffer from air pollution (EEA, 2020a). The most serious air pollutants, in terms of harm to human health, are particulate matter (PM), $NO_2$ and ground-level ozone ($O_3$). The analysis of concentrations in relation to the defined EU and World Health Organization (WHO) standards is based on measurements at fixed monitoring points, officially reported by the Member States. Supplementary assessment by modelling is also considered, particularly when it results in exceeding the legislated EU standards. But in parallel new monitoring techniques and strategies for observation of ambient air quality are available and applied which are discussed below.

The motivations for new developments in observation and instrumentation are, on the one hand, obtaining necessary information about air pollutant concentrations and exposure as a basis for compliance and health protection measures, on the other hand supporting improvements in weather, climate as well as air quality forecasts. Remote sensing techniques are developed further either to get 3-D coverage of observations globally by establishment of networks as e.g., with mini-LIDAR (so called ceilometers), for evaluation of satellite measurements, to contribute to atmospheric super sites (extension of in situ measurements), or for chemistry-transport model (CTM) evaluations. These techniques can provide nearly continuous monitoring data, only interrupted by certain weather conditions. Satellite measurements are becoming more important for air quality management because their spatial resolution can reach down to 1 km, while their information content is suitable for the assessment of modelling results and combination with modelling tasks (Hirtl et al, 2020). All these techniques enable unattended detection at different altitudes and thus of the composition, clouds, structure, and radiation fluxes of the atmosphere as well as Earth surface characteristics, relevant for atmosphere-surface feedback processes.





Some examples of modern remote sensing techniques as described in Foken (2021) are the sun photometer networks (determination e.g., of aerosol optical depth), MAX-DOAS (e.g., $NO_2$ and HCHO column densities), LIDAR (e.g., water vapour, temperature, wind, and air pollutants) and more recently ceilometers (e.g., cloud altitude and mixing layer height).

Machine learning algorithms, such as neural networks, are now deployed for remote sensing applications (Feng et al., 2020). Satellite observations have become available for column densities of e.g., aerosol, $NO_2$, CO, HCHO, $O_3$, $PM_{10}$, $CH_4$ and $CO_2$ as well as aerosol optical depth and various image analyses (Foken, 2021). Together with improved spatial coverage and high resolution, these data become increasingly important for assessment in urban areas (Letheren, 2016).

The distribution of ambient air composition exhibits large spatial variations, therefore high-resolution measurement networks

are required. This has become possible with LCS networks, which are used in both research and operational applications of air pollution measurement and in global networks of observations such as the World Meteorological Organisation (WMO) Global Atmosphere Watch (GAW) program (Lewis et al., 2017). WMO/GAW (Global Atmosphere Watch Programme | World Meteorological Organization (wmo.int)) is addressing atmospheric composition on all scales from global and regional to local and urban (see GAW Station Information System https://gawsis.meteoswiss.ch/GAWSIS/#/) and thus, providing information

and services on atmospheric composition to the public and to decision-makers which requires quality assurance elements and procedures as described by the WMO/GAW Implementation Plan: 2016-2023 (WMO, 2017). This topic is further discussed with respect to the related sensor, network, and data analysis requirements.

**4.2 Current status and challenges**

To describe the current trends of air quality monitoring, certain lines of research and technical development are formulated in the following section. This section concentrates on high-resolution measurement networks by the installation of a larger number of small and lower cost measurement sensors. The measurements by traditional in situ measuring as well as ground-based, aircraft-based, and space-based remote sensing techniques or integrated measuring techniques are no more considered. Also, satellite observations, which are a growing field of development towards even smaller and thus cost-effective platforms,

are not in the focus here.

The configurations of ambient air measurements can be described as a space, time, and precision-dimensional feature space shown as large arrows in Figure 6 where crowds with LCS (green) are distributed irregularly in space and time at low precision and high number. Stationary measurements (yellow) are performed at high precision and thus of the highest quality as well as continuously over time, but only at a few points in space requiring high effort and cost. Between the two layers, mobile

measurements are available on a medium level of precision: in one case regularly on certain routes (red) and in another case with high spatial density at a few points during intensive measurement campaigns (blue). The crowd measurements by LCS can be geo-statistically projected onto a higher quality level together with the high-precision measurements (thin black arrows). Following this, an overall higher information density at an elevated quality level than the sum of the individual measurements alone is possible, so that continuous data by LCS can be applied (Budde et al., 2017).





There is an increasing interest in air quality forecast and assessment systems by decision makers to improve air quality and public
health, mitigate the occurrence of acute air pollution episodes, particularly in urban areas, and reduce the associated impacts on
agriculture, ecosystems, and climate. Current trends in the development of modern atmospheric composition modelling and air quality
forecast systems are described in review by Baklanov and Zhang (2020) which are for instance the multi-scale prediction approach,
multi-platform observations and data assimilation as well as data fusion, machine learning methods and bias correction techniques.
This shows the general development towards spatial and temporal high-resolution as well as better knowledge of personal air pollution
exposure.

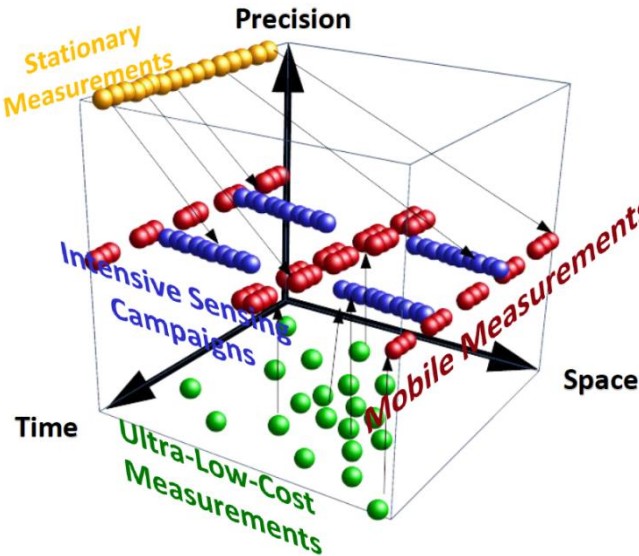

Figure 6. Configuration of ambient air measurements modelled as a space, time, and precision-dimensional feature space (large
arrows): crowds with low-cost sensors (green) scatter irregularly in space and time at low precision, however high number
(source: Budde et al., 2017).

### 4.2.1 *Low-cost sensors and citizen science for atmospheric research*

Many manufacturers (more than 50 worldwide, with their numbers growing fast) are working in the market for air quality
monitoring with different business models (Alfano et al., 2020). There are companies which produce and / or sell medium-
cost sensors (MCS) with a cost per compound in the order of 100 and 1000 Euros and LCS in the order of 10 and 100 Euros
for all key air pollutants (Concas et al., 2021). Furthermore, manufacturers and integrators often provide installation of LCS
and MCS for networks and on mobile monitoring platforms. The operation of such networked and mobile platform
measurements is also often supported by the companies which install the sensors. However, the monitoring of air pollutant
limit value exceedances is still a task of governmental agencies which are responsible for air quality.


These developments point to a new era in detecting the quality of air which we breathe (Munir et al., 2019, Schade et al., 2019,
Schäfer et al., 2021) where virtually everybody can measure air pollutants. Following this potential high number of sensors,





fine-granular assessment of air quality in urban areas is possible at lower costs. The data platforms of these LCS and MCS networks collect enormous amounts of data and new data products are supplied for users like personal exposure of air

pollutants, spatial distribution of air pollutants down to 1 m resolution, information about least polluted areas and forecast of air quality. Figure 7 shows these possibilities on the Internet of Everything with things, sensor data, open data platforms and citizen actions.

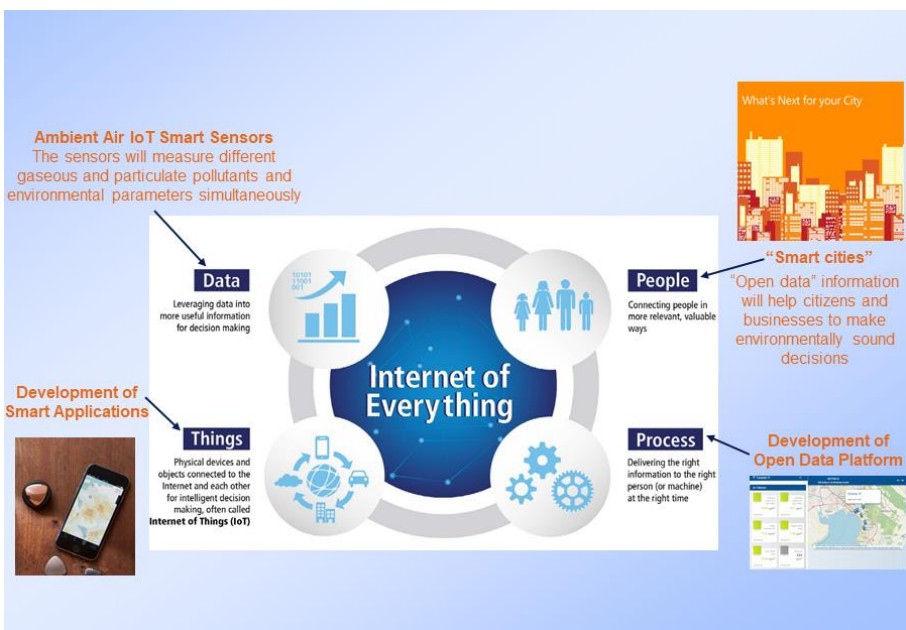

Figure 7: Exploitation of Internet of Everything technology with things, sensor data, open data platforms and actions of people. Algorithms from machine learning and big data, together with data from reference instruments as well as monitoring data owned by governmental agencies are often working on a central data server. Thus, an overall higher information density at an elevated quality level than the sum of the individual measurement components is possible. Also, a dynamic evaluation technique can be applied which is built upon mobile sensors onboard vehicles, for example, trams, buses, and taxis combined

with the existing monitoring infrastructure by intercomparison between any two devices which requires a corresponding high dynamic of their sensitivity. Pre- / post-calibrations are possible by using high-end instruments or adjustment in a reference atmosphere under prescribed laboratory and/or field conditions. Based on these achievements in the monitoring networks it is possible to identify emission hot spots and thus to assess spatially resolved, high-resolution emission inventories. Such emission inventories are a prerequisite for supporting high-resolution numerical simulations of air pollutant concentrations and

eventually the forecast of air quality.

Furthermore, because of the small size and low weight, sensors can be installed on-board of unmanned aerial vehicles (UAVs) so that these platforms become complex air quality (Burgués and Marco, 2020) and meteorological instruments. This means



vertical profiling is possible with aerial atmospheric monitoring to understand the influence of air pollutant emissions upon air quality.


### 4.2.2 Quality of sensor measured and numerical simulation data

An increasing number of evaluations of MCS and LCS as well as of networks based on such sensors are being performed and conclusions are available from these studies such as Thompson (2016), Morawska et al. suppl., (2018), Karagulian et al., (2019). It is well-known that these sensors suffer from drift and aging (Brattich et al., 2020). The drift can vary even among

the same model sensors that come from the same factory. Furthermore, sensor data evaluation is necessary due to cross-sensitivities of sensors with other air pollutants in ambient air and the influences of different temperatures and humidity in ambient air upon the sensor response.

Activities for the standardization of a protocol for evaluation of MCS and LCS at an international level and for inter-comparison exercises are ongoing, where MCS and LCS are tested at the same sites and at the same time (e.g., Williams et al.,

2019). The European Committee for Standardization / Technical Committee (CEN/TC) 264 / Working Group (WG) 42 "Ambient air – Air quality sensors" works for a Technical Specification of LCS (CEN - Technical Bodies - CEN/TC 264/WG 42). Such guidelines and sensor certifications are required for data products as e.g., personal air pollution exposure, emission source identification and nowcasting of air quality as well as for applications as traffic management (Lewis et al., 2018, Morawska et al., 2018).

In the area of high-resolution modelling, the creation of a model data standard for obstacle resolving models (www.atmodat.de) has started (Voss et al., 2020) as already done for coupled models (CESM - CMIP6 (ucar.edu)).

### 4.2.3 Importance of crowdsourcing, big data analysis and data assimilation

Data from high-resolution measurement networks can provide the base for application of small-scale 3-D process-based CTMs

by means of assessment of emission inventory and model results. Additionally, it can support the operation of statistical, artificial intelligence, neural network, machine learning, and hybrid modelling methods (Bai et al., 2018, WMO, 2020, Baklanov and Zhang, 2020). Statistical methods are simple but require a large amount of historical data and are extremely sensitive to them. Artificial intelligence, neural network and machine learning methods can have better performance but can be unstable and depend on data quality. Hybrid or combined methods often provide better performance. Such methods can

also improve the CTM forecast by utilising added observation data. For example, Mallet et al. (2010) have applied machine learning methods for the ozone ensemble forecast, performing sequential aggregation based on ensemble simulations and past observations. Latest results of the integration of air quality sensor network data with numerical simulation and neural network modelling results by data assimilation methods are for the Balkan region (Barmpas et al., 2020), Grenoble city, France (Zanini et al, 2020), city of Leipzig, Germany (Heinold et al., 2020), and inner city of Paris, France (Otalora et al., 2020) and show

how modelling can be used to support and consolidate information from observation data products.



The trend to improve air quality forecasting systems leads to the development of new methods of utilizing modern observational data in models, including data assimilation and data fusion algorithms, machine learning methods and bias correction techniques (Baklanov and Zhang, 2020). Typically, as a first step data verification and validation of different data sources is performed, including data from LCS and MCS networks, permanent monitoring networks as well as UAV-based,

aircraft-based, and satellite-based measurements (in situ and remote sensing). Subsequently, emission information data assimilation methods are applied for integration with urban-scale CTM or neural network modelling or fluid dynamics modelling or combining these modelling to provide a flexible framework for air quality modelling (Barmpas et al., 2020). Such approaches that combine the use of observations with models can lead to improved new tools to deliver high quality information about air quality, spatial high-resolution forecasts of air quality for hours up to days and health protection to the public.

Further, literature already provides QA/QC methods for MCS and LCS based on big data analyses and machine learning as well as data analyses in the cloud (Foken, 2021). Evaluation methods for measurement and modelling results are selected and combined to show the application potential of datasets of the new sensors, networks, and air quality model simulations. The further development and application of assimilation and quality evaluation methods is ongoing with the aim that distributed data sources will form the basis for new data products, making possible new applications for citizens, local authorities, and

stakeholders.

*4.2.4 Applicability of sensor observations*

Crowdsourcing of sensor observations is applied to get information for personal air pollution exposure and for supporting decisions on personal health protection measures such as e.g., information about least polluted areas for outdoor activities. Using this data-based information citizens can recognise heavily polluted areas, which could be especially important for

sensitive groups.

The platforms for the combination of ground-based stationery and mobile sensors, the complementation with 3-D measurement data by in situ and remote sensing observations as well as model evaluation and assessment can support such applications. This trend of cost-effective air quality monitoring includes user-oriented data services and education about air pollution and climate change to best exploit the knowledge and information content of measured data. Local authorities already use such

data (e.g., English, 2020) for identifying emission hot spots, management of city infrastructure and road traffic management towards improving air quality.

MCS and LCS and their advantages in operation and data availability via citizen sciences can also support the understanding of indoor air quality. The investigations of indoor air pollution in conjunction with outdoor air pollution monitoring provide more realistic data of personal air pollution exposure and for assessing measures of health protection.

*4.2.5 Modelling for urban air quality to support observation data products*

Numerical modelling results are traditionally evaluated against data from air quality monitoring networks (see also section 5). At high resolution, this process requires the use of a sensor network specifically configured to meet the needs of the exercise. Conversely, modelling can also be used to support air quality mapping based on observational data. Indeed, while the use of





LCS for high-density observations can provide information on the variability of pollutant concentration on a fine spatial scale,

the spatial (and temporal) global coverage of the areas being monitored nevertheless can prove to be irregular and incomplete. Data-driven modelling over combined stationary- and mobile-generated pollution data requires the deployment of dedicated statistical methodologies. Although little research effort has been devoted to such developments so far, recent advances in machine learning and artificial intelligence have highlighted the exciting potential of several statistical analysis tools (data envelopment analysis, unsupervised neural learning algorithms, decision trees etc.) to predict air quality at the city scale from

data generated by mobile sensors which are supported by citizen involvement (Mihaita et al., 2019).

Another approach that appears very promising to meet the operational challenges associated with fine spatial mapping is to combine sensor data with mapped data from models. The technique used is geostatistical data fusion, an approach similar to data assimilation, and based on kriging interpolation. It produces a new map whose added value lies in obtaining the most probable field of concentration, at the time when the sensor observations were made, but also the combination of information

provided by the two data sources (Ahangar et al., 2019, Schneider et al., 2017). A study carried out on a medium density urban area in France showed that the bias found between the outputs of an urban model and the data from the local air quality network was reduced from 8% to 2.5% following fusion with the sensor data. However, the results of the fusion technique are characterized by a lower dispersion than the input data sets, which leads to a smoothing of the peaks and thus an underestimation of the maximum values. Finally, the performance of fusion is logically degraded by the uncertainty in the

sensor measurements and the low correlation between the two data sources due to biases in the LCS measurements (Gressent et al., 2020). This underlines the importance of accurate calibration of portable devices to achieve reliable air quality mapping on a fine scale.

### 4.3. Emerging challenges

*4.3.1 Use of lower cost sensors*

Providing citizens and stakeholders with innovative information from large networks of sensors can yield added value and is fast becoming one of the main emerging challenges in air quality management. Nevertheless, with the greater range of observational techniques available now, there is a need for the application of instrumentation consistency, involving operation of mobile sensors by citizen for routine inter-calibrations and approaches for sensor intercomparison in networks, using

correction algorithms for sensors which should be described in a common way. When sensors are installed onboard vehicles or UAV, detailed information about the sensor response time should be provided taking account of the compatibility with its movement speed and data gathering frequency.

There is also the need to strengthen the linkages between existing measurement datasets e.g., air pollution monitoring networks of governmental agencies operating at local and national levels incorporating reference data with certified QA/QC methods

need to be explored to exploit numerical algorithms especially from artificial intelligence or dynamic data assimilation, for example as part of sensor and network certifications and standardisation so that these measurement methodologies and the available enormous amount of data can be useful for air quality research and assessment, including legislative reporting.





In the case of lower cost sensors, guidelines and sensor certifications for LCS and MCS are prerequisites for their application. Because such documentation is not consistently available up to now, LCS and MCS data cannot be used for official assessment

of WHO or EU limit value exceedances. Furthermore, the level of acceptable data quality of LCS and MCS is difficult to ascertain and presently the LCS and MCS networks are difficult to integrate into or extend the air pollution monitoring networks of responsible authorities.

### 4.3.2 Multipollutant instruments

Depending on the monitoring task of air quality or personal exposure, sensors for detection of all air pollutants including ultrafine particles (UFP) and particle size distribution (PSD) but also greenhouse gases (GHG) are necessary. In the application case of sensors embedded at the surface of clothes or carried by individuals, extended miniaturization of LCS and MCS must measure the personal air pollution exposure. Relevant developments could also include personal measurements of bioaerosols (e.g., pollen and fungi). Such data are required to study the combined health effects of air pollutants, bioaerosols and

meteorological parameters. In this sense the speciation or chemical composition and physical characteristics of particles of all sizes are needed too.

### 4.3.3 Modelling for urban air quality to support observation data products

The small-scale forecast of air quality for different applicants and personal health protection must be improved by adaptation

of corresponding numerical simulations of air pollution, based on online input data, which requires readily accessible sources like traffic counting and household heating activities. Alternatively, inverse modelling approaches can help quantify the strengths of diffusive emission sources and identify hot spots. Running spatial and temporal high-resolved numerical simulations requires online evaluation data from the combination of different platforms and the application of data algorithms from the area of machine learning or artificial intelligence.

The assimilation of small-scale data from measurements and numerical simulation of air pollution should be used for reduction of the space-time gaps of measurement networks. This is needed because measurement networks cannot be as dense as the spatial grids of numerical simulations. This implies e.g. further development of integration of observations by different platforms and methods as well as the assessment of numerical simulation results together with the application of crowdsourcing. Big data analyses and data assimilation methods can provide new areas of modelling applications in the field

of improvement of air quality, determination of air pollution emissions and emission inventories as well as development of personal health protection measures. Finally, it is necessary that this data eventually become suitable for monitoring and assessment of air quality in agreement with national and international guidelines.

Measurements and numerical simulation of coupled outdoor and indoor air quality must be supported for obtaining more realistic personal air pollution exposure information, given that most people are mainly exposed to indoor air which, in turn,

is strongly influenced by the quality of the outdoor air.



## 5. Air quality modelling from local to regional scales

### 5.1 Brief overview

Over the last years, it became obvious that our understanding of pollution and exposure processes at the urban scale could be improved by combining multi-scale models and creating new dedicated numerical approaches, and that the representation of scale interactions for dynamic phenomena, pollutant emission sources and pollutant aging would be a critical element in the realism of the simulation outputs. New developments have therefore aimed at restoring the spatial variability and heterogeneity of air pollution due to the turbulent transport of pollutants, whether in urbanized valleys, city centres or confined urban spaces such as canyon streets.

The motivation of these works is to address societal issues with a focus on street-level representations of pollutant concentration fields to support the assessment of individual exposure to pollution. In this context, it is now acknowledged that statistical and other data analysis techniques such as machine learning have an important role to play in identifying underlying patterns and trends as well as relationships between different parameters. At the same time, air quality monitoring has been progressing by improving ensemble techniques that allow for more in-depth model evaluation and provide a solid basis for consistent operational work on air quality. The following section reviews current challenges and highlights emerging areas of research covering the development, application and evaluation of air quality models.

### 5.2 Current status and challenges

#### 5.2.1 Innovative combinations of models

To meet the need to represent concentration gradients of primary pollutants in large agglomerations, the use of urban-scale dispersion models has increased since the 2010s (Singh et al., 2014, Soulhac et al., 2012). These models indeed allowed the resolution of dispersion effects in a complex emitting and built environment, whereas chemistry-transport models (CTMs) cannot provide an explicit representation of near-source characteristics and meet computational time issues as the resolution increases. However, both the lack of connection between local emission effects and the regional transport of pollutants and the absence of a relevant representation of atmospheric reactivity limit the scope of this type of model. Therefore, interest is progressively turned to the nesting of CTMs and urban models, that allows to exploit the advantages of both approaches. Over the last decade, approaches either coupling or nesting Eulerian models with Gaussian source dispersion models (Hood et al., 2018, Hamer et al., 2020), microscale CFD models (Tsegas et al., 2015), obstacle resolving Lagrangian particle models (Veratti et al., 2020) and/or street models (Jensen, et al. 2017, Kim et al., 2018; Khan et al., 2019b) have thus been developed with the aim of producing comprehensive cross-scale simulations of air quality in the city. An organization chart for such combined models is illustrated in Figure 8.





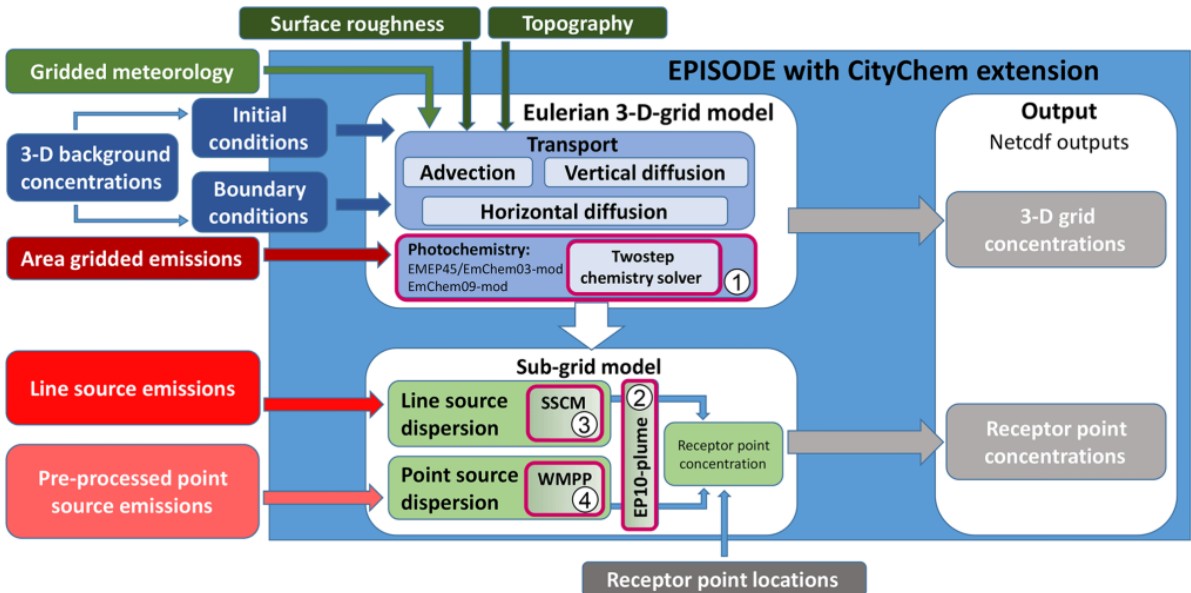

Figure 8. Schematic diagram of the EPISODE model with the CityChem extension (EPISODE–CityChem model), from Karl et al. (2019b).


The interest of the "CTM-Urban dispersion model" approaches called Plume-in-Grid or Street-in-Grid lies in the fact that they allow in a single time step to simulate urban background and to solve at low cost the dispersion of near-field emissions, for more resolved and realistic pollutant concentration fields. Compared to an urban model alone, those systems improve $NO_2$ scores in areas upwind of urban sources, as well as the average concentration levels of compounds that have a strong long-

range transport component such as $PM_{2.5}$, $PM_{10}$ and ozone (Hood et al., 2018). Implemented at the scale of an agglomeration or a region, this approach demonstrated its ability to represent the diversity of urban micro-environments (e.g. proximity to road traffic versus urban background, effect of building density and street configuration), that was until now poorly considered by the Eulerian approach alone. The representation of road traffic and its influence on urban air quality has been the main focus of these studies. Reaching a resolution from a few meters to a few tens of meters, the simulation outputs indeed accurately

reproduce the gradients observed along road axes (see Figure 9) and show greater comparability with urban-scale measurement data than CTMs alone (especially for $NO_2$). Particularly improved performances have been observed under stable winter conditions and for some studies, the deviation from measurements is within the 15% maximum uncertainty allowed by the EU directive for continuous measurements (Hamer et al., 2020). Mostly, the results show a better representation of the amplitude of the local signal than an improvement of the correlation with the observed concentrations, and it is concluded that these

multi-scale approaches are a significant advance to predict local peaks and episodes. These skills set them apart as essential tools for providing high-resolution air quality data for street-level exposure purposes (Singh et al., 2020b). Statistical evaluations of the model outputs based on the EU DELTA Tool have been carried out as part of several studies: they show





that the models comply well with the quality objectives of the FAIRMODE approach (https://fairmode.jrc.ec.europa.eu/QAQC.html). In the end, although the performances of the models remain dependent on the relative importance of local emissions, as well as transport and chemical processes at each computation grid point, most of the residual biases could be attributed to a lack of realism in the emissions. This includes the presence of poorly characterized local sources (works on the street, road particulate resuspension processes), but also insufficient temporal refinement of road traffic profiles. In this respect, it should be emphasized that the improvement of particulate representation in the model, as well as the restitution of near-field chemical equilibria, are also expected as major evolution pathways for the models.



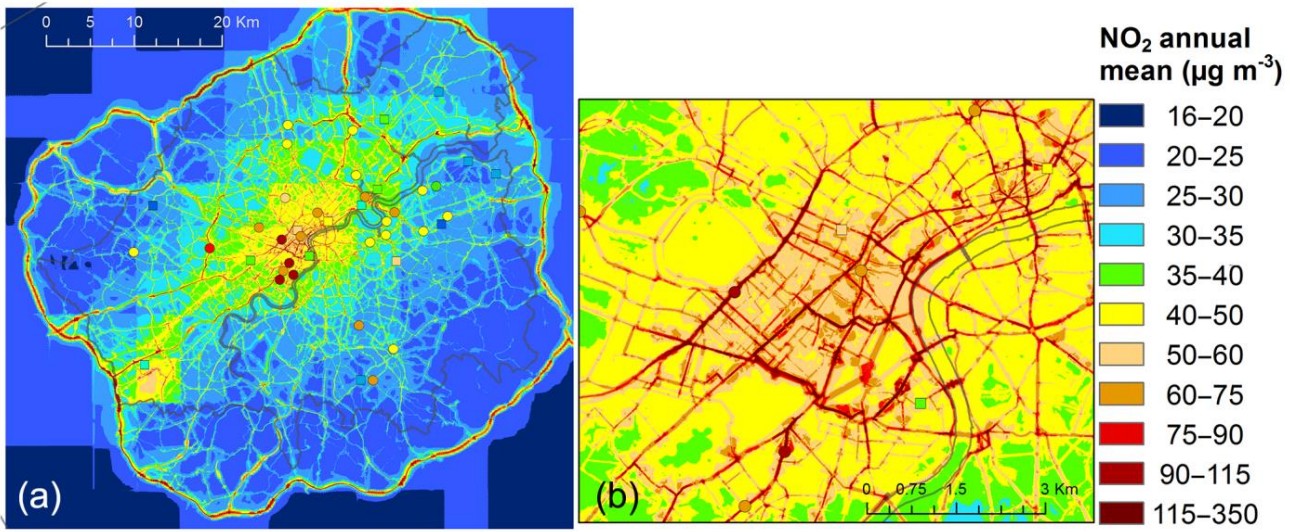

Figures 9. a-b. NO₂ annual average concentrations from the coupled ADMS-Urban / EMEP4UK model for (a) the whole of Greater London and (b) an area of central London. Monitoring data is overlaid as colored symbols (Hood et al., 2018)


The study of the impact of shipping activities on urban air quality has also benefited from these multi-scale modelling approaches. Indeed, while conventional CTM approaches simulating the effect of shipping emissions in coastal areas of the North and Baltic Seas agreed on the average contribution of shipping to air pollution (around 15-30% of elevated concentrations of $SO_2$, $NO_2$, ozone and $PM_{2.5}$, see (Aulinger et al., 2016; Jonson et al., 2015; Karl et al., 2019a; Geels et al. 2021; Moussiopoulos et al., 2019), the use of urban and plume dispersion models made it possible to refine this diagnosis and assess near-field effects. As for road traffic, the influence of ship emissions on air quality induces pollution gradients in the city. Karl et al. (2020) thus found out that, in residential areas up to 3600m from a major harbour, the ultrafine particle concentrations were increased by a factor of two or more compared with the urban background.



*5.2.2. Improved turbulence and dynamics for higher resolution assessment of urban air quality*

In parallel, the need for higher resolution assessment of urban air quality poses new demands on flow and dispersion modelling. As an additional difficulty besides complex geometry induced phenomena, we are reaching a spatial resolution of meters and a temporal resolution of seconds, thus entering the space/time scales of atmospheric turbulence. Therefore, the exposure related parameters cannot be described only deterministically without considering their stochastic component. A recent step forward

in this direction is the increased use of Large Eddy Simulations (LES) methodology dealing directly with the stochastic behaviour of flow and concentration parameters (Wolf et al., 2020).

Advanced computational fluid dynamics (CFD), including Reynolds-averaged Navier–Stokes equations (RANS) models that provide concentration standard deviation also have appeared in literature for some time (Andronopoulos et al., 2019). More precisely, the implementation of LES class models solving the most energetic part of turbulence explicitly as well as 3D

primitive hydro-thermodynamical equations and the structural details of the complex urban surface has been carried out at the scale of agglomerations, in meteorological conditions corresponding to typical stratified winter pollution situations, and fed with emission data from the city authorities (residential combustion as well as maritime and road traffic in particular). More specifically, advanced CFD models such as LES, have shown to better characterize the very fine-scale variability of primary urban pollution, for example regarding the irregular spatial distribution of concentrations in proximity to road traffic at complex

built-up intersections which makes it possible to open a reflection on the representativeness of the levels measured and their regulatory use, and to define criteria for the optimization of measurement networks. LES local scale modelling has been used to refine urban air quality predictions either standalone (Esau et al., 2020) or embedded in an urban scale model (San José et al., 2020). Also, wider use of CFD has taken place to improve understanding of pollution distribution inside a built environment, especially for critical infrastructure protection (Karakitsios et al., 2020).

Microscale models are particularly powerful to resolve the turbulent flow and pollutant dispersion around urban obstacles to reconstruct pollutant concentrations variability within the urban canopy. Recent micro-scale model simulations also showed the importance of barrier effects for emissions from large ships. It was thus shown that turbulence at the stern of the ship may cause a significant decrease in exhaust pollutants, leading to higher concentrations near the ground and, most likely, higher exposure of the nearby urban population (Badeke et al., 2020). The application of LES (Esau, 2020; Wolf-Grosse et al., 2017;

Resler et al., 2020; Werhahn et al., 2020; Hellsten et al., 2020, Khan et al., 2020) and CFD (San José et al., 2020; Gao et al., 2018; Flageul et al., 2020; Koutsourakis et al., 2020; Nuterman et al., 2011; Buccolieri et al., 2020, Kurppa et al., 2018; Kurppa et al., 2019; Karttunen et al., 2020; Kurppa et al., 2020) models for air quality assessment in urban environments is becoming a frequent approach. Many papers implementing the PALM LES model (Maronga et al., 2015) have been presented at the 12[th] International Conference on Air Quality – Science and Application. Yet, their application is still limited by difficulties dealing

with urban scale atmospheric chemistry and by the relevant computational resources required - as the use of advanced models such as LES requires increased computational capabilities. On the other hand, the heavy computational burden of urban LES computations can be reduced by approximately 80 % or even more by employing the two-way coupled LES-LES nesting technique, recently developed within the LES model PALM (Hellsten et al., 2021). Precomputation of LES in operational





modelling can be an acceptable solution especially combined with big data compression methodologies (Sakai et al., 2013).
Another possibility is to focus on limited urban areas with special interest (e,g, street canyons and 'hot spots'); however, one should in this case take into account the effect on turbulent transport from the surrounding larger-scale turbulent phenomena. In the problem of urban air quality, an assisted approach in the selection/classification process is the use of clustering (Chatzimichailidis et al., 2020) and artificial intelligence/machine learning technologies (Gariazzo et al., 2020).

### 5.2.3 Use of advanced numerical approaches and statistical models

At the same time, the complementary role of prognostic and diagnostic approaches has been explored. New methodologies based on artificial neural network models, machine learning or autoregressive models have been developed in order to achieve a more realistic representation of air quality in inhabited areas than achieved by CTMs (Kukkonen et al., 2003, Niska et al., 2005, Carbajal-Hernández et al., 2012, Wang et al., 2015b, Zhan et al., 2017, Just et al;, 2020, Alimissis et al., 2018). Likewise,
Pelliccioni and Tirabassi (2006) employed neural networks to improve the outputs of Gaussian and puff atmospheric dispersion models. Also, Mallet et al. (2009) applied machine learning methods for ozone ensemble forecast, and performed sequential aggregation based on ensemble simulations and past observations.

Kukkonen et al. (2003), through an extensive evaluation of the predictions of various types of neural network and other statistical models, concluded that such approaches can be accurate and easily usable tools of air quality assessment, but that
they have inherent limitations related to the need to train the model using appropriate site- and time-specific data. This dependence has prevented their use in the evaluation of air pollution abatement scenarios or for the evaluation of multidecadal time series of pollutant concentrations. The works of Li et al. (2017) confirmed that methods based on machine learning, and more specifically neural networks, can accurately predict the temporal variability of $PM_{2.5}$ concentrations in urban areas, but that the model performance may be improved using explanatory training variables. Prospective neural network modelling
works were also conducted in a canyon street by Goulier et al. (2019). They proposed a comparison of model outputs with measurements (based on Pearson correlation, rank correlation by Spearman, Modelling Quality Indicator's index from FAIRMODE, mainly), for a set of gaseous and particulate pollutants. They confirmed that the modelled data were able to reproduce with a very good accuracy the variability of the concentrations of some gaseous pollutants ($O_3$, $NO_2$), but that there was still a significant margin for improvement of the models, notably for particles. Again, an important part of the expected
progress lies in the choice of model predictors.

As for multi-scale modelling, the main research efforts associated with these numerical approaches are directed towards the downscaling of simulated pollutant concentration fields in urban areas, the improvement of CTM forecast using additional observation data, and a refined representation of individual exposure at the street scale (Berrocal et al., 2020, Elessa Etuman et al., 2020). Gariazzo et al. (2020) used a random forest model to enhance CTM results and produce improved population
exposure estimates at 200m resolution, in a multi-pollutant, multi-city and multi-year study conducted over Italy. In addition to reduced bias, the outputs presented much greater physical consistency in their temporal evolution, when compared to measurements.





Other applications, such as advancing knowledge about exposure in urban microenvironments, have also been made possible
by these approaches Thus, the use of Bayesian statistics has shown an ability to predict the concentration gradients of primary
pollutants in the immediate vicinity of an air quality monitoring station, by iterating between observations and the outputs of
a micro-scale simulation approach - including both a CFD and a Lagrangian dispersion model (Rodriguez et al., 2019).

*5.2.4 Implementation of activity-based data*

To take full advantage of the high-resolution simulation capability of these new modeling tools, and to achieve a more
comprehensive approach to the determinants of air quality in urban areas, modelers have relied on a new generation of activity-
based emissions data.

As for traffic, new methodologies relying on individual data collected through surveys, geocoded activities, improved emission
factors and measured traffic flows (Gioli et al., 2015, Sun et al., 2017) or involving traffic models simulating origin-destination
matrices for city dwellers on the road network (Fallah-Shorshani et al., 2017) have been developed to serve as input to the
urban dispersion models. Their implementation on a case study in Italy, with a horizontal resolution of 4m, showed that detailed
traffic emission estimates were very effective in reproducing observed NOx variability and trends (Veratti et al., 2020).

Residential wood combustion has also proven to act as a major source of harmful air pollutants in many cities in Europe, and
especially in Northern-Central and Northern European countries which have a strong tradition of wood combustion. Yet, until
the early 2010s, RWC inventories were still heavily burdened with uncertainties related to actual wood consumption, the
location of emitters, emission factors depending on heating equipment, and practices driving the temporality of emissions. To
represent RWC emissions more accurately in urban air quality models, new emission estimation methods based on
environmental and activity variables that drive pollutant emissions have been developed. They include for example outdoor
temperature, housing characteristics and equipment, available heating technologies and associated emission factors, or
temporal activity profiles from official wood consumption statistics (Grythe et. al., 2019, Kukkonen et al., 2020b). Kukkonen
et al. notably showed with this approach that the annual average contribution of RWC to $PM_{2.5}$ levels could be as high as15%
to 22% in Helsinki, Copenhagen and Umeå and up to 60% in Oslo. Overall, although the results show a better horizontal and
vertical spatial distribution of emissions compared to non-specific inventories, improvements are expected, especially on the
use of meteorological parameters and regarding emission factors for specific devices.

Finally, for emissions associated with maritime activity in port areas, the inventories developed specifically for high-resolution
modelling approaches include information on the fleet, the ship rotations in the harbour and the emission heights. The
implementation of the EPISODE-CityChem model within a CTM showed that in Baltic Sea harbour cities such as Rostock
(Germany), Riga (Latvia) and Gdansk-Gdynia (Poland) shipping activity could have contributed to 50 to 80% of $NO_2$
concentrations within the port area (Ramacher et al., 2019). As for the other sources, improvements are expected. They concern
for instance the energy consumption of the different ships and the propulsion power of the auxiliary systems of the ships during
their stay in port.





Because they allow detailed mapping of air quality in urban areas, and realistically represent emitting activities, those
approaches allow tackling issues such as chronic exposure and source-concentration relationships, but also provide elements
for increased policy and technical measures, as discussed below: regulation, information campaigns and economic steering.

*5.2.5 Contribution of modelling to policy making and urban management strategies*

Applying air quality and emission models allows for projections of future developments in air quality that can shed light on

the different effects of alternative policy options, e.g., new regulations or effects of changes in the emissions from certain
emission sectors. As an example (Figure 10), the OSCAR model was run over London to quantify the contribution of sources
- such as traffic - to the urban $PM_{2.5}$ concentration gradients.


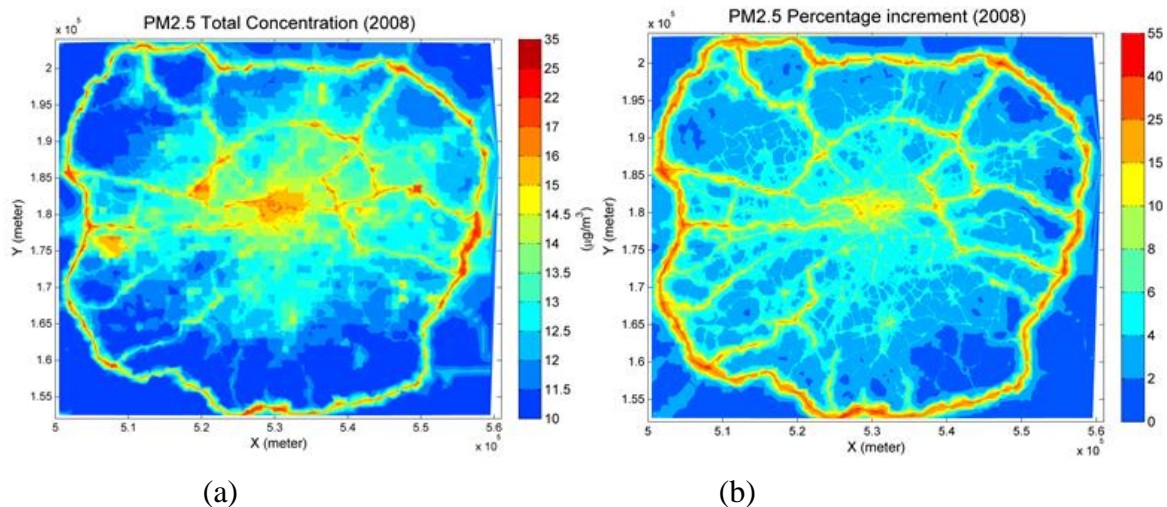

(a)                                    (b)

Figures 10. a-b. (a) Predicted spatial distributions of the annual mean $PM_{2.5}$ concentrations in µg/m³, and (b) Urban traffic
contributions to the total $PM_{2.5}$ concentrations, in %, for London for year 2008 (Singh et al., 2014).

Air quality modelling is expected to gain relevance following the review of air quality legislation announced as part of the
European Green Deal (EC, 2019), whereby the European Commission will also propose to strengthen provisions on
monitoring, modelling and air quality plans to help local authorities achieve cleaner air. The construction of these future air
quality modelling scenarios can be demanding, in particular when the goal is to be realistic and consistent with technological
potentials as well as economic and societal developments (in particular, reductions in the use of fossil fuels driven by climate

policies).

Another field of action recently explored is that of technology-based and management-based traffic control strategies, and in
particular the implementation of Low Emission Zones (LEZs) in urban areas (e.g. in Portugal (Dias et al., 2016), France (Host





et al., 2020), and India (Sonawane et al., 2012)). The quantification of the expected gains in terms of pollutant concentrations

in ambient air, but also of economic benefits and reduction in the occurrence of chronic respiratory diseases or vascular accidents provides concrete and robust elements for political and citizen debate and helps to move towards greater acceptability of the measures. In this framework, the degree of realism of the simulated scenarios, the spatial refinement of the approaches used, but also the capacity to evaluate them at the sub-urban scale (street, individual) can become determining elements of their scientific relevance and their legitimacy in the policy debate. Therefore, an increasing number of studies favour the use

of multi-scale models with the introduction of puff or gaussian dispersion models, as well as canyon-street models, with CTMs. When modelled scenarios serve as a basis for political decisions, it is highly valuable to include relevant authorities and decision makers from the beginning in the scenario design. This can be done in common workshops with relevant stakeholders where questions about technological trends and possibilities for emission reduction are discussed.

The analysis of simulation data for the estimation of health impacts can be ensured by integrated approaches - such as the

EPA's Environmental Benefits Mapping and Analysis Program (BenMAP) - or more simply by algorithms derived from epidemiology such as population-attributable fractions, which are standard methodology used to assess the contribution of a risk factor to disease. In terms of emissions, depending on the focus of the study, survey data on residential practices or activity-based road traffic models (as well as marine traffic models where appropriate) are increasingly used. Supplementary traffic algorithms can sometimes more accurately represent the effects of congestion on roadway emissions. Finally, for more realism,

the scenarios considered can be either derived from the relevant air quality plans implemented at the scale of agglomerations or projections on vehicle fleet evolution (Andre et al., 2020). Some of the models also include the feedback effects of changes in practice, such as the estimate of emission increase due to the energy demand for electric vehicle charging (Soret et al., 2014). Very small-scale modelling has also been used in other fields such as support in evaluating the effect of roadside structures on near-road air quality. Several studies - mainly based on CFD models, including LES approaches have thus focused on the

performance of air pollution dispersion by green infrastructures in open areas and street canyons, even characterizing the capacity of parked vehicles to reduce pedestrian exposure to pollutants (see review article in Abhijith et al., 2017). Also, the link between the morphology of urban buildings, the dispersion of emissions and air quality is often apprehended through CFD models (Hassan et al., 2020). At an even more operational level, LUR models (based on the spatial analysis of air quality data) have been coupled to high-resolution CTM runs to allow a precise identification of land use classes more exposed to $PM_{10}$,

$SO_2$ and $NO_2$. The results provided a methodological framework that could be used by authorities to assess the impact of specific plans on the exposed population, and to include air quality in urban development policies (Ajtai et al., 2020).

Examples also exist is in the area of shipping emissions, where several EU funded projects either involved stakeholders such as IMO and HELCOM from the beginning (e.g., Clean North Sea Shipping, ENVISUM, CSHIPP, EMERGE) or made use of their knowledge in dedicated expert elicitation workshops (e.g., SHEBA). Future scenarios for shipping, some of them

developed in these projects, were presented for the North and Baltic Seas (Johansson et al., 2013, Matthias et al, 2016, Karl et al., 2019a, Jonson et al. 2015), for Chinese waters (Zhao et al., 2020b) as well as globally (Sofiev et al., 2018, Geels et al.,





2020). However, the process of scenario generation in cooperation with authorities and other stakeholders is rarely described in scientific literature, or fully detailed in publications that address various policy options.

*5.2.6 Ensemble modelling for air quality research applications*

In parallel, statistical developments also serve the evolution of ensemble models. During the last decade, ensemble-building methodologies have been questioned and improved in several international collaborations and the inclusion of new observational data has allowed a better assessment of the relevance of these approaches. Ensemble forecasting can be implemented using multiple models or one model but with different inputs (e.g., varying meteorological input forcings, emission scenarios, chemical initial conditions) or different process parameters (e.g., varying chemical reaction rates) or
different model configurations (e.g., varying grid spacings) or different models (Hu et al., 2017; Galmarini et al., 2012). A comprehensive study on ensemble modelling of surface $O_3$ was done as part of the Air Quality Model Evaluation International Initiative (AQMEII), including 11 CTMs operated by European and North American modelling groups (Solazzo et al., 2012). One of the main conclusions was that even if the multi-model ensemble based on all models performed better than the individual models, a selection of both top- and low-ranking models can lead to an even better ensemble. It was also shown that
outliers are needed in order to enhance the performance of the ensemble.

Within the CAMS regional forecasting system for Europe, multi-model ensemble modelling is a part of daily operational production (https://www.regional.atmosphere.copernicus.eu/) for several air quality components. Statistical analyses have shown that an ensemble based on the median of the individual model gives a robust and efficient setup, also in case of outliers and missing data (Marécal et al., 2015). By combining global and regional-scale models, Galmarini et al (2018) have taken
this kind of ensemble modelling a step further, by setting up a hybrid ensemble to explore the full potential benefit of the diversity between models covering different scales. The analysis indeed showed that the multi-scale ensemble leads to a higher performance than the single-scale (e.g. regional-scale) ensemble, highlighting the complementary contribution of the two types of models.

**5.3 Emerging challenges**

*5.3.1. On multiscale interaction and subgrid modelling*

The advances in computational capacity, the progress on big data management and the recent developments on low-cost sensors technology, together with the significant developments in closing the gaps of knowledge when dealing with finer spatial and temporal scales (up to the order of meters and seconds, respectively) gives the opportunity for further achievements
in terms of innovation and outcome reliability in urban to local scale flow and air quality assessment. In such applications, very high spatial resolution modelling outputs are required together with dynamic and geocoded demographic data to conduct health monitoring on the impacts of air pollutants. However, new sub-grid / local approaches such as LES, advanced CFD-RANS, machine learning statistical tools, as well as interfaces among different modelling scales (regional, urban, local/sub-grid) require further R&D work, especially when in the interfacing of models using different parameterisations or
computational approaches.



Of specific interest here is the case of model nesting in regimes where it has not been extensively applied in the past, as is the case of implementation and validation of multiply nested LES (see e.g., Hellsten et al. 2021), as well as coupling of urban-scale deterministic models with local probabilistic models. In both areas, complications arise due to the nature of different parameterisations and the way boundary conditions are traditionally treated in LES models, highlighting the need for further validation and tools for the numerical evaluation of coupling implementations. Further areas of development include the better articulation between CTMs and subgrid models, towards solving overlay problems like emission double counting and mass conservation across interpolated interfaces, both critical points for their successful application as assessment tools.

### 5.3.2. On chemistry and aerosol modelling

One important aspect is the fact that local scale models often include simple approaches to tropospheric chemistry. Although such an approach can be justified from the fact that computation domain time scales are usually well below lifetime scales of priority pollutants, it also poses limitations that need to be addressed. For example, the lack of full representation of NOx/VOC chemistry, or not considering a delay in establishing the photostationary NO-NO2-O3 equilibrium, can introduce a significant bias in the restitution of concentration gradients at very fine scales. Particle size resolved schemes, including for example the discrimination of particle removal phenomena, are also expected to be important developments for these local models. How do simplified chemistry and physics impact on treating traffic emissions in cities, what is their role in the restitution of particle growth, SOA formation and ozone chemistry? These issues require special attention. They are also relevant to the treatment of other urban sources generating strong concentration gradients, such as shipping. Thus, the impact of the representation of VOC behaviour on particle formation and aging, or the effect of NO2 removal, both in the early phases of ship plume dispersion, should also be investigated.

More globally, there remain issues on the representation of reactivity in multi-scale modelling approaches and air quality forecasting. On the one hand, although some studies have shown that high-resolution models are good at predicting the occurrence (or non-occurrence) of local pollution events, it has been observed that they do not always capture the full range of pollutant concentrations and, especially, the amplitude of the strongest concentration peaks. On the other hand, there remains a very strong interaction between locally emitted pollutants and those resulting from LRT to the city. This may be determinant for the operational forecasting of air quality at the urban scale. Thus, the representation, on a fine scale, of the fundamental processes of reactivity is one next challenging issue of multi-scale modelling. For local scale modelling it is indeed important to make sure that at least we include chemical transformation with time scales significantly smaller than the time ranges imposed by the considered computational domain.

### 5.3.3 On fine scale model input and emission data

As we move to finer scales and more advanced modelling, the input data – whether meteorological, descriptive of the urban environment, or related to the sources of pollutants also require additional knowledge of their time and space variation, even including sufficiently detailed statistical behaviour. The refinement of meteorological and chemical input fields for statistical


approaches is an important challenge. Indeed, the application of LES or statistical models in a fine domain embedded into a
larger domain where ensemble-average modelling data are available and needed, raises the question of how to generate fine-
scale or statistical input data that are both mathematically consistent and physically correct. It was highlighted that the role of
statistical models based on machine learning is increasing, especially for urban AQ applications. This is due to growing
computer and IT-networking possibilities, but also to new types of numerous observations, e.g. crowdsourcing, low-cost
sensors or citizen science approaches. The ability of machine learning to capture these new data sources and identify new
applications in fine-scale air quality and personal exposure is therefore a great challenge for the coming years.

As far as emissions are concerned, the gain in realism has become a prerequisite to produce decision-support scenarios, and
requires a strong grounding in reality - i.e., emissions must be based on a census of the activities and on the specificities of the
emitters (e.g., car engines, heating equipment and rotation of boats in the port) which requires increasingly complex phases of
model implementation over a territory, and the intervention of a multiplicity of actors for data supply. In this context, tabulated
emission inventories - even those based on actual activity data - have limited scope for use in future air quality and exposure
scenarios. To be realistic, the scenarios must be able to reproduce the variation in emitting activity in relation to changes in
transport supply, urban planning, energy costs and individual or collective energy consumption practices. Therefore, a
significant part of the work is now focused on developing air quality modelling platforms integrating emission models centred
on the individual (see Figure 11, Elessa Etuman and Coll, 2018).

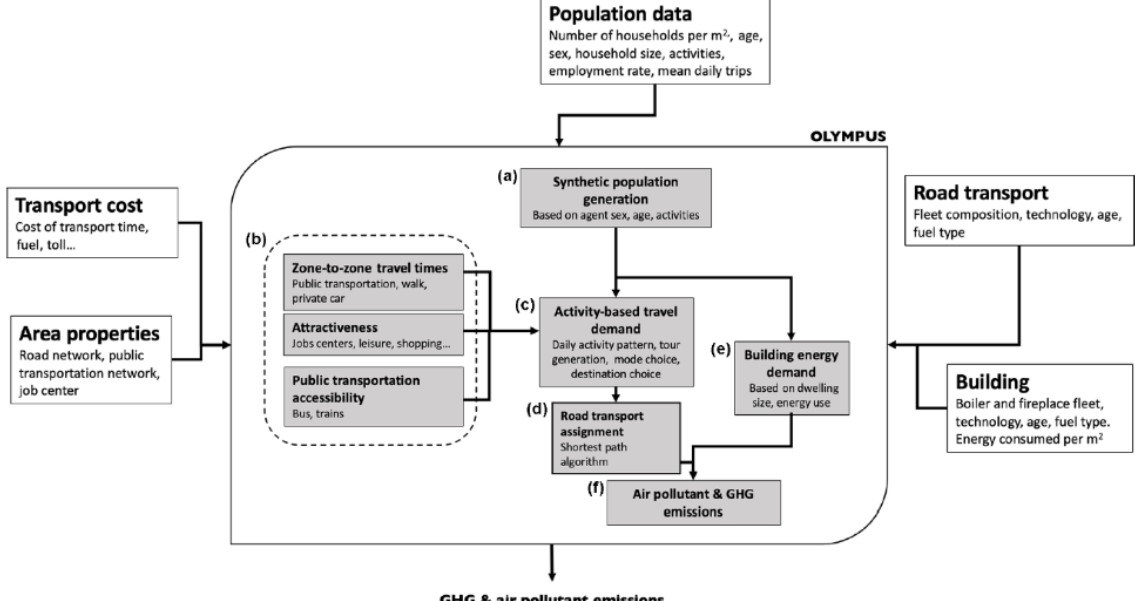

Figure 11. Schematic representation of OLYMPUS emission operating system (Elessa Etuman and Coll., 2018)



There, the main challenges are related to the representation of individual mobility for both commuting and private activities
as well as domestic heating and more broadly energy consumption practices on one side, and the consideration of traffic
parameters such as urban freight, the distribution of traffic and its speciation, driving patterns or the effects of road congestion
on the other side (Lejri et al., 2018, Coulombel et al., 2019).

Another emerging issue is also how to cope with short-time hazardous emissions in urban areas. Such emissions can be related
to accidents or deliberate releases that are of increased concern today. An important characteristic of associated exposures is
their inherent stochastic behaviour (Bartzis et al., 2020). Novel modelling approaches are needed to properly assess the impact
and support relevant mitigation measures.

### 5.3.4 On model evaluation

To act on these numerous and expected developments, and use their results for operational decision support, multi-scale models
need validation. An often-overseen basic prerequisite here is the availability and representativeness of validation data,
particularly at smaller scales. The model's performance indeed needs to be explored in more spatial detail and in all covered
spatial scales, preferably as part of multi-scale urban to rural intercomparison projects in order to be able to provide finer
assessment on air quality and exposure. Such efforts can be supported by networks of inexpensive sensors as well as smart
tags (Sevilla et al., 2018) and other sources of distributed information acting complementary to traditional local monitoring
and flow profiling technologies. To obtain methodology and data refinement as well as outcome reliability, more experience
through additional case studies is also needed. Finally, consideration should be given to specific model performance evaluation
criteria for various regulatory purposes, including prospective mode operation, i.e. the ability of a model to accurately predict
the air quality response to changes in emissions. To this end, evaluations can draw on the very large methodological work that
has been carried out since 2007 by the Forum for AIR quality MODelling in Europe (FAIRMODE) for the assessment of
CTMs (Monteiro et al., 2018). The objective was to develop and support the harmonized use of models for regulatory
applications, based on $PM_{10}$, $NO_2$ and $O_3$ assessments. The main strength of this approach was to produce an in-depth analysis
of the performance of different model applications, combining innovative and traditional indicators (Modelling Quality Index
and Modelling Quality Objectives) and considering measurement uncertainty. Although FAIRMODE was successful in
promoting a harmonized reporting process, there remain major ways of improvement that can be critical for its regulatory
acknowledgement - in particular regarding inconsistencies between indicators of different time horizons - and a methodology
dedicated to data assimilation assessments.

## 6. Interactions between air quality, meteorology and climate
### 6.1 Brief overview

There is a need to increase prediction capabilities for weather, air quality and climate. The new trend in developing integrated
atmospheric dynamics and composition models is based on the seamless Earth System Modelling (ESM) approach (WWRP,
2015) to evolve from separate model components to seamless meteorology-composition-environment modelling systems





(WMO, 2016). One driver for improvement is the fact that information from predictions is needed at higher spatial resolutions, longer lead times. In addition, we have to consider two-way feedbacks between meteorological and chemical processes on the

one hand and aerosol-meteorology feedback on the other hand; where both are needed to meet societal needs. Continued improvements in prediction will require advances in observing systems, models and assimilation systems. There is also growing awareness of the benefits of closely integrating atmospheric composition, weather and climate predictions, because of the important role that aerosols (and atmospheric composition in general) play in these systems.

While this section also considers challenges related to air quality modelling, it differs in emphasis to Section 5, by examining

interactions that operate on multiple scales and includes multiple processes that affect air quality, especially for cities.

## 6.2 Current status and challenges

### 6.2.1 Interactions and coupled chemistry-meteorology modelling (CCMM)

Meteorology is one of the main uncertainties of air quality modelling and prediction. Many studies have investigated the role

of meteorology on air quality in the past (e.g. Fisher et al., 2001, 2005 and 2006, Kukkonen et al., 2005a,b) and even more recently (e.g. McNider and Pour-Biazar, 2020; Rao et al., 2020; Gilliam et al., 2015; Parra, 2020). The relationship between meteorology and air pollution cannot be interpreted as a one-way input process due to the complex two-way interaction between the atmospheric circulation and physical and chemical processes involving trace substances in both gas and aerosol form. The improvement of atmospheric phenomena prediction capability is, therefore, tied to progresses in both fields and to

their coupling.

The advances made by mesoscale planetary boundary layer meteorology during the last decades have been recently reviewed by Kristovich et al. (2019). During the last decade significant advances have been made even in the capabilities to predict air quality and to model the many feedbacks between air quality, meteorology and climate, including radiative and microphysical responses (WMO, 2016, 2020; Pfister et al., 2020). Due to advances in air quality models themselves and the availability of

more computing resources, air quality models can be run at high spatial resolution and can be tightly (on-line) or weakly linked to meteorological models (thorough couplers). This is a pre-requisite to improve prediction skills further, while air quality models themselves will be improved as our knowledge of key processes continues to advance.

Online-coupled meteorology atmospheric chemistry models have greatly evolved during the last decade (Flemming et al., 2009; Zhang et al., 2012a,b; Pleim et al., 2014; WWRP, 2015; Baklanov et al., 2014; Mathur et al., 2017; Bai et al., 2018).

Although mainly developed by the air quality modelling community, these integrated models are also of interest for numerical weather prediction and climate modelling as they can consider both the effects of meteorology on air quality, and the potentially important effects of atmospheric composition on weather (WMO, 2016). Migration from offline to online integrated modelling and seamless environmental prediction systems are recommended for consistent treatment of processes and allowance of two-way interactions of physical and chemical components, particularly for AQ and numerical weather prediction (NWP)

communities (WWRP, 2015; Baklanov et al., 2018a).





It has been demonstrated that prediction skills can be improved through running an ensemble of models. Intercomparison studies such as MICS and AQMEII (Tan et al., 2020; Galmarini et al., 2017; Zhang et al., 2016) serve as important functions of demonstrating the effectiveness of ensemble predictions and helping to improve the individual models. Predictions can also

be improved through the assimilation of atmospheric composition data. Weather prediction has relied on data assimilation for many decades. In comparison, assimilation in air quality prediction is much more recent, but important advances have been made in data assimilation methods for atmospheric composition (Carmichael et al., 2008; Bocquet et al., 2015; Benedetti et al., 2018). Community available assimilation systems for ensemble and variational methods make it easier to utilize assimilation (Delle Monache et al., 2008; Mallet, 2010). Furthermore, the amount of atmospheric composition data available

for assimilation is increasing, with expanding monitoring networks and the growing capabilities to observe aerosol and atmospheric composition from geostationary satellites (e.g., Kim et al., 2020). The operational systems such as CAMS (Copernicus Atmospheric Monitoring Service) have advanced current capabilities for air quality prediction (Marécal et al., 2015; Barré et al., 2020).

Currently, NWP centres around the world are moving towards explicitly incorporating aerosols into their operational forecast

models. Demonstration projects are also showing a positive impact on seasonal to sub seasonal forecast by including aerosols in their models (Benedetti and Vitart, 2018). Even the usual subdivision between global scale NWP models and limited area models employed to resolve regional to local scales is going to be revised. Many groups are building new earth system models and taking advantage of global refined grid capabilities that facilitate multiscale simulations in a single model run, as in the case of *Model* for Prediction Across Scales *(*MPAS*)* (Skamarock et al., 2018; Michaelis et al., 2019) and MUSICA (Pfister et

al, 2020) approaches.

*6.2.2 Aerosol-meteorology feedbacks for predicting and forecasting air quality for city scales*

Multiscale CTMs are increasingly used for research and air quality assessment but less for urban air quality. Recently, there are examples of coupled urban and regional models which allow the prediction and assessment of local, urban and regional air

quality affecting cities (Baklanov et al., 2009; Kukkonen et al., 2012, Sokhi et al., 2018; Kukkonen et al., 2018, Khan et al., 2019b). In particular, a downscaling modelling chain for prediction of weather and atmospheric composition on the regional, urban and street scales is described and evaluated against observations by Nuterman et al., (2021). Kukkonen et al. (2018) described a modelling chain from global to regional (European and Northern European domains) and urban scales, and a multidecadal hindcast application of this modelling chain.

There are still uncertainties in prediction of PM components such as secondary organic aerosols (SOA), especially during stable atmospheric conditions in urban areas which can cause severe air pollution conditions (Beekmann et al., 2015). Moreover, aerosol feedback and interaction with urban heat island (UHI) circulation is a source of uncertainty in CTM predictions. Several studies (Folberth et al., 2015; Baklanov et al., 2016; Huszar et al., 2016) demonstrated that urban emissions of pollutants, especially aerosols, are leading to climate forcing, mostly at local and regional scale, through complex





interactions with air quality (Figure 12).  These are in addition to almost 70% of global CO2 emissions arising from urban

areas, and hence urban areas pose a considerable source of climate forcing species.

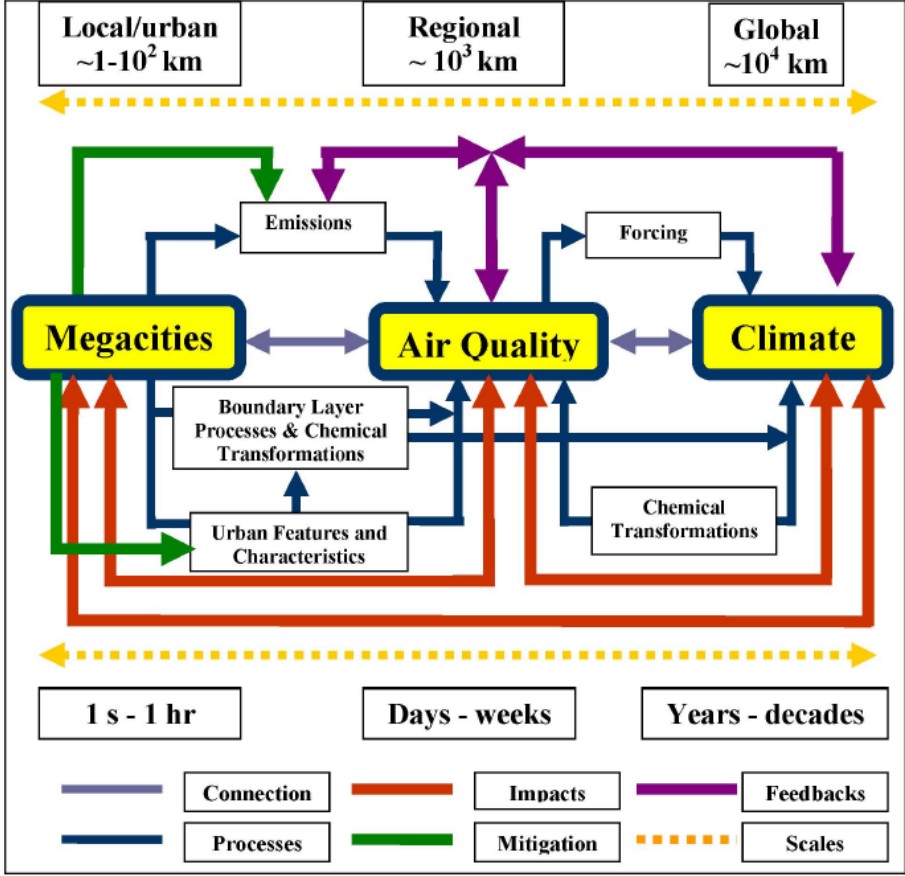

Figure 12. Main linkages between urban emissions, air quality and climate. (from: Baklanov et al., Adv. Sci. Res., 4, 115–120, 2010; www.adv-sci-res.net/4/115/2010/; doi:10.5194/asr-4-115-2010).


It is necessary to highlight that the effects of aerosols and other chemical species on meteorological parameters have many

different pathways (e.g., direct, indirect, semidirect effects) and must be prioritized in integrated modelling systems. Chemical

species influencing weather and atmospheric processes over urban areas include greenhouse gases (GHGs), which warm near-

surface air, and aerosols, such as sea salt, dust, and primary and secondary particles of anthropogenic and natural origin. Some

aerosol particle components (black carbon, iron, aluminium, polycyclic and nitrated aromatic compounds) warm the air by

absorbing solar and thermal-infrared radiation, while others (water, sulphate, nitrate, and most organic compounds) cool the

air by backscattering incident short-wave radiation to space. It has been demonstrated (Sokhi et al, 2018; Baklanov et al., 2011;

Huszar et al 2016) that the indirect effects of urban aerosols modulate dispersion by affecting atmospheric stability (the





difference in deposition fields is up to 7%), besides its effects on the urban boundary layer (UBL) thickness could be of the

same order of magnitude as the effects of the UHI (a few hundred meters for the nocturnal boundary layer).

*6.2.3 Urban scale interactions*

Meteorology is one of the main uncertainties in air quality assessment and forecast in urban areas where meteorological characteristics are very inhomogeneous (Hidalgo et al., 2008; Ching, 2013; Huszar et al., 2018, 2020). For these reasons,

models used at the urban level must achieve greater accuracy in the meteorological fields (wind speed, temperature, turbulence, humidity, cloud water, precipitation).

Due to different characteristics of the surface properties (e.g. heat storage, reflection properties), a heat island effect occurs in cities. Urban areas can therefore be up to several degrees Celsius warmer than the surrounding rural areas and experience lighter winds due to the increased drag of urban canopy. This heating impacts the local environment directly, as well as

affecting the regional air circulation with complex interactions that can induce pollutants recirculation, worsen stagnation episodes and influence ozone and secondary aerosol formation and transport.

Studies over the past decade (e.g. McCarthy et al. 2010; Cui and Shi, 2012; González-Aparicio et al., 2014; Fallmann et al., 2016; Molina, ) have shown that the effects of the built environment, such as the change in roughness and albedo, the anthropogenic heat flux, and the feedbacks between urban pollutants and radiation, can have significant impacts on the urban

air quality levels. A reliable urban scale forecast of air flows and meteorological fields is of primary importance for urban air quality and emergency management systems in the case of accidental toxic releases, fires, or even chemical, radioactive, or biological substance releases by terrorists.

Improvements (so called "urbanization") are required for meteorological and NWP models that are used as drivers for urban air quality (UAQ) models. The requirements for the urbanization of UAQ models must include a better resolution in the vertical

structure of the urban boundary layer and specific urban features description. One of the key important characteristics for UAQ modelling is the mixing height, which has a strong specificity and inhomogeneity over urban areas because of the internal boundary layers and blending heights from different urban roughness neighbourhoods (Sokhi et al., 2018, Scherer et al., 2019). Modern urban meteorology and UAQ models (e.g. WRF, COSMO, ENVIRO-HIRLAM) successfully implemented different complexity (a hierarchy of) urban parameterizations and reached suitable spatial resolutions (Baklanov et al., 2008; Salamanca

et al., 2011, 2018; Sharma et al., 2017; Huang et al., 2019; Mussetti et al., 2020; Trusilova et al., 2016; Wouters et al., 2016; Schubert and Grossman-Clarke, 2014) for an effective description of atmospheric flow in urban areas. The application of urban parameterizations implemented inside limited area meteorological models is becoming a common approach to drive urban air quality analysis, allowing the improved urban meteorology description in different climatic and environmental conditions (Ribeiro et al., 2021; Salamanca et al., 2018; Gariazzo et al., 2020; Pavlovic et al., 2020; Badia et al., 2020). However, activities

to improve the parameterizations (Gohil and Jin; 2019) and provide reliable estimation of the input urban features (Brousse et al., 2016) are continuing.



*6.2.4 Integrated weather, air quality and climate modelling*

Since cities are still growing, intensification of urban effects is expected, contributing to regional or global climate changes, including intensification of floods, heat waves and other extreme weather events, air quality issues caused by pollutants production and transport. This requires a more integrated assessment of environmental hazards affecting towns and cities.

The numerical models most suitable to address the description of mentioned phenomena within integrated operational urban weather, air quality and climate forecasting systems are the new generation limited-area models with coupled dynamic and

chemistry modules (so called Coupled Chemistry-Meteorology Models, CCMM). These models have benefited from rapid advances in computing resources, along with extensive basic science research (Martilli et al., 2015; WMO, 2016; Baklanov et al., 2011, 2018). Current state-of-the-art CCMMs encompass interactive chemical and physical processes, such as aerosols-clouds-radiation, coupled to a non-hydrostatic and fully compressible dynamic core that includes monotonic transport for scalars, allowing feedbacks between the chemical composition and physical properties of the atmosphere. These models are

incorporating the physical characteristics of the urban built environment. However, simulations using fine resolutions, large domains and detailed chemistry over long time durations for the aerosol and gas/aqueous phase are computationally demanding given the models' high degree of complexity. Therefore, CCMM weather and climate applications still make compromises between the spatial resolution, domain size, simulation length and degree of complexity for the chemical and aerosol mechanisms.

Over the past decade integrated approaches have benefited from coupled modelling of air quality and weather enabling a range of hazards to be assessed. Research applications have demonstrated the advantages of such integration, and the capability to assimilate aerosol information in forecast cycles to improve emission estimates (e.g. for biomass burning) impacting both weather and air quality predictions (Grell and Baklanov, 2011; Kukkonen et al., 2012, Klein et al., 2012, Benedetti et al., 2018).


## 6.3 Emerging challenges

*6.3.1. Earth systems modelling for air quality research*

Full integration of aerosols across the various applications requires advances in earth system modelling, with explicit coupling between the biosphere, oceans and the atmosphere, taking advantage of global refined grid capabilities that facilitate multiscale

simulations in a single ESM run. The earth system models offer many advantages but also make arise new challenges. Data assimilation in these tightly coupled systems is a future research area and we can anticipate advances in assimilation of soil moisture and surface fluxes of pollutants and greenhouse gases are expected.

The expected advance of the Earth System approach requires an increased research effort for the different communities to work more closely together to expand and to evolve the Earth observing system capacity. For what concerns the atmospheric

models, the improvement of aerosol-cloud interaction description, related sulphate production and oxidation processes in the





aqueous phase are important to provide a better estimate of aerosol and CCN production impacting on weather and climate. It is still to be investigated their impact on surface PM concentrations, especially in areas with very low SOx emissions like Europe during present days (Schrödner et al., 2020; Genz et al., 2020; Suter, 2020).

*6.3.2 Constraining models with observations*

The use of coupled regional scale meteorology-chemistry models for AQF represents a desirable advancement in routine operations that would greatly improve the understanding of the underlying complex interplay of meteorology, emission, and chemistry. Chemical species data assimilation along with increased capabilities to measure plume heights will help to better constrain emissions in forecast applications.

While important advances have been made, present challenges require advances in observing systems and assimilation systems
to support and improve air quality models. From the perspective of air quality modelling, there are still uncertainties in the emission estimates (especially those driven by meteorology and other conditions such as biomass burning and dust storms).

The impacts of data assimilation of atmospheric composition are limited by the remaining major gaps in spatial coverage in our observing systems. Major parts of the world have limited or no observations (Africa is an obvious case). This is changing thanks to the forthcoming new constellation of geostationary satellites (Sentinel-4, TEMPO and GEMS; Kim et al., 2020)
measuring atmospheric composition, and with the advances in low-cost sensor technologies. Machine learning applications will play important roles in improving predictions through better parameterizations, better ways to deal with bias, and new approaches to utilize heterogeneous observations, for example new models for relating aerosol optical depth (AOD) to surface $PM_{2.5}$ mass and composition.

Reanalysis products of aerosols and other atmospheric constituents are now being produced (Inness et al., 2019). These can
support many applications and continued development is strongly encouraged and will benefit from the observations and data assimilation advances discussed above.

*6.3.3. Multiscale interactions affecting urban areas*

For urban applications the main science challenges related to multiscale interactions involved the non-linear interactions of
urban heat island circulation and aerosol forcing and urban aerosol interactions with clouds and radiation. In order to improve air quality modelling for cities, advances are needed in data assimilation of urban observations (including meteorological, chemical and aerosol species), development of model dynamic cores with efficient multi-tracer transport capability, and the general effects of aerosols on the evolution of weather and climate on different scales. All these research areas are concerned with optimised use of models on massively parallel computer systems, as well as modern techniques for assimilation or fusion
of meteorological and chemical observation data (Nguyen and Soulhac, 2021).

In terms of atmospheric chemistry, the formation of secondary air pollutants (e.g. ozone, and secondary organic and inorganic aerosols) in urban environments is still an active research area and there is an important need to improve the understanding and treatment within two-way coupled chemistry-meteorology models.





Urban areas interact at many scales with the atmosphere through their physical form, geographical distribution and metabolisms from human activities and functions. Urban areas are the drivers with the greatest impact on climate change. The exchange processes between the urban surface and the free troposphere needs to be more precisely determined in order to define and implement improved climate adaptation strategies for cities and urban conglomerations. The knowledge of the 3D structure of the urban airshed is an important feature to define temperature, humidity, wind flow and pollutant concentrations

inside urban areas. Although computational resources had great improvement, time and spatial resolution are still imposing some limitations to the correct representation of urban features especially for the street scale. Urban areas are responsible for the urban heat island circulation, which interacts with other mesoscale circulations, such as the sea-breeze and mountain-valley circulations, determining the pathways of primary pollutants emitted in the atmosphere but even the production and transport of ozone (see e.g. Finardi et al., 2018) and secondary aerosols (Figure 13).


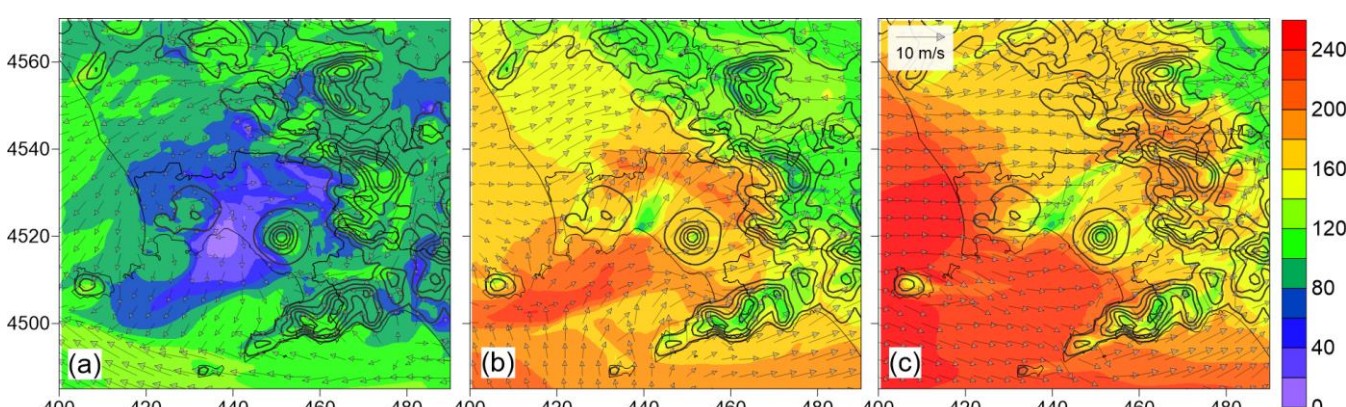

Figure 13. Near-surface ozone concentrations (ug m-3) predicted for 15 Jul 2015 at (a) 0800, (b) 1200, and (c) 1700 LST over Naples. Wind field at 10-m height is represented by gray arrows. From: Finardi et al., Journal of Applied Meteorology and Climatology, 57, 5, 1083-1099. https://doi.org/10.1175/JAMC-D-17-0117.1, 2018


Challenges remain on how to include scale dependent processes and interactions for urban and sub-urban scale modelling. These include spatial and temporal distribution of heat, chemical and aerosol emission source activities down to building-size resolution; flow modification at the micro scale level by the urban canopy structure and by the urban surface heat balance; enhancement/damping of turbulent fluxes in the urban boundary layer due to surface and emission heterogeneity; chemical

transformation of pollutants during their lifetime within the urban canopy sublayer. Obviously, the scale interaction issues facing air quality-meteorology-climate models are quite in line with those described in Chapter 5 for multi-scale air quality modelling. Thus, on coupling regional to urban and building scales, CTM coupled with urbanised meteorological models are needed to describe the city scale atmospheric circulation and chemistry in the urban airshed, the building and evolution of urban heat island, especially strong during heat waves (Halenka et al., 2019), including the combined effects of urban, sub-

urban and rural pollutant emissions. High spatial resolution is also needed to capture pollutant concentration spatial variability





at pedestrian level in an urban environment, answering e.g. epidemiological research questions or emergency preparedness issues. In the near future, microscale CFD, including LES modelling will probably become an appropriate tool for urban air quality assessment and forecasting purposes due to the expected continuous increase of computational resources enabling the inclusion of chemical reactions (Figure 14). Nevertheless, nowadays computational resources still limit their application to

short term episodes and often to stationary conditions, while climatological studies require for instance a multi-year approach. Parameterized street scale models (Singh et al., 2020a; Hamer et al., 2020; Kim et al., 2018) or a database created with CFD simulations of several scenarios (Hellsten et al., 2020) can be alternative ways for the downscaling from the mesoscale to the city and street scale, together with obstacle resolving Lagrangian particle models driven by Rokle type diagnostic flow models (Veratti et al., 2020; Tinarelli and Trini Castelli et al., 2019).

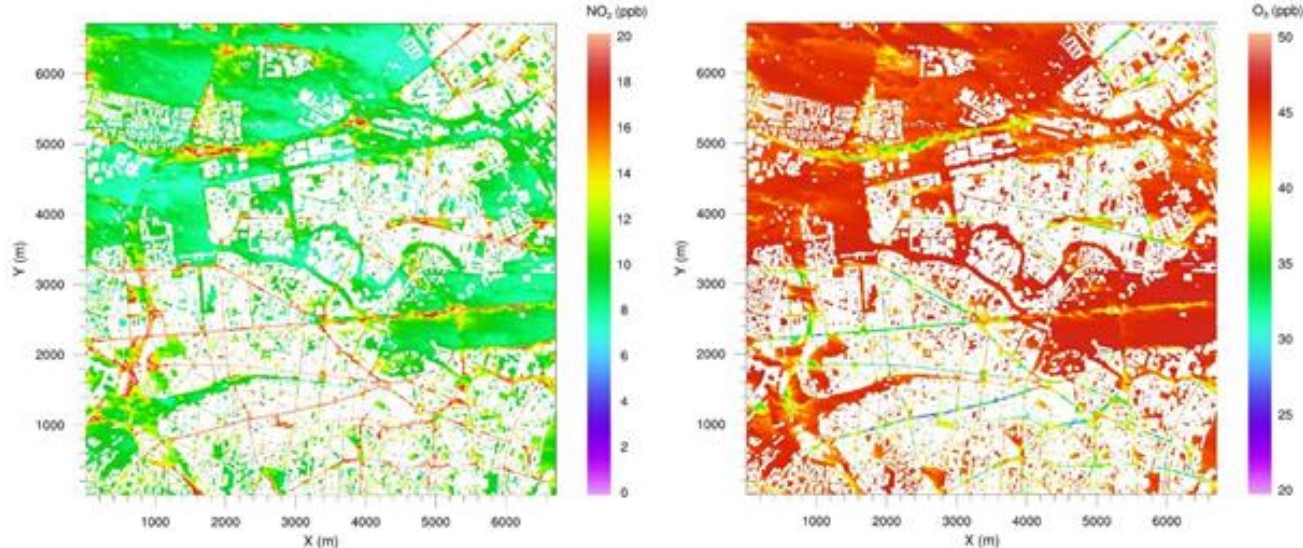


Figure 14: Modelled distribution of ground-level nitrogen dioxide (left) and ozone (right) at 8 pm CEST for a 6.7 km x 6.7 km subarea of Berlin around Ernst Reuter Platz. The simulation was performed with the chemistry mechanism CBM4 and a horizontal grid size of 10 m. From Khan et al, 2020; Geosci. Model Dev., https://doi.org/10.5194/gmd-2020-286.

*6.3.4 Nature-based solutions for improving air quality*

The growing interest for nature-based solutions requires the improvement of models' capability to describe biogenic emissions (Cremona et al., 2020) and deposition processes (Petroff et al., 2008; 2010) resolving the different species leaf features, biomass density and physiology. The balance between vegetation drag, pollutant absorption and BVOC emissions determines the net positive or negative air quality impact at local and city scales (Karttunen et al., 2020; San Josè et al., 2020; Santiago et

al., 2017; Jeanjean et al., 2017, Jones et al., 2019). In most cases this feature cannot be explicitly considered, being necessary some parametrized approach, such as the canyon one, to deal with it. Nevertheless, the present capabilities of UAQ models to





describe biogenic emission together with gas and particle deposition over vegetation covered surfaces (including green roofs and vertical green surfaces) need to be improved to include nature-based solutions impact into air quality plans evaluation.

## 7. Air quality exposure and health

### 7.1. Brief overview

A substantial amount of research has been conducted regarding the health effects of air pollution, especially those attributed to particulate matter (PM). Nevertheless, it is not conclusively known, which properties of PM are the most important ones in terms of the health impacts (e.g., Brook et al., 2010; Beelen et al., 2014; Pope et al., 2019; Schraufnagel et al., 2019). For

example, a review article by Hoek et al. (2013) addressed cohort studies and reported an excess risk for all-cause and cardiovascular mortality due to long-term exposure to $PM_{2.5}$.

In this section, we have therefore addressed three topical research areas, associated with air quality and health: (i) the health impacts of particulate matter in ambient air, (ii) the combined effects on human health of various air pollutants, heat waves and pandemics, and (iii) the assessment of the exposure of populations to air pollution. Research that has been reviewed is

based on selected international research projects and publications, but generally these are expected to reflect the general consensus, as both the projects and resulting publications involved a significant section of the air quality and health research community. Regarding pandemics, we will focus on the most recent one that has been caused by the severe acute respiratory syndrome coronavirus 2 (SARS-CoV-2). The research and interdependencies of these topics have been illustrated in Figure 15.

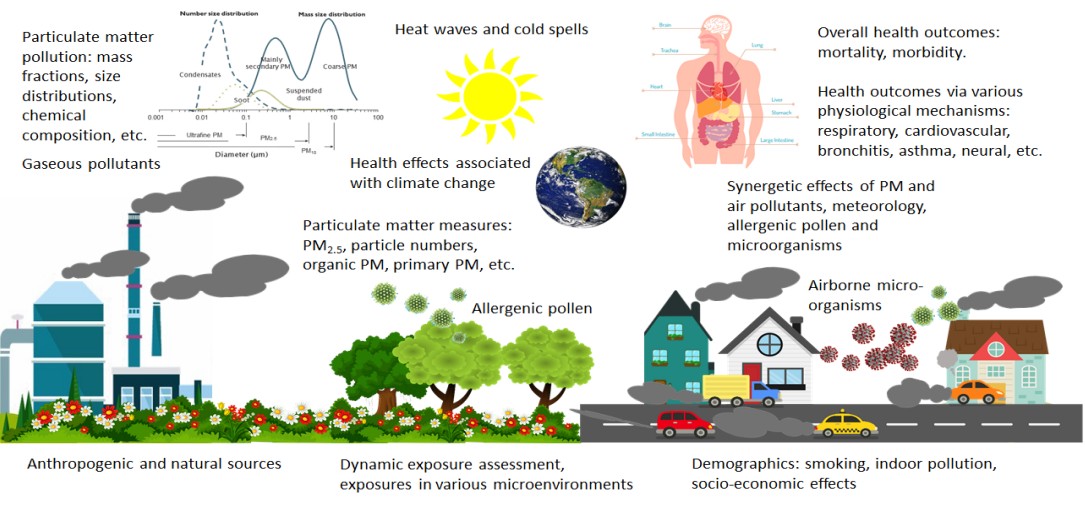


Figure 15. A schematic diagram, which illustrates some of the main factors in the evaluation of the exposure and health impacts of particulate matter.



As illustrated in the figure, particulate matter pollution originates from a wide range of anthropogenic and natural sources, and its characteristics can vary in terms of size distributions, chemical composition and other properties. The resulting health

outcomes also vary substantially, depending on the target physiological system or organ of an individual. In addition, the assessments of the interrelations of PM pollution and health outcomes are challenged by various combined and in some cases synergetic effects caused by, e.g., heat waves and cold spells, allergenic pollen and airborne microorganisms.

**7.2 Current status and challenges**


*7.2.1 Health impacts of particulate matter*

*(i) Overview of the health impacts of particulate matter pollution*

In addition to cardiovascular and respiratory diseases, exposure to ambient air PM may result in acute and severe health problems, such as cardiovascular mortality, cardiac arrhythmia, myocardial infarction (MI), myocardial ischemia and heart

failure (Dockery et al., 1993; Schwartz et al., 1996; Peters et al., 2001; Pope et al., 2002). The Organization for Economic Co-operation and Development (OECD) evaluated in its outlook (OECD, 2012) that PM pollution will be the primary cause of deaths of African population by 2050, in comparison to hazardous water and poor hygiene. Pražnikar and Pražnikar (2012) comprehensively addressed in their review several epidemiological studies throughout the world; they reported a strong association between the PM concentrations and respiratory morbidity, cardiovascular morbidity and total mortality.


Global assessments of air quality and health require comprehensive estimates of the exposures to air pollution. However, in many developing countries (e.g., Africa, see Rees at al., 2019) ground-based monitoring is sparse or non-existent; quality-control and the evaluation of the representativeness of stations may also be insufficient. An inter-disciplinary approach to exposure assessment for burden of disease analyses on a global scale has been recently suggested jointly by WHO, WMO and

CAMS (Shaddick et al., 2021). Such an approach would combine information from available ground measurements with atmospheric chemical transport modelling and estimates from remote sensing satellites. The aim is to produce information that is required for health burden assessment and the calculation of air pollution-related Sustainable Development Goal (SDG) indicators.

*(ii) Health effects associated with the long-term exposure to particulate matter*

Long-term exposure may potentially affect every organ in the body and hence worsen existing health conditions and it may even result in premature mortality (see e.g., a recent review by Schraufnagel et al., 2019; Brook et al., 2010; Brunekreef and Holgate, 2002; Beelen et al., 2015; Vodonos et al., 2018; Pope et al., 2019). For example, a review article by Hoek et al. (2013) addressed cohort studies and reported an excess risk for all-cause and cardiovascular mortality due to long-term exposure to $PM_{2.5}$. Beelen et al. (2015) analyzed an extensive set of data from 19 European cohort studies; they found that long-term

exposure to $PM_{2.5}$ sulphur was associated with natural-case mortality. Similar results regarding long-term exposure to $PM_{2.5}$ and mortality were also presented in other recent studies conducted by Vodonos et al. (2018), and Pope et al. (2019).





Studies conducted in the framework of the European Study of Cohorts for Air Pollution Effects (ESCAPE) project showed that long-term exposure to PM air pollution was linked to incidences of acute coronary (Cesaroni et al., 2014), cerebrovascular events (Stafoggia et al., 2014) and lung cancer in adults (Adam et al., 2015). Moreover, findings from the same project revealed

that other health effects related to PM air pollution were reduced lung function in children (Gehring et al., 2013), pneumonia in early childhood and possibly otitis media (MacIntyre et al., 2014), low birthweight (Pedersen et al., 2013) and the incidence of lung cancer (Raaschou-Nielsen et al., 2013). In addition, another finding of the ESCAPE project was the connection between traffic-related $PM_{2.5}$ absorbance and malignant brain tumors (Andersen et al., 2018).

The Biobank Standardisation and Harmonisation for Research Excellence in the European Union (BioSHaRE-EU) project,

which included three European cohort studies, presented the association between long-term exposure to ambient $PM_{10}$ and asthma prevalence (Cai et al., 2017). In the framework of three major cohorts (HUNT, EPIC-Oxford and UK Biobank) it was shown that, after adjustments for road traffic noise, incidences of CVD diseases were attributed to long-term PM exposure, (Cai et al., 2018). Hoffman et al. (2015) suggested that long-term exposure to both $PM_{10}$ and $PM_{2.5}$ is linked to an increased risk for stroke and it might be responsible for incidences of coronary events.

*(iii) Health effects associated with the short-term exposure to particulate matter*

Collaborative studies such as the APHENA (Air Pollution and Health: A European and North American Approach) and the MED-PARTICLES project in Mediterranean Europe have evidenced that short-term exposure to PM has been associated with all-cause cardiovascular and respiratory mortality (Katsouyanni et al., 2009; Zanobetti and Schwartz, 2009; Samoli et al., 2013; Dai et al., 2014;), hospital admissions (Stafoggia et al., 2013) and occurrence of asthma symptom episodes in children

(Weinmayr et al., 2010).

*(iv) Health effects associated with the chemical constituents of PM*

The chemical composition of PM is associated with the health effects related to PM concentrations, in addition to the mass concentrations of particulate matter (e.g., Maricq, 2007). Chemical composition of particles is complex; generally, it depends on the source origin of particles and their chemical and physical transformations in the atmosphere (e.g., Prank et al., 2016).

Some prominent examples of the components of PM are sulphate ($SO_4$), nitrate ($NO_3$), metals, elemental and organic carbon (Yang et al., 2018), ammonium ($NH_3$) (Pražnikar and Pražnikar, 2012), sea salt and dust (Prank et al., 2016).

The PM components also include biological organisms (e.g., bacteria, fungi, and viruses) and organic compounds (e.g., polycyclic aromatic hydrocarbons, PAHs and their nitro-derivatives (NPAHs) (Morakinyo et al., 2016; Kalisa et al., 2019). Their content can vary significantly with regard to time and for various climatic regions (Maki et al., 2015; Gou et al., 2016).

Hime et al. (2018) have reviewed studies which investigated which PM components could be mostly responsible for severe health effects. Such studies included the National Particle Component Toxicity (NPACT) initiative, which combined epidemiologic and toxicologic studies. That study concluded that the concentrations of $SO_4$, EC, OC and PM were mainly originated from traffic and combustion and had a significant impact on human health (Adams et al., 2015). The European Study of Cohorts for Air Pollution Effects (ESCAPE) project aimed at examining the association of elemental components of

PM (copper (Cu), iron (Fe), potassium (K), nickel (Ni), sulphur (S), silicon (Si), vanadium (V) and zinc (Zn)) with


inflammatory blood markers in European cohorts (Hampel et al., 2015). They focused, together with the TRANSPHORM project (Transport related Air Pollution and Health impacts – Integrated Methodologies for Assessing Particulate Matter), on investigating the relationship of these components with cardiovascular (CVD) mortality (Wang et al., 2014).

Moreover, other studies conducted within the framework of ESCAPE and the TRANSPHORM projects provided evidence that mortality was linked to long-term exposure to $PM_{2.5}$ sulphur (Beelen et al., 2015), as well as to the particle mass and nitrogen oxides ($NO_2$ and $NO_x$) (Beelen et al., 2014). As part of the NordicWelfAir project, Hvidtfeldt et al. (2019b) connected the risks of being exposed long-term to $PM_{2.5}$, $PM_{10}$, BC, $NO_2$ with all-cause and CVD mortality. In another paper, Hvidtfeldt et al. (2019a) demonstrated the association between long-term exposure to $PM_{2.5}$, elemental and primary organic carbonaceous particles (BC/OC), secondary organic aerosols (SOA) and all-cause mortality. They also demonstrated the connection between $PM_{2.5}$, BC/OC, and secondary inorganic aerosols (SIA) with CVD mortality. Recently, a continuation of this study included all Danes born between 1921 and 1985, showing higher mortality related to exposure to $NO_2$, $O_3$, $PM_{2.5}$ and BC (Raaschou-Nielsen et al., 2020).

In the framework of the Particle Component Toxicity (NPACT) project, Lippmann et al. (2013) showed that $PM_{2.5}$ mass, and EC were linked to all-cause mortality; EC was also connected with ischemic heart disease mortality. The latter result was quite similar to Ostro et al.'s (2010, 2011, 2015) findings, including OC, $SO_4$, $NO_3$, and SO in addition to EC. Concerning cardiopulmonary disease mortality, a strong association was observed for the exposure to $NO_3$ and $SO_4$ (Ostro et al., 2010, 2011). Luben et al. (2017) and Hoek et al. (2013) in their reviews observed the association of BC with cardiovascular disease hospital admissions and mortality.

In a meta-analysis work conducted by Achilleos et al. (2017), elemental carbon (EC), black carbon (BC), black smoke (BS) organic carbon (OC), sodium (Na), silicon (Si), and sulfate ($SO_4$) were associated with all-cause mortality, and BS, EC, nitrate ($NO_3$), ammonium ($NH_4$), chlorine (Cl), and calcium (Ca) were linked to CVD mortality. In addition, some American cohort studies pointed out that long-term exposure to $SO_4$ was positively connected with all-cause, cardiopulmonary disease and lung cancer mortality (Dockery et al., 1993; HEI, 2000; Pope et al., 2002; Ostro et al., 2010).

In addition, other kinds of severe health effects related to PM components have been reported. For example, Wolf et al. (2015) showed that long-term exposure to PM constituents, especially of K, Si, and Fe, which are indicators of road dust, provoked coronary events. The findings of a systematic review, where 59 studies were included, indicated that chronic obstructive pulmonary diseases (COPD) emergency risk was attributed to short-term exposure to $O_3$ and $NO_2$, whereas short-term exposure to $SO_2$ and $NO_2$ was responsible for COPD acute risk in developing countries (Li et al., 2016). The review of Li et al. (2016) also reported that short-term exposure to $O_3$, CO, $NO_2$, $SO_2$, $PM_{10}$, and $PM_{2.5}$) was linked to respiratory risks.

Poulsen et al. (2020), using detailed modelling and Danish registers from 1989–2014, showed stronger relationships between primarily emitted black carbon (BC), organic carbon (OC) as well as combined carbon (OC/BC) a malignant brain tumors. Furthermore, the risk for lung cancer was linked to several different compounds and sources of aerosol particles; they found that particles containing S and Ni might be two of the most important components associated with lung cancer (Raaschou-Nielsen, 2016). Park et al. (2018) found that $PM_{2.5}$ particles emitted from diesel and gasoline engines were more toxic for



humans than, for example, particles from biomass burning or coal combustion. In a recent study, it was concluded that traffic-specific PM components, and in particular $NH_4$ and $SO_4$, lead to higher risks of stroke than PM components linked to industrial sources (Rodins et al., 2020).

*(vi) The uncertainties associated with concentration-response functions*

Based on previous research, WHO/Europe have recommended in 2015 a set of linear concentration-response functions for the

main air pollutants and related health outcomes (Héroux et al. 2015). These functions are currently widely used for health assessments, e.g., on a European scale by EEA. EEA (2019) estimated that more than 340000 premature deaths per year in Europe can be related to the exposure to $PM_{2.5}$. However, it is currently widely debated, what is the optimal shape of the concentration-response functions and whether there should be a threshold or lower limit.

A prominent example is the highly cited study by Burnett et al. (2018) on the developments of the Global Exposure Mortality

Model (GEMM). By combining data from 41 cohorts from 16 different countries, Burnett and colleagues have constructed new hazard ratio functions that to a wider degree than previous studies include the full range of the global exposure to outdoor $PM_{2.5}$. The GEMM functions for $PM_{2.5}$ and nonaccidental mortality generally follow a supralinear association at lower concentrations and near-linear association at higher concentrations (Burnett et al., 2018).

The GEMM functions would indicate that health impacts related to $PM_{2.5}$ exposure have been underestimated, both at the

global and regional scales. In a recent European study on cardiovascular mortality, the GEMM functions were combined with concentration fields from a global atmospheric chemistry–climate model. The results pointed towards a total of 790000 premature deaths attributed to air pollution in Europe per year, which is significantly higher than, e.g., the value previously estimated by EEA (Lelieveld et al., 2019). Several reviews or meta-analyses have focused on low exposure levels; the conclusion has been that significant associations can be found between $PM_{2.5}$ and health effects also at levels below the

concentrations of 10-12 ug/m$^3$. These values are equal or below, e.g., the WHO guidelines (10 ug/m$^3$) and the US EPA standards (12 ug/m$^3$) (Vodonos et al., 2018; Papadogeorgou et al., 2019).

*(vii) The use of high-resolution multi-decadal datasets for extensive regions*

Developments of air pollution modelling and more efficient computing resources have made it possible to compute high-resolution air pollution datasets that cover larger regions, as well as longer, even multi-decadal time periods (Figure 16). The

combination of such data with national or international health registers, or cohorts from several countries improves the representativeness of statistical analyses. The use of more extensive datasets will also reduce the selection biases related to the sizes of the cohorts.

This has resulted in, for example, a better detection of the links between air pollution exposure and new health endpoints, such as, e.g., psychiatric disorders (e.g., Khan et al., 2019a; Antonsen et al., 2020) and cognitive abilities (e.g. Zhang et al., 2018).

Based on high resolution (1 km x 1 km) air pollution data covering the period 1979-2015 and population-based data from the Danish national registers, Thygesen et al. (2020) found that exposure to air pollution (specifically $NO_2$) during early childhood was associated with the development of Attention-Deficit/Hyperactivity Disorder (ADHD).



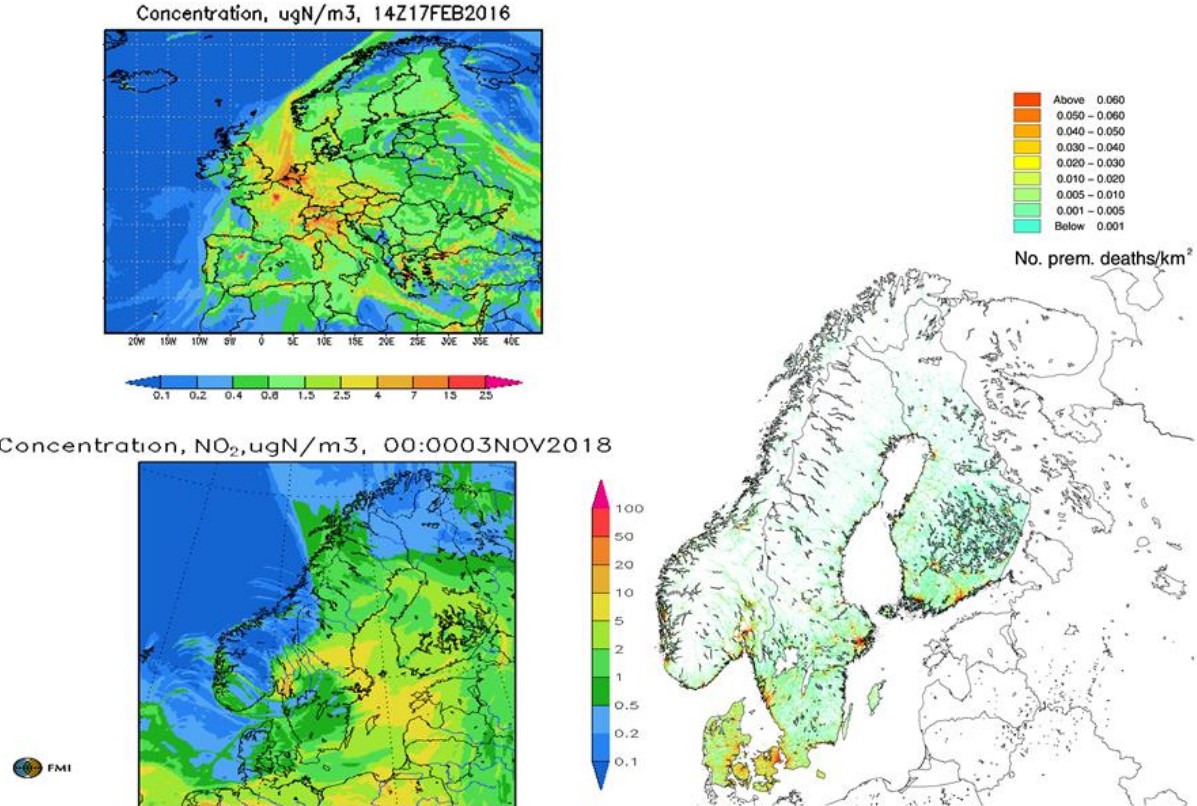

Figure 16. An illustration on how concentration predictions on a high spatial and temporal resolution (panels on the left-hand side) could be used for high resolution health impact assessments (panel on the right-hand side). The concentration distributions were predicted with the chemical transport model SILAM. The health impact assessment was made with the EVA model in a high-resolution setup for the Nordic region, giving an estimate of the number of premature deaths due to exposure to air pollution (Lehtomaki et al., 2020). The concentrations used in EVA was from the chemical transport system DEHH-UBM providing 1 km x 1 km concentration across the Nordic region.

Kukkonen et al (2018) presented a multi-decadal, global and European scale modelling of a wide range of pollutants, and the finer resolution urban-scale modelling of $PM_{2.5}$ in the Helsinki Metropolitan Area. All of these computations were conducted for a period of 35 years, from 1980 to 2014. The regional background concentrations were evaluated based on reanalyses of the atmospheric composition on global and European scales, using the chemical transport model SILAM. These results have been used for health impacts assessments by Siddika et al. (2019 and 2020). The predicted air quality and meteorological data is available to be used also in any other region globally in health impact assessments.

*7.2.2 Combined effects of air pollution, heat waves and pandemics on human health*





It is widely known that poor air quality has severe impacts on human's immune system (Genc et al., 2012). In particular, some of the acute health effects include chronic respiratory and cardiovascular diseases (Ghorani-Azam et al., 2016), respiratory infection (e.g., Conticini et al., 2020), and even cancer and death (IOM 2011; Villeneuve et al., 2013). Polluted air can cause, for example, damages in epithelial cilia (Cao et al., 2020), which leads to a chronic inflammatory stimulus (Conticini et al., 2020). It has also been shown that the SARS-CoV-2 can stay viable and infectious on aerosol particles that are smaller than 5 μm in diameter for more than three hours (van Doremalen et al., 2020). Therefore, atmospheric pollutants might play an important role in spreading the virus.

*(i) The role of air pollution in pandemics*

Previously, Cui et al. (2003) found that the long-term exposure to moderate or high air pollution levels was positively correlated with mortality caused by SARS-CoV-1 in the Chinese population. Therefore, it is possible that poor air quality would enhance the risk of mortality during epidemics or pandemics, such as the COVID-19 disease, caused by the SARS-CoV-2. Moreover, poor air quality can enhance the human health effects of heat waves, cold spells, and allergenic pollen. This is because exposure to ambient air pollutants together with microorganisms tend to make the health impacts of pathogens more severe; at the same time, they weaken human immunity, resulting in an increased risk of respiratory infection (e.g., Xu et al. 2016, Horne et al. 2018, Xie et al. 2019, Phosri et al. 2019).

Conticini et al. (2020) concluded that weakened lung defence mechanisms due to continuous exposure to air pollution could partly explain the higher morbidity and mortality caused by SARS-CoV-2 in areas of poor air quality in Italy. Zhu et al. (2020) used the data of daily confirmed COVID-19 cases, air pollution and meteorology from 120 cities in China to study the association between the concentrations of ambient air pollutants ($PM_{2.5}$, $PM_{10}$, $SO_2$, $CO$, $NO_2$ and $O_3$), and COVID-19 cases. By applying a generalized additive model, they found a significant correlation between $PM_{2.5}$, $PM_{10}$, $CO$, $NO_2$ and $O_3$ and daily counts of confirmed COVID-19 patients, while $SO_2$ was negatively associated with the daily number of new COVID-19 cases.

Ogen (2020) studied 66 regions in Italy, Spain, France and Germany; he also found a spatial correlation between high $NO_2$ concentration and the fatality from COVID-19. According to this study, 83% of all fatalities occurred in the regions having maximum $NO_2$ concentration above 100 $\mu$mol/m$^2$ and only 1.5% of all fatalities took place in areas, in which maximum $NO_2$ concentration was below 50 $\mu$mol/m$^2$. However, Pisoni and Van Dingenen (2020) didn't find a similar phenomenon in the UK, where the number of deaths was higher than in Italy, despite a significantly lower $NO_2$ concentration.

Xie and Zhu (2020) used temperature data from 122 cities mainly in Eastern part of China and observed a linear relationship between ambient temperature and daily number of confirmed COVID-19 counts in cases when the temperature was below 3°C. At higher temperatures, no correlation was found. This indicates that daily counts of COVID-19 did not decline at warmer atmospheric conditions; although such a dependency was expected based on the previous studies related to coronaviruses SARS-CoV and MERS-CoV (e.g. Van Doremalen et al. 2013, Bi et al. 2007, Tan et al. 2005). However, the study of Xie and Zhu (2020) was conducted in winter; the highest temperatures were around 27°C.



Chen et al. (2017) investigated statistically the correlation between influenza incidences and the concentrations of $PM_{2.5}$ in 47 Chinese cities 14 months during 2013 - 2014. Based on the results, they concluded that about 10% of the influenza cases were induced by the exposure to ambient $PM_{2.5}$. They also classified the days as cold, moderately cold, moderately hot, and hot separately to each city and found that the risk for influenza transmission associated with ambient air $PM_{2.5}$ was enhanced during cold days.

*(ii) Combined effects of air pollutants or heat waves*

Siddika et al. (2019) found that the prenatal exposure to both $PM_{2.5}$ and $O_3$ increased the risk of preterm birth in Finland in the 1980's. The risk was more pronounced, if the mother was exposed to both higher $PM_{2.5}$ and higher $O_3$ concentrations. They explained that $O_3$ might deplete antioxidants in the lung and therefore the defence mechanism needed against reactive oxygen species formation was reduced due to the exposure to $PM_{2.5}$. Also, the $O_3$ concentrations can cause changes in lung epithelium so that it is more permeable for particles to absorb into the circulatory system. The population selected to the study were living in southern Finland in the 1980's, in relatively good air quality. However, the concentrations of many pollutants, e.g. those of $PM_{2.5}$, have been shown to have been in the 1980's twice as high, compared with the corresponding pollutant levels in the same region during the second decade of the 21st century (Kukkonen et al., 2018).

Wang et al. (2020) presented that $PM_{2.5}$ exposure strengthened the effect of moderate heat waves (short or only moderate temperature rise) associated with preterm births during January 2015 - July 2017 in Guangdong province, China. However, during the intensive heat waves, the effects were not additive.

Analitis et al. (2018) studied synergetic effects of temperature, $PM_{10}$, $O_3$ and $NO_2$ on cardiovascular and respiratory deaths. They found some correlation between the effects of high ambient temperatures, and that caused by $O_3$ and $PM_{10}$ concentrations. However, during the heat waves, no clear synergetic effect was found. In a review article, Son et al. (2019) concluded that there is some evidence between the mortality related to high temperatures and air pollution.

Li et al. (2017) wrote a comprehensive literature review about the role of temperature and air pollution on mortality. They determined individual spatial temperature ranges and grouped them in "low", "medium", and "high" based on the information given in each study about typical local weather conditions. After a careful selection based on the quality of the datasets, they performed a meta-analysis by using data of 21 studies; they found that high temperature significantly increased the risk of non-accidental and cardiovascular mortality, caused by the exposure to $PM_{10}$ or $O_3$. The risk of cardiovascular mortality due to $PM_{10}$ decreased during low temperature days in the prevailing climate. However, the exposure to both low temperature and the concentrations of $O_3$ increased the risk of non-accidental mortality. The similar effects were not found for the concentrations of $SO_2$ or $NO_2$ and temperature. Lepeule et al. (2018) found that short-term rise in outdoor air temperature and relative humidity were linked to deteriorated lung function of elderly people. A simultaneous exposure to black carbon amplified the health effects.



There have also been discussions, whether antibiotic-resistant bacteria due to the excessive use of antibiotics could explain
mortality in some areas (e.g., Kirchhelle et al., 2020; Strathdee et al., 2020). Strathdee et al. (2020) reported that 6.9% of the
COVID-19 patients are associated with bacterial infection, which could be fatal, without the use of a proper medication.

### 7.2.3 Estimation of exposures

#### (i) Modelling of individual exposure

The currently available epidemiological studies use measured or modelled outdoor concentrations in residential areas or at
home addresses, to correlate the concentrations with health effects. However, several studies have pointed out that it is critical
to use the exposure of people as indicators for the health effects (Kousa et al., 2002, Soares et al., 2014; Kukkonen et al.,
2016b; Smith et al., 2016, Singh et al. 2020a, Li and Friedrich 2019, Li 2020). It is obvious that the effects of air pollutants on
human health are caused by the inhaled pollutants, instead of the pollutants at a certain point or area outdoors. Thus, exposure
is a much better indicator for estimating health risks than outdoor concentrations. The individual exposure of a person to air
pollutants is defined here as the concentration of pollutants at the sites where the person is staying weighted by the length of
stay at each of the places of stay and averaged over a certain time span, e.g. a year. The places of stay are in this context called
microenvironments. Exposure of a group of people with certain features (e.g. sex, age, place of living) is the average exposure
of the individuals in the subgroup. In general, the exposure of a person is calculated by first estimating the concentration of air
pollutants in the microenvironments where the person or population subgroup are staying and then by weighting this
concentration with the length of time the person has been at the respective microenvironment (Li and Friedrich, 2019, Li,
2020). The result of modelling exposure can be verified by measuring the exposure with personal sensors (e.g. Dessimon et
al., 2021)

Exposures to ambient concentrations of $PM_{2.5}$ can be substantially different in different microenvironments. The concentrations
in microenvironments can be either measured or modelled. Computational results of activity based dynamic exposures by
Singh et al. (2020a) demonstrate that the total population exposure was over one quarter (-28%) lower on a city-wide average
level, compared with using simply outdoor concentrations at residential locations, in case of London in the 2010's. Smith et
al. (2016) have shown by modelling that exposure estimates based on space-time activity was 37% lower than the outdoor
exposure evaluated at residential addresses in London. However, this proportion will be different for other urban regions and
time periods, or when addressing specific population sub-groups.

The exposure to particulate matter is substantially influenced by indoor environments, as people spend 80–95% of their time
indoors (e.g., Hänninen et al., 2005; Schweizer et al., 2007). Indoor air quality mainly depends on the penetration of pollutants
in outdoor air, on ventilation, and on indoor pollution sources. For estimating the indoor concentration, commonly a mass-
balance-model is applied (Hänninen et al. 2004, Wallace and Williams 2005, Li 2020). With a mass-balance model, the indoor
concentration is calculated based on the outdoor concentration, a penetration factor, the air exchange rate, the decay rate, the
emission rates of the indoor sources and the room volume, and, if available, by parameters of the mechanical ventilation
system.





A complex stochastic model has been developed for estimating the annual individual exposure of people or groups of people in the European Union to $PM_{2.5}$ and $NO_2$, using characteristics of the analysed subgroup, such as age, gender, place of

residence, socioeconomic status a.s.o. (Li et al., 2019a; Li et al., 2019c; Li and Friedrich, 2019; Li, 2020). The probabilistic model incorporates an atmospheric model for estimating the ambient pollutant concentrations in outdoor microenvironments and a mass-balance model for estimating indoor concentrations stemming as well from outdoor concentrations as from emissions from indoor sources. Time-activity patterns (which specify, how long a person stays in each microenvironment) were derived from an advancement of the Multinational Time Use Study (MTUS) (Fisher and Gershuny, 2016). The exposures

can also be estimated for past years. It is therefore possible to analyse the exposure for the whole lifetime of a person, by using a lifetime trajectory model that predicts retrospectively the possible transitions in the past life of a person.

An exemplary result from Li and Friedrich (2019) is shown on Figure 17. It displays that the $PM_{2.5}$ annual average exposure averaged over all adult persons living in the EU increased since the 1950s from 19.0 (95% Confidence Interval, CI: 3.3-55.7) μg/m³ to a maximum of 37.2 (95% CI: 9.2- 113.8) μg/m³ in the 1980s. The exposure then declined gradually afterwards until

to 20.1 (95% CI: 5.8-51.2) μg/m3. Indoor air pollution contributes considerably to exposure. In recent years more than 45% of the $PM_{2.5}$ exposure of an average EU citizen has been caused by indoor sources.

The most important indoor sources are environmental tobacco smoke, frying, wood burning in open fireplaces and stoves in the living area, and the use of incense sticks and candles. In addition, nearly all indoor activities include abrasion processes that produce fine dust. For $NO_2$, indoor sources cause around 24% of the exposure, with main contributions from cooking with

gas and from biomass burning in stoves and open fireplaces (Li and Friedrich, 2019). The solid black line in Fig. 7.3 shows the background outdoor concentration at the places where EU citizens spent their lives on average. Urban background concentrations refer to urban concentrations that are not in the immediate vicinity of the emission sources, especially of streets. The average exposure is higher than the average outdoor background concentration. Epidemiological studies correlate outdoor concentrations with health risks and thus neglect the exposure caused by indoor sources. Such studies therefore implicitly

assume that the contribution of indoor sources is the same at all places and for all people. Thus, calculating the burden of disease using exposures to $PM_{2.5}$ will yield years of lives lost and other chronic diseases that are about 40% higher than those calculated with outdoor background concentrations (Li, 2020). Using exposure data, a 70-year-old male EU citizen will have experienced a reduction of life expectancy of about 13 (CI 2-43) days per year of exposure with $PM_{2.5}$, since the age of 30 (Li, 2020). For a person who is now 40 years old or younger, the life expectancy loss per year will be less than half as much as that

of a 70-year-old person.





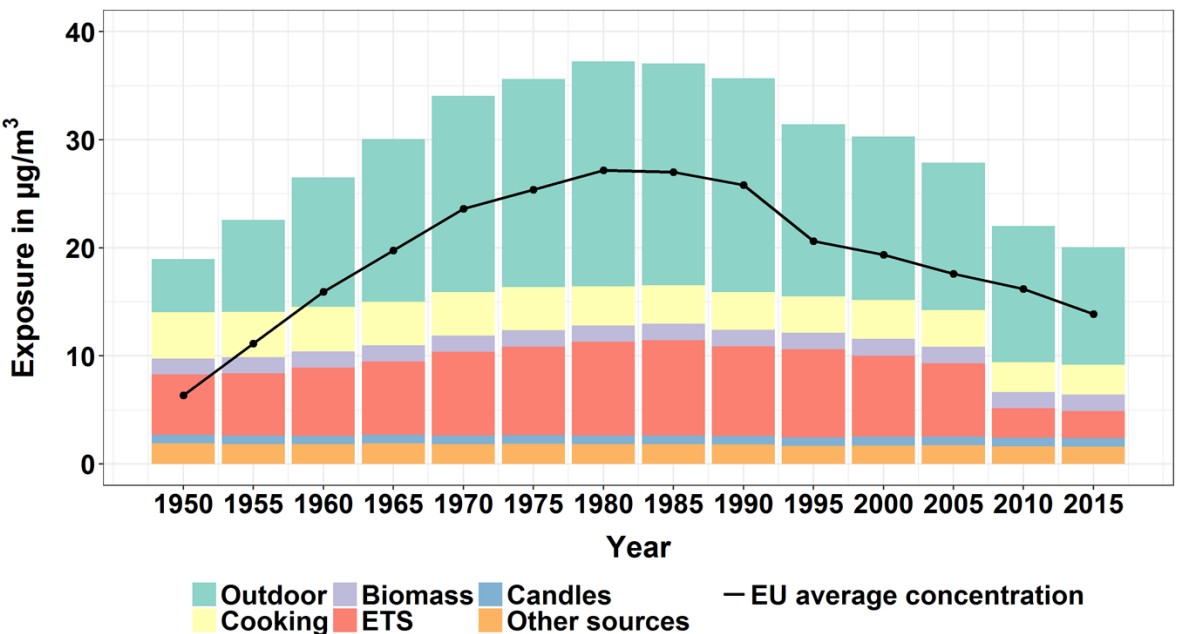

Figure 17. Temporal evolution of the annual average exposure of EU adult persons to $PM_{2.5}$ from 1950 to 2015 (Li and Friedrich 2019). All sources, except 'outdoor', refer to indoor sources. ETS = environmental tobacco smoke.

A similar approach for estimating the 'integrated population-weighted exposure' of the Chinese population to $PM_{2.5}$ has been used by Aunan et al. (2018) and Zhao et al. (2018). Aunan et al. (2018) estimated a mean annual averaged $PM_{2.5}$ exposure of 103 [86–120] μg/m³ in urban areas and 200 [161–238] μg/m³ in rural areas, with 50 % in urban areas and 78% in rural areas originating from domestic biomass and coal burning.

*(ii) Measurements of indoor concentrations and individual exposure*

Zhao et al. (2020a) made measurements of PM concentrations of different size classes in 40 homes in the German cities of Leipzig and Berlin. Measurements were made in different seasons simultaneously inside and directly outside the homes. Only homes without smokers were analysed. Mean annual indoor $PM_{10}$ concentrations were 30% larger than the outdoor concentrations near the houses. However, the mean indoor concentration of $PM_{2.5}$ was 6% smaller than the outdoor concentration. The infiltration factor was evaluated to be 0.5. They therefore concluded that the indoor concentration of $PM_{2.5}$

was considerably influenced by both indoor and outdoor sources; the former included cooking and burning of candles.

Vardoulakis et al. (2020) made a comprehensive literature review on indoor concentration of selected air pollutants associated with negative health effects and listed the main results (concentrations) and other features (e.g. main sources) for the analysed studies. They express the need for 'standardized IAQ (indoor air quality) measurement and analytical methods and longer monitoring periods over multiple sites'.

Some studies have focused on the measurements of personal exposure to ambient air concentrations using portable instruments in different microenvironments. For instance, Dessimon et al. (2021) describe the development and use of a personal sensor



for measuring PM$_1$, PM$_{2.5}$ PM$_{10}$, BC, NO$_2$ and VOC together with climate parameters, location and sleep quality. Clearly, such measurements can provide valuable and accurate information on the spatial and temporal variations of exposure, and they can be used to validate exposure models.


### 7.3 Emerging challenges
### 7.3.1 Emerging challenges for health impacts of particulate matter

*(i)      Classification of particulate matter measures and characteristics, and potential health outcomes*

Various studies have described PM in terms of the overall aerosol properties, such as the mass fractions (most commonly PM$_{2.5}$

and PM$_{10}$), the size distributions (mass, area, volume), the chemical composition, primary versus secondary PM, the morphology of particles, and source attributed PM. Some studies have adopted more specific properties of PM derived based on the above-mentioned overall properties. Such properties include, to mention a few of the most common ones, particle number concentrations (PNC), PNC's evaluated separately for each particulate mode, ultrafine PM, nanoparticles, secondary organic PM, primary PM, other combinations of chemical composition, suspended dust (specific class of source-attributed

PM), the content of metals, and toxic or hazardous pollutants.

An important emerging area is therefore to understand better which PM properties or measures would optimally describe the resulting health impacts. As mentioned above, one potentially crucial candidate for such a property is particulate number concentration (PNC). Kukkonen et al. (2016a) presented the modelling of the emissions and concentrations of particle numbers on a European scale and in five European cities. Comparison of the predicted PNC's to measurements on regional and urban

scales showed a reasonable agreement. However, there are still substantial uncertainties, especially in the modelling of the emissions of particulate numbers.

Health outcomes can also be classified as overall outcomes, and physiologically more specific outcomes. Prominent examples of overall outcomes are mortality and morbidity. Relatively more specific impacts include, e.g., respiratory and cardiovascular impacts, bronchitis, asthma, neurological impacts, impacts on specific population groups (such as, e.g., infants, children, the

elderly, prenatal impacts, and persons suffering various diseases).

*(ii)      Uncertainties and challenges on evaluating the health impacts of particulate matter*

Additional uncertainty is included in the concentration versus health response functions, which may be, e.g., linear or logarithmic, or combination of both of these, and including or excluding a threshold value. Clearly, in the evaluation of the health impacts of PM, there are also numerous confounding factors. For population-based studies, these include active and

passive smoking, sources and sinks that influence indoor pollution, gaseous pollutants, allergenic pollen, socio-economic effects, age, health status and gender.

In addition, the health impacts of PM are related to the impacts of other environmental stressors, such as heat waves and cold spells, allergenic pollen and airborne microorganisms. Commonly, it is challenging to decipher such effects in terms of each other. The factors may also have either synergistic or antagonistic effects. For instance, the health impacts of PM may be





enhanced in the presence of a heat wave. The impacts of various PM properties are also known to be physiologically specific, i.e., such a property may contribute to a certain health outcome, but not to some other outcomes.

In summary, there are many associations of various PM properties and measures to various health outcomes. Some of these inter-dependencies are known relatively better, either qualitatively or quantitatively, while there are also numerous associations, which are currently known poorly.

*(iii)*        *Research recommendations for deciphering the impacts of various particulate matter properties*

For evaluating the relative significance of various PM properties and measures on human health, a denser measurement network on advanced PM properties would be needed, both on regional and urban scales. The required PM properties would include, in particular, size distributions and chemical composition. Clearly, such a network would be especially valuable in cities and regions, which include high-quality population cohorts. The most important requirement in terms of PM modelling

would be improved emission inventories, which would also include sufficiently accurate information on particle size distributions and chemical composition.

Pražnikar and Pražnikar (2012) and Rodins et al. (2020) stressed the importance of the identification of the specific sources and the evaluation of the chemical composition of PM responsible for acute health effects. For instance, Hime et al. (2018) reported that there is a severe lack of epidemiological studies investigating the health impacts originating from exposure to

ambient diesel exhaust PM. In addition, they pointed out that there is no clear distinction between PM originating from diesel emissions and from other sources; thus, there is a limited number of studies assessing the respective health impacts.

Despite the substantial amount of research on the impacts of various PM properties and measures, the results on the importance of the more advanced measures (in addition to PM mass fractions) are to some extent inconclusive. One reason for this uncertainty is that there are so many associations of various PM properties and measures to various health outcomes.

One prominent emerging area is the evaluation of long-term, multi-decadal concentrations and meteorology on a sufficient spatial resolution. Long-term and lifetime exposures are known to be more important in terms of human health, compared with short-term exposures. Comprehensive datasets are therefore needed, which will include multi-decadal evaluation of air quality, meteorology, exposure and a range of health impacts. Some first examples of such datasets have already been reported (Kukkonen et al., 2018, Siddika et al., 2019, Raaschou-Nielsen et al., 2020, Thygesen et al., 2020, Siddika et al., 2020).

Although it is clear that chronic diseases and chronic mortality are caused by exposure to fine PM over many years, information is scarce regarding the critical length of the exposure period in terms of, e.g., premature death.

Elderly people are generally regarded as more sensitive to air pollution. It is well-known that the overall trend towards an aging population can counteract against improvements in air pollution levels in the future (e.g., Geels et al., 2015). However, detailed knowledge is scarce regarding whether exposure during specific periods in life can increase the risk of chronic

morbidity or mortality. Inequalities in both the exposure to PM and the related risks across different population groups (like gender, ethnicity, socioeconomic position, etc) due to underlying differences in health status will also need further investigations, to ascertain that future mitigation strategies will benefit all population groups (Fairurn et at., 2019; Raaschou-Nielsen et al., 2020). With regard to chronic diseases caused by $NO_2$, it is still uncertain, whether $NO_2$ is the cause of the



diseases or whether other pressures or a combination of pressures, that are correlated with the $NO_2$ concentration, are
responsible.

The introduction of green spaces in urban areas can contribute either negatively or positively to air quality. Green spaces can also potentially act as sources of allergenic pollen. The health impacts of introducing green spaces would therefore need to be clarified (Hvidtfeldt et al, 2019b; Engemann et al., 2019).

**7.3.2 Emerging challenges for the combined effects of air pollution and viruses**

Studying the combined effects of air pollution, heat waves or cold spells, and viruses is challenging, due to numerous confounding factors and incidental correlations. For instance, air pollution is commonly a serious problem in areas where the population density is also high. The high population density tends to allow viruses to spread more easily, compared with the situation in more sparsely populated areas.

Morbidity or mortality due to pandemics is also dependent on the age distribution of the population, cultural and social differences, the level of health care, living conditions, common hygiene and other factors. Clearly, such demographic differences should be taken into account, when comparing the frequencies of virus infections in different areas.

Due to limited data and the still evolving COVID-19 pandemic, it is difficult to draw definite conclusions related to the role of air pollution or meteorological drivers (like temperature or relative humidity) in transmission rates or in the severity of the

disease. Global interdisciplinary studies, open data sharing, and scientific collaboration are the key words towards better understanding of the interaction of COVID-19 and meteorological and environmental variables. Moreover, it is important to know what the role is of, for example PM in spreading the SARS-CoV-2. Indoor or laboratory dispersion experiments are needed to find out if the virus is spreading not only in droplets, but also in smaller aerosol particles. Together with a fully validated computational fluid dynamics model, it is possible to get facts about dispersion distances in different conditions and

to study for example the effect of ventilation systems, furniture placements, air cleaners, etc. to give information-based recommendations to make the environment as safe as possible without complete lockdowns.

Allergenic pollen can periodically cause substantial health impacts for numerous people. As PM is transported in the atmosphere, microbial pathogens such as bacteria, fungi and viruses can be attached on the surfaces of particles (Morakinyo et al., 2016); clearly, these may provide an additional risk (Kalisa et al., 2019). The combination of both biological and

chemical components of PM can further enhance some health effects, such as, e.g., asthma and COPD (Kalisa et al., 2019).

Adverse health impacts can also be associated with short-term exposure to atmospheric particles. The short-term impacts may be important, especially during air pollution episodes. Such episodes may be caused, for example, by the emissions originating from wild-land fires, dust storms or severe accidents. Episodes can also be caused by extreme meteorological conditions; two prominent examples are heat waves and extremely stable atmospheric conditions.




### 7.3.3 Other emerging challenges

First attempts have been made to quantify exposures by estimating concentrations in microenvironments, combined with space-
time-activity data. However, improvements will be necessary for virtually all the components of exposure modelling.
Regarding the emissions used for concentration modelling, especially the evaluation of the emission rates from indoor sources
should be improved on a broader empirical basis.

Emission rates depend on human behaviour, for which more detailed information is needed. E.g., how many people smoke
indoors, and how many family members are exposed to passive smoking? Are kitchen hoods used when cooking and frying?
How often are chimneys open, and how often wood stoves are used? For estimating indoor concentrations, one would need
further information of ventilation habits in different seasons. Better information would be needed regarding the use of
mechanical ventilation with heat recovery in new homes and office buildings.

To validate the results of the indoor air pollution models, one would need more measurements of indoor air concentrations in
rooms with different emission sources and ventilation systems. Furthermore, measurements of concentrations are needed in
various microenvironments, such as, e.g., in cars, buses and the underground.

With growing knowledge of the relation between exposure and health impacts, more detailed exposure indicators might be
necessary. For instance, a further differentiation according to size and species of $PM_{2.5}$ and $PM_{10}$ might be needed, furthermore,
the specification of the temperature and of the breathing rate, in other words the intake of pollutants with the respiration. The
use of more dynamic exposure data in epidemiological studies in the future could substantially improve the accuracy of health
impact assessments.

## 8. Air Quality Management and Policy Development

### 8.1 Brief Overview

While making decisions about air quality management and policy development are based on political consideration, it is a
scientific task to provide evidence and decision support for designing efficient air pollution control strategies that lead to an
optimization of welfare of the population. To do this, integrated assessments of the available policies and measures to reduce
air pollution and their impacts are made. In such assessments, two questions are addressed:
(i) Is a policy or measure or a bundle of policies or measures beneficial for society? Does it increase the welfare of society,
i.e., do the benefits outweigh the costs (including disadvantages, risks, utility losses)?
(ii) If several alternative policy measures or bundles of policy measures are proposed, how can we prioritize them according
to their efficiency, i.e., which should be used first to fulfil the environmental aims?

To analyse these questions, two methodologies have been developed: cost-effectiveness analyses and cost-benefit analyses.
The concept of 'costs' is used here in a broad sense, referring to all negative impacts including - in addition to financial costs
- also non-monetary risks and disadvantages, such as time losses, increased health risks caused by climate change, comfort
losses and so on, which are monetized to be able to add them to the monetary costs. Benefits encompass all positive impacts





including avoided monetary costs, avoided health risks, avoided biodiversity losses, avoided material damage, reduced risks caused by climate change, and time and comfort gains.

With a cost-effectiveness analysis (CEA) the net costs (costs plus monetized disadvantages minus monetized benefits) for improving a non-monetary indicator used in an environmental aim with a certain measure are calculated, e.g., the costs of

reducing the emission of 1 t of $CO_{2,eq}$. The lower the unit costs, the higher the cost-effectiveness or efficiency of a policy or measure. The CEA is mostly used for assessing the costs associated with climatically active species, as the effects are global. The situation is different for air pollution, where the damage caused by emitting 1 tonne of a pollutant varies widely depending on time and place of the emission.

Cost-benefit analysis (CBA) is a more general methodology. In a CBA, the benefits, i.e., the avoided damage and risks due to

an air pollution control measure or bundle of measures are quantified and monetized. Then, costs including the monetized negative impacts of the measures are estimated. If the net present value of benefits minus costs is positive, benefits outweigh the costs, thus the measure is beneficial for society, i.e., it increases welfare. Dividing the benefits minus the nonmonetary costs by the monetary costs of a specific measure will result in the net benefit per € spent, which can be used for ranking policies and measures. For performing mathematical operations like summing or dividing costs and benefits, they have first to

be quantified and then converted into a common unit, for which a monetary unit, e.g., Euros, is usually chosen. Integrated assessment means that – as far as possible – all relevant aspects (disadvantages, benefits) should be considered, i.e., all aspects that might have a non-negligible influence.

When setting up air pollution control plans, it is essential to also consider the effect of these plans on greenhouse gas emissions. Air pollution control measures usually lead to a decrease, but sometimes also to an increase in GHG emissions. And vice versa,

most measures for GHG reduction influence, in fact in most cases reduce the emissions of air pollutants. Thus, an optimized combined air pollution control and climate protection plan is necessary to avoid contradictions and inconsistencies.

Looking at the current praxis in the EU countries, still separate plans are made for air pollution control and climate protection. Air pollution control plans currently more and more estimate the reduction (or sometimes increase) of GHG emissions, but they do not assess these reductions e.g., by monetizing them, and thus they cannot be accounted for in a cost-benefit analysis.

In the assessment of the National Energy and Climate Plans the EC states: 'Despite some efforts made, there continues to be insufficient reporting of the projected impacts of the planned policies and measures on the emissions of air pollutants by Member States in their final plans. Only 13 Member States provided a sufficient level of detail and/or improved analysis of the air impacts compared to the draft plans. The final plans provide insufficient analysis of potential trade-offs between air and climate/energy objectives (mostly related to increasing amounts of bioenergy).' (EC, 2020).

So, in an integrated assessment, the assessment of air pollution control measures should always take into account the impact of changes in greenhouse gas emissions. Correspondingly, climate protection plans should take changes in air pollution into account (Navrud and Ready, 2002, 2007, Friedrich et al., 2020). In the following, advancements in the quantification and monetizing of avoided impacts from reducing emissions from air pollutants and greenhouse gases are described.





## 8.2 Current status and challenges

### 8.2.1 *Estimation and monetization of impacts from air pollution*

Integrated assessments, that include as a relevant element the assessment of air pollution, encompass many areas, especially the assessment of energy and transport technologies and of policies for air pollution control and climate protection. The development of such integrated assessments started in the early 1990s with a series of EU research projects, which have been called 'ExternE-external costs of energy'. A summarized description of the developed methodology can be found in Bickel and Friedrich (2005); further descriptions and project results are addressed in (ExternE, 2012). The framework for integrated environmental assessments has been further consolidated and developed within the EU research projects INTARESE and HEIMTSA: the description of the advanced methodology is published as online guidance system and toolbox in IEHIAS (2014). The processes of an integrated assessment are shown in Figure 18 (Briggs, 2008; IEHIAS, 2014), where important elements like issue framing, scenario construction, provision of data and models, uncertainty estimation and stakeholder consultation are addressed. In the beginning of an assessment, the relevant air pollutants have to be identified, which are those that cause substantial damage. In many cases, primary and secondary particulate matter of different size classes and $NO_2$ will cause the worst damage followed by $O_3$.

The element in the framework, that is representing the assessment of air pollution, i.e. the 'impact pathway approach', is shown in detail in Figure 19. This figure already includes one of the emerging developments described in Chapter 8.3, namely the estimation of individual exposure instead of outdoor concentration. First, scenarios of activities are collected, for instance, the distance driven with a Euro 5 diesel car or the amount of wood used in wood stoves. Multiplying the activity data with the appropriate emission factors will result in emissions. The emission data is input for chemical–transport-models that are used to calculate concentrations on regional, continental or global scales; for Europe the EMEP model (EMEP, 2021) and world-wide the TM5-FASST model (van Dingenen et al., 2018) are often used – cf. Section 5 of this paper.

In the next phase, concentration-response functions derived from epidemiological studies are used to estimate health impacts. For the most relevant pollutants PM10, PM2.5, NO2 and O3, the WHO (2013a) made a meta-analysis of the epidemiological studies available until 2012 and recommended exposure-response-relationships for use in integrated assessments, which are still widely used. Newer epidemiological studies especially investigating the relation between fine particulate air pollution and human mortality have been analysed by Pope et al., 2020, where for cardiopulmonary disease mortality caused by PM2.5 a nonlinear exposure-response function with a decreasing slope is presented. The most important concentration-response functions for impacts of air pollution on human health are described in Sarigiannis and Karakitsios (2018) and HEIMTSA (2011).

Beneath health damage, which is the most important damage category, impacts on ecosystems, especially biodiversity losses, and on materials and crop losses should also be considered. Impacts on ecosystems are usually quantified as pdf, 'potentially disappeared fraction of species' per m² land (Dorber, 2020) and thus as biodiversity losses. A first methodology was developed by Ott et al. (2008), which is still used in some studies. Further approaches, partly adopted from methods developed for LCIA (life cycle impact assessment), were developed later (e.g. Souza et al. 2015, Förster, J. et al. 2019), but because of the





simplifications and uncertain assumptions made none of these approaches reached the same full acceptance as the approaches
for the other damage categories. For material damage and crop loss deposition-response-relationships have been developed in
the ExternE - External Costs of Energy (ExternE, 2012) project series and are described in Bickel and Friedrich (2005); they
are still used.

Finally, the health effects and the other impacts are monetized, which means that they are converted into financial costs; for
the non-monetary part of the impacts results of contingent valuation (willingness to pay) studies are used (as described e.g. in
OECD, 2018). As numerous contingent valuation studies have been made in the past, it is not necessary to carry out a further
willingness-to-pay study, instead results of existing studies which found monetary values for the damage endpoints to be
analysed can be used. Of course, as the contingent valuation studies are usually made at another time, in another area and with
other cultural situations than the planned assessment, the monetary values must be transformed with a methodology called
'benefit transfer' from the original time, place and cultural features to the ones of the assessment (see e.g. Navrud and Ready,
2007). The most important monetary value in the context of air pollution is the value for a statistical life year lost (VLYL)
caused by a premature death at the end of life after a life-long exposure to air pollutants. It is often based on a study of
Desaigues et al. (2011). The result for average EU citizens – transformed to 2020 - is 75.200 (47.000 - 269.450) €$_{2020}$ per
VLYL. A list of monetary values for health endpoints, that are used in most studies, can be found in HEIMTSA (2011).

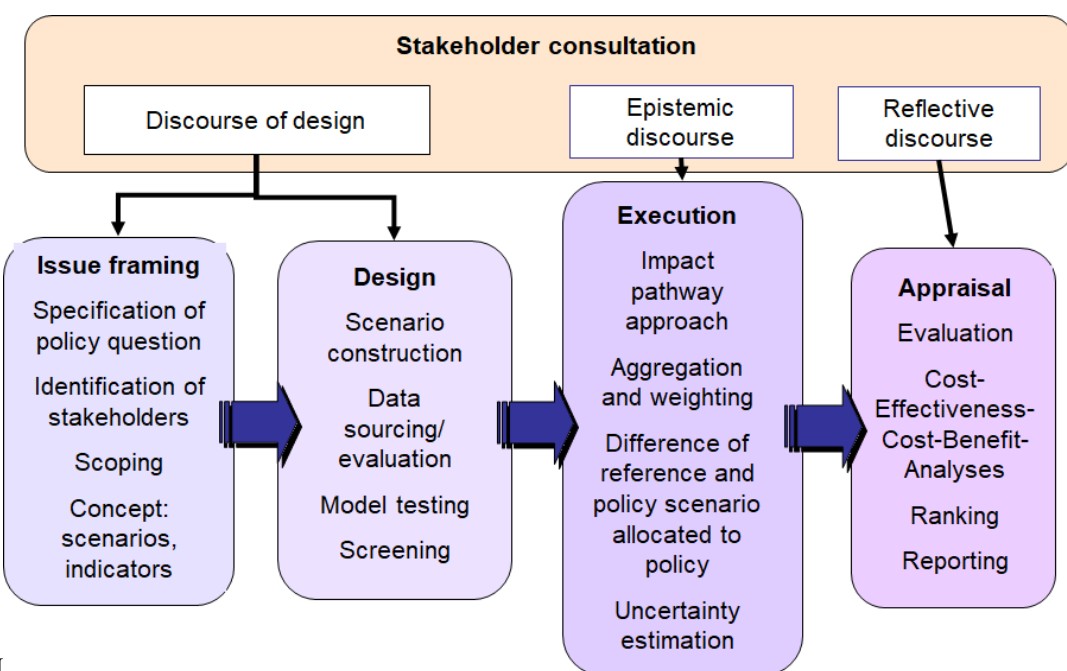

I

Figure 18. Integrated assessment process involving air pollution (IEHIAS, 2014)





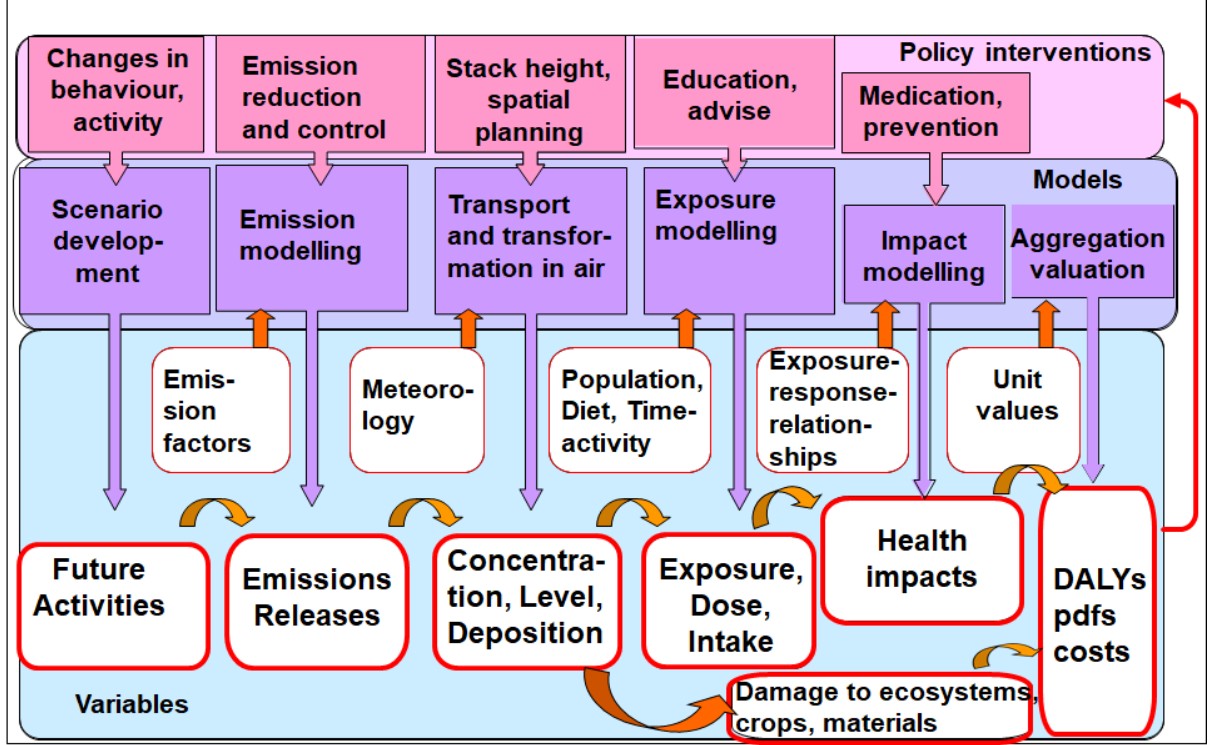

Figure 19. Schematic presentation of the use of models and the flow of data in the enhanced impact pathway approach (Friedrich, 2016).


Based on this principal approach, a growing number of tools have been developed and applied for supporting air quality control for urban, national, and regional to global scales. The tool used for the assessments for DG Environment and for the Convention on Long-range Transboundary Air Pollution of the UN ECE is GAINS (Greenhouse gas - Air pollution Interactions and Synergies) developed by IIASA (Amann et al., 2017 and Amann, 2018).

A specific development in GAINS is the use of source-receptor matrices as a proxy for using an atmospheric model. A limitation of chemical transport models has been the substantial computational requirements for running the models for estimating hourly concentration values caused by an emission scenario for an entire year. To be able to simulate many scenarios within a short time, results of certain runs with the complex atmospheric model EMEP (EMEP, 2021) were transformed into source-receptor matrices, which provided information of the relationship between a change of emissions in a country and the

change of the concentration in grid cells of a European grid. However, because of the relatively large size of the grid cells for European-wide models, concentrations in cities were underestimated, thus an 'urban increment' was introduced for cities (Concawe, 2004, Torras et al. 2013, Torras 2012). Thunis (2018), however, points out that this approach has certain weaknesses. Thus, newer approaches use nested modelling with regional atmospheric models using varying grid sizes (e.g., Brandt 2012) or modelling of typical days instead of a whole year with a finer grid (Bartzis et al., 2020, Sakellaris, 2021). The





ECOSENSE model uses a similar method as GAINS, however, distinguishing between parts of larger countries and emission heights, furthermore a monetary assessment of greenhouse gas emissions is made (ExternE, 2012).

As a major application of the GAINS tool, the European Commission, DG Environment regularly assesses its directives for air pollution control. A well-known example is the impact assessment carried out for assessing the Thematic Strategy on Air Pollution and the Directive on "Ambient Air Quality and Cleaner Air for Europe" (EC, 2005). It was shown that the monetized

benefits of implementing the thematic strategy for air pollution control are much higher than the costs. In the most recent assessment, DG Environment assessed the costs and benefits of the so-called NAPCPs, the national air pollution control programmes, which the member states had to provide by 2019 to show, how they plan to comply with the emission reduction commitments of the National Emission Reduction Commitments Directive (NEC Directive). The benefits considered were the monetized reduced health and environmental impacts caused by the requested air pollution control measures. The results show

that the health benefits alone with 8-42 billion €/a are much larger than the costs of the considered measures with 1,4 billion €/a (EC, 2021) and that further emission reductions might also be efficient. The UN ECE (UN Economic Commission for Europe) has launched eight so-called protocols guided by the Convention on Long-range Transboundary Air Pollution, which require the member states to provide information on air pollution in their countries and to take actions to improve it (UNECE, 2020). The latest protocol entering into force was the revised Protocol to Abate Acidification, Eutrophication and Ground-

level Ozone, as amended on 4 May 2012. To prepare for these protocols, the effects of air pollution on ecosystems, health, crops, and materials have been assessed with the same methods as used by the EC, i.e., using the GAINS model.

The OECD recommends carrying out cost-benefit analyses with the impact pathway methodology (OECD, 2018). Similarly, national authorities, e.g., the German Federal Environmental Agency has proposed to use the methodology for the assessment of environmental policies and infrastructure projects (Matthey and Bünger, 2019). In Denmark, the method has been used in

the EVA system (Economic Valuation of Air pollution, Brandt et al., 2013) to estimate the external costs related to air pollution, as part of the national air quality monitoring programme (Ellermann et al., 2020). The same system has been used to assess the impact from different emission sectors and countries within the Nordic area, by using a CTM model with a tagging method (Im et al. 2019). Kukkonen et al. (2020c) developed an integrated assessment tool based on the impact pathway principle that can be used for evaluating the public health costs. The model was applied for evaluating the concentrations of fine particulate

matter ($PM_{2.5}$) in ambient air and the associated public health costs of domestic $PM_{2.5}$ emissions in Finland. Several further integrated assessment models have been described in Thunis et al. (2016). Not only in Europe, but also in the USA integrated assessment of air pollution is an issue. Keiser and Muller (2017) provide an overview of integrated assessment models for air and water in the US and hint at the intersections between air and water pollution.

Several studies are using the impact pathway approach from Fig. 8.2 for estimating health impacts and aggregate them to

DALYS (disability adjusted life years), but without monetizing the impacts, i.e.. they calculate the burden of disease or the overall health impacts stemming from air pollution. The WHO has estimated the burden of disease from different causes, including air pollution, in the Global Burden of Disease Study (GBDS, 2020). The European Environmental Agency regularly estimates the health impacts from air pollution in Europe and found 4.381.000 life years lost attributable to the emissions of





PM$_{2.5}$ in 2018 in the EU28 (EEA, 2020b). Hänninen and Knol (2011) analysed the burden of disease of nine environmental

stressors, including particulate matter, for Europe. Lehtomäki et al. (2018) quantified the health impacts of particles, ozone, and nitrogen dioxide in Finland and found a burden of 34,800 DALYs per year, with fine particles being the main contributor (74%). Recent studies are also including future projections of emissions and climate. Huang (2018) assessed and monetized the health impacts of air pollution in China for 2010 and for several scenarios until 2030. Likewise, Tarin-Carrasco et al. (2021) projected the number of premature deaths in Europe towards 2050 and found that a shift to renewable energy sources (to a

share of 80%) is effective in reducing negative health impacts.

In the following, we address recent improvements in the methodology of integrated assessment with a focus on air pollution control. A milestone was the publication of concentration-response functions for NO$_2$ by the WHO (2013a), following this more and more studies calculated health impacts not only from exposure to PM$_{10}$, PM$_{2.5}$ and ozone, but also to NO$_2$ (e.g., Balogun et al., 2020, Siddika et al., 2019, 2020),

Ideally, human health risks should be evaluated based on exposures instead of ambient concentrations (cf. Section 7). Until now, measured or modelled ambient (outdoor) air concentrations are input to the concentration-response functions used to estimate health risks. However, it is obvious that people are affected by the pollutants that they inhale, and that is decisive for the health impact. So, a better indicator for estimating health impacts than the outside background concentration is exposure, which is the concentration of pollutants in the inhaled air averaged over a certain time interval. Only recently, in the EU

projects HEALS and ICARUS, methodologies have been developed to estimate personal exposure, i.e. the concentration in the inhaled air averaged over a year or a number of years as basis for estimating health impacts from air pollutants. Furthermore, the time span used in the exposure response relationships commonly ranges from hourly to annual mean concentration values. By far the most important health effects are chronic effects. Although the indicator used to estimate chronic impacts is the annual mean concentrations, chronic diseases develop over several years, or even during the whole lifetime. This is the reason

why the EC regulates a three year 'average exposure index' of PM$_{2.5}$ in the Air Quality Directive. But the relevant time period for the exposure might be larger than three years. Thus, exposure over a lifetime is important for estimating risks to develop chronic diseases and premature deaths, which are the most important health impacts. The methods for evaluating lifetime exposure have been addressed in Chapter 7.2.3 (cf. Li and Friedrich, 2019, Li et al., 2019b, Li et al, 2019c).

Thus, as a major improvement of the impact pathway approach, the exposure to pollutants should be used as an indicator for

health impacts, instead of the exposure estimated from outdoor air concentrations at permanent locations. However, epidemiological studies, that directly relate health impacts to exposures to air pollutants are not yet available. Instead, the existing concentration-response functions are transformed into exposure-response functions by calculating the increase of the exposure (e.g. x µg/m³) caused by the increase of 1 µg/m³ increase in the outdoor concentration. Dividing the concentration-response relationship by x will then convert it into an exposure-response relationship (Li, 2020). Of course, it would be better

to use results of epidemiological studies, that directly relate exposure data with health effects. Thus, such studies should be urgently conducted.





Clearly, indoor pollution sources also influence exposure. It is therefore important to assess possibilities to reduce the contributions of indoor sources to exposure. These might include, e.g., raising awareness of the dangers of smoking at home indoors, the development of more effective kitchen hoods and promoting their use, ban of incense sticks, and mandatory use

of inserts in open fireplaces.

Secondly, a reduction of exposure is also possible by increasing the air exchange rate with ventilation or by filtering the indoor air. E.g., if old windows are replaced by new ones, the use of mechanical ventilation with heat recovery might be recommended or even made mandatory. Also, the enhancement of HEPA filters and their use in vacuum cleaners will help as well as using air purifiers/filters. These systems will also be helpful to reduce the indoor transmission of SARS-CoV-2.

Furthermore, there is growing evidence that $PM_{10}$ and $PM_{2.5}$ concentrations in underground trains and in metro stations can be much higher than the concentration in street canyons with dense traffic (e.g. Loxham and Nieuwenhuijsen 2019, Mao et al. 2019, Smith et al. 2020). Using ventilation systems with filters might improve this situation.

*8.2.2 Monetization of impacts of greenhouse gas emissions*

As explained above, air pollution control strategies usually influence and, in most cases, reduce emissions of greenhouse gases

(GHGs). Thus, in an integrated assessment, both the reduction of air pollution and of greenhouse gas emissions should be quantitatively assessed. In practice, however, many national air pollution control strategies do not take changes in GHGs into account in the assessment, instead the national authorities develop separate climate protection plans. Similarly, although DG Environment estimates the changes in GHG emissions in their assessment of air pollution control strategies, the changes are not assessed resp. monetized.

An exception is the UK, where estimations of the 'social costs of carbon' are used in assessments (Watkiss et al., 2008, DBEIS, 2019). The UK government currently recommends using a carbon price of £69/tCO2eq in 2020 rising to £355/tCO2eq in 2075-2078 at 2018 prices.

How can the benefits of a reduction of greenhouse gas emissions be monetized? A possibility is to use the same approach as with air pollution, i.e., estimate the marginal damage costs (i.e. the monetized damages and disadvantages) of emitting one

additional ton of $CO_2$. These marginal costs would then be internalised e.g., as a tax per t of $CO_2$ emitted to allow the market to create optimal solutions (Baumol, 1972). Thus, first scenarios of greenhouse gas (GHG) emissions would be set up, then concentrations of GHGs in the atmosphere would be calculated, next changes of the climate followed by the estimation of changes of risks and damages would be calculated. However, this does not lead to useful results. Uncertainties are too high and assumptions of economic parameters like the discount rate or the use of equity weighting influence the result considerably,

so that the range of results encompasses several orders of magnitude. Furthermore, the precautionary principle tells us that we should avoid possible impacts, even if they cannot (yet) be quantified and thus not be included in the quantitative estimation of impacts.

An alternative approach to estimating marginal damage costs is to use marginal abatement costs. A basic law of environmental

economics is that for pollution control a pareto-optimal state should be achieved, where marginal damage costs (MDC) are





equal to marginal abatement costs (MAC). Thus, if MAC at the pareto-optimal state are known, they could be used instead of the MDC. However, the pareto-optimal state is not known if MDC are not known. But one could use an environmental aim, that is universally agreed by society, and assume that they represent the optimal solution in the view of society and then estimate the MAC to reach this aim, which is then used for the assessment. This approach was first proposed by Baumol and
Oates (1971).

For assessing GHG emissions, especially the aim of the so-called Paris agreement, which was agreed on at the 2015 United Nations Climate Change Conference, COP 21 in Paris by a large number of countries. The most important aim was to keep a global temperature rise this century well below 2 degrees Celsius above pre-industrial levels and to pursue efforts to limit the temperature increase even further to 1.5 degrees Celsius. This objective could be used as the basis for generating MAC.
Bachmann (2020) has carried out a literature research of as well MDCs as MACs for GHG emissions. Based on this review, MACs calculated by a meta-analysis of Kuik et al. (2009) are used here as basis for the calculation of marginal abatement costs for reaching the above aim, resulting in 286 (162-503) €$_{2019}$ per t of $CO_{2,eq}$ in 2050. However, we propose starting with the most efficient measures now and gradually increasing the specific costs, until they reach the costs mentioned above in 2050. If future innovations lead to a reduction of the avoidance costs, the costs of carbon can be adjusted accordingly. With a
real discount rate of 3%/a social costs of $CO_{2eq}$ to be used in 2020 would be 118 (67-207) €$_{2020}$ per t of CO2, eq.

**8.2.3** *Effect of integrating air pollution control and climate protection*

In most cases, especially if a substitution of fossil fuels with carbon-free energy carriers or a reduction of energy demand is foreseen, a reduction of emissions of as well greenhouse gases as of air pollutants is foreseen; thus, taken both air pollution control and climate protection into account will considerably improve the efficiency of such measures.
An example, that the choice and ranking of measures for combined air pollution control and climate protection is different from the ranking in separate plans is shown in Figure 20. In the frame of the EU TRANSPHORM project, 24 measures to reduce air pollution and climate change caused by transport in the EU have been assessed with an integrated assessment. Fig. 8.3 shows the 10 most effective measures for both avoided health impacts as for reduced climate change, where both benefits are converted into monetary units and combined (Friedrich, 2016). As can be seen, measures with benefits in both air pollution
control and climate protection improve their rank compared to the separate rankings for these damage categories. The most effective measure is the travel with trains instead of air planes for routes of less than 500km.





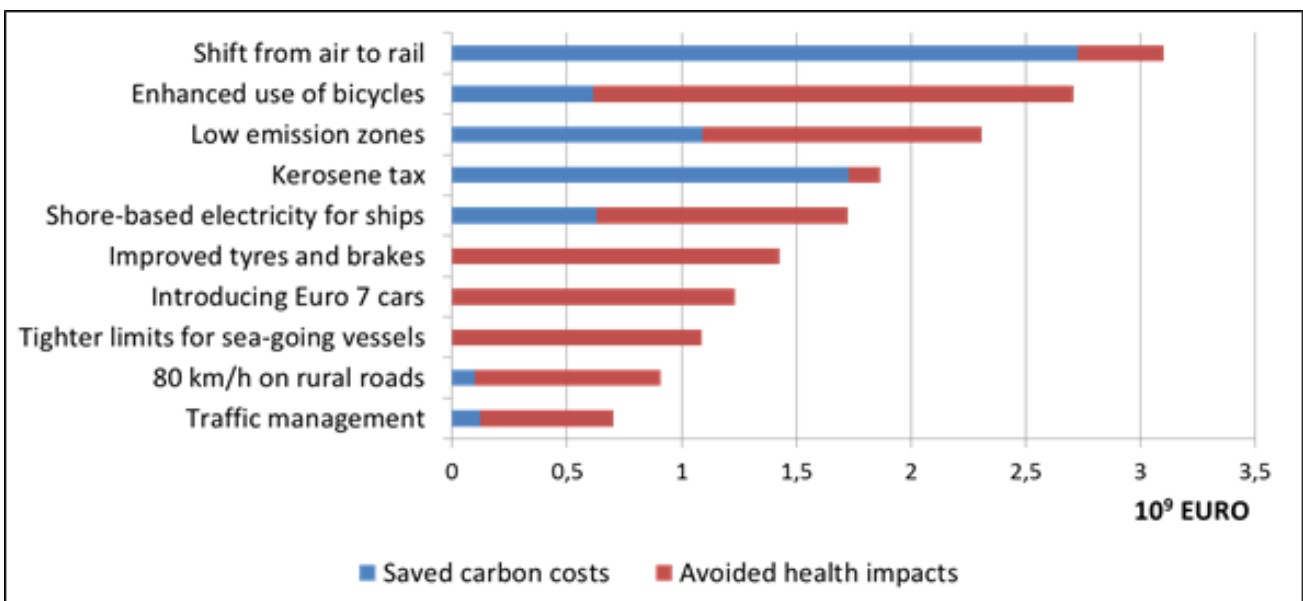

Fig. 20. ranking of measures in transport in the EU regarding their effectiveness for reducing air pollution and climate change; the effectiveness is expressed in bn €$_{2010}$/a (Friedrich, 2016).


With another example, Markandya et al (2020) demonstrate, that especially for developing and emerging countries the costs for meeting the aims of the Paris agreement will be outweighed by the benefits that are achieved by avoiding health impacts from air pollution, so that the climate protection comes without net costs. This is due to the fact, that in developing countries the use of fossil fuels is less accompanied by the use of emission reduction technologies (filters), so replacing fossil fuels by

electricity from wind or solar energy or saving energy will result in a much higher reduction in air pollution than doing the same in OECD countries. For Europe, the effects of integrating the damage costs of air pollution into the optimization of energy scenarios have been analysed by Korkmaz et al. (2020) and Schmid et al. (2019). Two effects are important: firstly, biomass burning is significantly reduced, as firing biomass is climate friendly, but leads to air pollution. Secondly, the marginal avoidance costs per t of avoided carbon are reduced, especially for the period 2020-2035. The reason is, that in this period

more efficient measures like the replacement of oil and coal with electricity from carbon-free energy carriers (except biomass) and measures for energy savings will reduce emissions of air pollutions, while later more expensive measures like producing and using power-to-X as fuels will have a lower effect on air pollution reduction.

In most cases, integrated assessment improves the efficiency of measures for environmental and climate protection. In the following an example is shown, where an efficient climate protection measure gets inefficient, if air pollution is included in

the assessment. This example is the use of small wood firings in cities. Wood firings are climate friendly but emit lots of fine particles and NOx. Huang et al. (2016) and Friedrich and Vogt (2020) show that for wood firings that are operated in cities, the damage of more health impacts outweighs the benefit of less greenhouse gas emissions.





Figure 21 shows the social costs per year, these are the monetary costs, the monetized impacts of climate change and the
monetized health impacts caused by air pollution for different heating techniques, that are operated in a single-family house in
the centre of the city of Stuttgart. The social costs are highest for wood and pellet combustion caused by their high air pollution
costs, although the climate change costs of the wood combustion are very low. The social costs marked with Sota- state of the
art - are calculated for new technologies fulfilling the currently valid strict regulations for small firings in Germany (1.
BImSchV, 2020). Older stoves have emissions and thus impacts that are much larger than those shown. However, even if we
further enhance the emission reduction and especially equip the wood and pellet combustion with a particulate filter - these
are represented by the columns marked with 'Iapc - improved air pollution control'' -, the ranking is not changed. The reason
are the high NOx emissions of wood combustion. These results suggest that firstly wood combustion should be equipped with
a particulate filter and secondly, that in cities with high NOx concentrations a ban of small wood combustion might be
considered, unless they are also equipped with SCR filters.


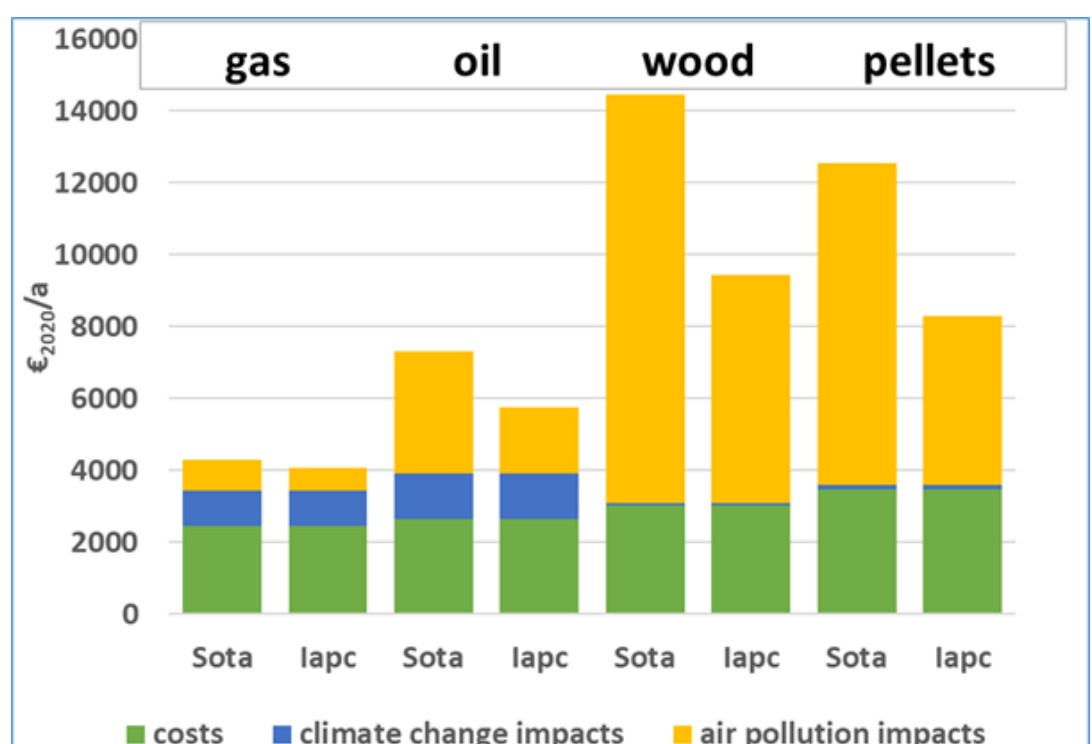

Figure 21. Social costs of different heaters for a single-family house in Stuttgart, 'Sota = state of the art' are currently available
new technologies, 'Iapc = improved air pollution control' are future technologies with enhanced air pollution control,
especially the use of particulate filters for wood and pellet combustion; Huang at al. (2016), Friedrich and Vogt (2020).






### 8.3 Emerging challenges

### 8.3.1 Challenges in improving the methodology for integrated assessments

Estimations of damage costs caused by air pollution and climate change still show large uncertainties. Li (2020) reports a 95%
confidence interval of $3.5*10^{11}$ to $2.4*10^{12}$ € for the damage costs caused by the exposure to $PM_{2.5}$ and $NO_2$ for one year (2015) for the adult EU28+2 population. Kuik et al. (2009) report an uncertainty range for the marginal avoidance costs to reach the '2° aim' of 162-503 €$_{2020}$ per t of $CO_{2,eq}$ in 2050. In addition, systematic errors might occur, for instance still unknown exposure-response relationships. Thus, methodological improvements are necessary.

In principle, all model steps and related input data shown in Table 5 would need improvement. Most of the improvements
necessary for models and the data shown in Table 5 have already been addressed in the previous sections. Challenges for improving the estimation of emissions of indoor and outdoor sources are described in Chapter 3.3. Improvements in atmospheric modelling are addressed in Chapter 5.3. Exposure modelling is a relatively new field, so a lot of gaps have to be filled (see sections 7.3.3). Further epidemiological studies, especially for analysing the health impacts of specific PM species and PM size classes are urgently needed to concretize the and contingent valuation studies are needed to improve the
methodology. The challenges for these topics are addressed in the relevant sections above and will thus not be repeated here. However, two further methodological improvements have not been mentioned and are thus described in the following.

When assessing a policy measure for the reduction of air pollution, the first step is to estimate the reduction of emissions caused by the policy measure. Measures can be roughly classified in technical measures, that improve emission factors (e.g., by demanding filters), and non-technical measures that change the behaviour or choices of emission source operators (e.g., by
increasing prices of polluting goods). Especially if non-technical measures are chosen, e.g., the increase in the price for a good that is less environmentally friendly, the identification of the reaction of the operators of the emission sources is not straightforward. Do they keep using the good, although it is more expensive, does he substitute the good or does he renounce the utility of the good by using neither the good nor substitutes anymore? For energy saving measures, it is well-known that after implementing such a measure, the users do not save the full expected energy amount, but instead increase their comfort,
for instance by increasing the room temperature. This is known as rebound effect. The traditional way to deal with behavioural changes is using empirically found elasticity factors. For the transport sector, where most of the applications are made, Schieberle (2019) compiled a literature search for elasticities in the transport sector and demonstrated their use in integrated assessments. However, as a further development recently first attempts to use agent-based modelling have been made to estimate the behavioural changes of people confronted with policy measures (Chapizanis et al., 2021).

With regard to the marginal costs of $CO_2$ reduction used in the assessments, further investigations taking into account emerging innovations are necessary. Furthermore, the stated estimates are quite high, so that the question arises, whether the values and thus the Paris aim gain worldwide social acceptance. More emphasis might be laid on research to develop measures for more efficient climate protection as well as measures to remove GHGs from the atmosphere and also to develop adaptation measures.





### 8.3.2 Challenges for the reduction of ambient air pollution


In recent years, regulations have been implemented that will decrease emissions in two important sectors considerably:

(i) for ships, the IMO (International Maritime Organisation) adopted a revised annex VI to the international Convention for the Prevention of Pollution from Ships, now known universally as MARPOL, which reduces the global sulphur limit from 3.50% to 0.50%, effective from 1 January 2020 (IMO, 2019). This will drastically reduce the $SO_2$ emissions

around Europe outside of the sulphur emission control areas Baltic Sea, North Sea and English Channel. Furthermore, the IMO has adopted a strategy to reduce greenhouse gas emissions by at least 40% by 2030, pursuing efforts towards 70% by 2050, compared to 2008. A revised, more ambitious plan is currently being discussed. Geels et al. (2021) assess the effects of these new regulations by generating several future emission scenarios and assessing their impact on air pollution and health in Northern Europe.

(ii) diesel cars now must comply with the Euro 6d norm, which drastically will reduce the real driving emissions of $NO_x$ on streets and roads. In addition, electric vehicles are now promoted and subsidized in many EU countries.

In the context of a revision of the EU rules on air quality announced in the European Green Deal, the European Commission is expected to strengthen provisions on monitoring, modelling and air quality plans to help local authorities achieve cleaner air, notably proposing to revise air quality standards to align them more closely with the World Health Organization

recommendations (which will be updated in 2021).

The European Commission is also expected to introduce a new Euro 7 norm for passenger cars in 2025. The Industrial Emissions Directive demands permanent reviews of the EU Best Available Techniques reference documents (BREFs), resulting in decreasing emissions from large industrial emitters. The EU has decided to reduce EU's greenhouse gas emissions by at least 55% compared to 1990. Furthermore, national reduction plans for GHG lead to a further reduction of the combustion

of fossil fuels. Thus, emissions of air pollutants from combustion processes will significantly decrease with one exception: wood and pellet firings < 500kW$_{th}$. Hence, regarding combustion, the main challenge is the development of further $PM_{2.5}$ and NOx reduction measures for small wood firings. Similar trends will be observed in a number of other countries, e.g., the USA wants to reduce their greenhouse gas emissions from 2005 to 2030 by 50% and China wants to reach carbon neutrality by 2060.

As emissions of particulates from combustion decrease, diffuse emissions, e.g., from abrasion processes, bulk handling or demolition of buildings and from evaporation of volatile organic compounds get more and more dominant. So more emphasis should be put on the determination and reduction of these emissions. Especially the processes leading to diffuse emissions are not well known. In transport, emissions from tyre and brake wear and road abrasion are heavily depending on driving habits, speed, weather conditions and especially the traffic situation and lay-out of the road network. However, emission factors for

diffuse emissions are still largely expressed in g per vehicle-km, not taking situations where braking is necessary e.g., because of traffic jams or crossroads into account. Furthermore, reduction measures like the development of tyres and brakes with longer durability should be considered and assessed.



A key challenge for reducing secondary particulates, especially ammonium nitrates, is a further reduction of $NH_3$ emissions from agriculture. Certain national reduction commitments for the EU countries from 2005 until 2030 are regulated by the
National Emission Reduction Commitments Directive (NEC Directive) of the EU, but further reductions might be necessary.

### 8.3.3 Challenges for the reduction of indoor air pollution

A more precise understanding of personal exposure to air pollution and the use of exposure-response relationships (instead of relationships linking outdoor concentration with responses) will potentially change the focus of air pollution control. As people
are most of the time indoors, now the reduction of indoor pollution is becoming important. Of course, reducing ambient concentration will also reduce indoor pollution, as pollutants penetrate from outside into the houses. However, around 46% of the total exposure with $PM_{2.5}$ for an average EU citizen stem from indoor sources; for $NO_2$ about 25% are caused by indoor sources (Li and Friedrich, 2019). Thus, indoor sources cannot be neglected. The reduction of exposure to emissions from passive smoking, frying and baking in the kitchen, using open fireplaces and older wood stoves and incense sticks and candles
is especially important. Indoor concentrations can be reduced by reducing the emission factor of the source, changing behaviour, when using the source, by banning the use of a source, by increasing the air exchange rate with ventilation and by using air filters.

### 9. Discussion, synthesis and recommendations

This review has covered a larger number of research areas and identified not only the current status but also the emerging research needs. There are of course cross cutting needs that are a prerequisite to further air quality research and develop more robust strategies for reducing the impact of air pollution on health. The following section discusses some of the key areas and synthesises these in the form of recommendations for further research.

**9.1. Connecting emissions and exposure to air pollution**

There is a progressively important need to move from static, annual inventories to those that are dynamic in terms of activity patterns and of higher temporal resolution. This is driven partly by the need for activity dependent exposure modelling and because there is an increasing availability of online observations from sensors to arrive at a better spatial and temporal resolution of emission rates and factors. Clearly community efforts are necessary for identifying and reducing uncertainties in
emissions that have a large impact on the resulting air quality and exposure predictions including, e.g., benefiting from source apportionment methods.

One gap is the evaluation of agricultural emissions, which are still poorly understood, and improvements will support both air quality and climate change assessment, leading to co-benefits. While considerable effort has been devoted to estimating NOx emissions, there are still uncertainties in the estimation of VOC emissions. These uncertainties have direct implications, when
quantifying changes in ozone levels and contributions from secondary organic aerosols to regional and global scales. One prominent example of such uncertainties is the estimation of VOC profiles in terms of the chemical species and their





evaporation rates, including in particular those from shipping activities in the vicinity of ports, as a shift has occurred to both low sulphur and carbon neutral or non-carbon fuels.

Similarly, as exhaust emissions reduce with the increase of electric vehicles, the assessment of the consequences of airborne non-exhaust emissions is becoming more and more important. However, this needs to be examined in the context of policy driven tighter controls on petrol and diesel vehicles. Emission factors for ultrafine particles are also uncertain; these are also spatially and temporally highly variable, which reduces the reliability of particle number predictions necessary for estimating exposure of people (Kukkonen et al., 2016a).

Exposure connects emissions to concentrations and their impact on health. As exposure to a particular air pollutant is determined by all sources of that air pollutant, both indoor and outdoor sources are important. Indoor sources are considerable in number and variety from tobacco smoking, cooking and heating fuels, indoor furniture, body care to cleaning products and perfumes. Not only are emission factor data for these sources needed, stricter regulations are necessary for indoor sources (e.g. indoor cleaning products and wood burning for residential heating).

## 9.2 Extending observations for air quality research

Our review has highlighted the urgent need to strengthen the integration of observations from different platforms, including from reference instruments, mobile and networked lower cost sensors as well as other data sources, such as satellite instruments and other forms of remote sensing. In addition to providing greater spatial extent and fine-scale resolution of observations in urban and other areas, these integrated datasets can form the basis of inputs for dynamic data assimilation. Data assimilation can also be performed using machine learning and/or artificial intelligence approaches. These developments can improve the accuracy of chemistry-transport models, including air quality forecasts.

Additional requirements for lower cost sensors are (i) improving their reliability for both the gaseous and particulate matter measurements (including especially VOC's), (ii) extending the measured size range of particulate matter up to ultra-fine particles and (iii) including their scope to measure also bioaerosols, such as allergenic pollen species and fungi. Integrating these sensors into existing infrastructures, such as permanent air quality measurement networks, traffic counting sites and indoor monitoring, would provide a richer dataset for air quality and exposure research. Further effort is required to determine the health-relevant PM information, including especially the chemical composition of PM.

As citizen science and crowdsourcing increases, its uses in air quality research needs to be more clearly defined. It could potentially provide near real time air pollution information as well as information to be used for personal health protection and lifestyle decisions. Most challenging for this objective is data quality characterisation and acceptance of these new data provisioning tools, which do not easily allow an analytical quality assurance and control.

## 9.3 Bridging scales and processes with integrated air pollution modelling

Continuing developments in fine, urban and regional modelling has elevated *scale interactions* as a key area of interest. As highlighted above, research is needed to develop new approaches to connect processes operating on different scales. Baklanov



et al (2014) has reviewed online approaches that include coupling of meteorology and chemistry within an Eulerian nested framework but on the whole air pollution applications are limited to different modelling systems including Eulerian for regional, Gaussian for urban and street and LES and advanced CFD-RANS for even finer scales. Challenges remain on how to best integrate the fast-emerging machine learning statistical tools and how parameterizations and computational approaches

have to be adapted. These scale interactions are of critical importance when examining the impact of air pollution in cities which are subject to heterogeneous distribution of emissions and rapidly changing dispersion gradients of concentrations. New modelling approaches will enable multiple air quality hazards that affect cities to be examined within a consistent multi-scale framework for air quality prediction and forecasting, local air quality management, quantifying the impact of episodic high air pollution events involving LRT and even meteorological and climate hazards as cities prepare for the future.

One major development in this vain is that of Earth system modelling (ESM) approaches which in the past have been focussed on global scales but has the potential of higher resolution applications (e.g. WWRP 2015). Within Earth system models, there is potential for integration of observations (e.g. through data assimilation of soil moisture and surface fluxes of short-lived pollutants and greenhouse gases). These developments are to some degree being aided by the rapidly evolving area of parallel computer systems. While the representation of urban features and processes within ESMs still require further effort, these

models have the potential to include dynamical and chemical interactions on a much wider scale than is possible with traditional approaches (e.g. mesoscale circulations, urban heat island circulation, sea-breeze and mountain-valley circulations, floods, heat waves, wildfires, air quality issues, and other extreme weather events).

As primary air pollution emissions are reducing the role of secondary air pollutants (e.g., ozone, and secondary organic and inorganic aerosols) in urban environments require more research, Here coupled systems and potential ESMs in the future will

have a key role based on two-way interaction chemistry-meteorology models combining the effects of urban, sub-urban and rural pollutant emissions with dynamics. This is especially true in a changing climate scenario.

Cities are routinely facing multiple hazards in addition to high levels of air pollution including storm surges, flooding, heat waves, and a changing climate. Moving towards integrated urban systems and services poses research challenges but is viewed as essential to meet sustainable and environmentally smart cities development goals, e.g. SDG11: Sustainable Cities and

Communities (Baklanov et al., 2018b; Grimmond et al., 2020). More integrated assessment of risk to urban areas necessitates observation and modelling that brings together data from hydrometeorological, soil, hydrology, vegetation and air quality communities including sophisticated and responsive early warning and forecast capabilities for city and regional administrations.

**9.4 Improving air quality for better health**

As air quality science continues to develop, the need to improve our understanding of PM properties and resulting health impacts remains a priority. In particular the areas that stand out are the need to better quantify particle number concentrations (PNC), particle size distributions (PSD) and the chemical composition of PM especially in urban areas where population density is higher. An ongoing challenge for the science community is to investigate which of the PM properties or measures





optimally describe the resulting health impacts. To aid research, a denser measurement network on advanced PM properties is

needed for quantifying chemical and physical characteristics of PM in cities and regionally. Another important requirement is

is the availability of improved higher resolution emission inventories of PM components and for different sizes (see Section

9.1). To support epidemiological studies, comprehensive long-term datasets are needed including both (i) multi-decadal

evaluations of air quality, meteorology and exposure, and (ii) information on a range of health impacts.


### 9.5 Challenges of global pandemics

In addition to the multiple hazards facing cities mentioned in Section 9.3, the COVID-19 pandemic has starkly demonstrated

how society can be dramatically affected across the world. Studies are indicating dramatic impact on air quality due to the

lockdown as well as possible connections between air pollutants such as aerosols in spreading the SARS-CoV-2 virus (e.g.,

Gkatzelis et al., 2021). To fully assess the interactions of viruses and air pollutants, studies need to consider both indoor and

outdoor transmission as well as meteorological and climatological influences. A recent preliminary review WMO (2021) has

concluded that there are mixed indications of links between meteorology and air quality with COVID-19, and more thorough

studies are needed to ascertain the direct and indirect effects. Given the complexity of the topic, cross- and interdisciplinary

studies would be needed, including a collaboration of microbiologists, epidemiologists, health professionals, and atmospheric

and indoor pollution air scientists.

### 9.6 Integrating policy responses for air quality, climate and health

Most control policies and measures targeted at air pollution will also change GHG emissions, which implies that taking them

both into account in integrated assessments will in most cases provide considerable co-benefits. There are cases, for example

in the case of biomass burning, which will increase air pollution emissions and hence additional abatement measures (e.g.,

cleaning systems) will be required. On the whole, however, integrating climate change and air pollution policies where possible

have the potential of making the integrated policy more efficient than separate policies for improving air quality and limiting

the impact of climate change. Thus, integrated environmental policies based on assessing simultaneously reductions of impacts

on health, the environment and materials caused by air pollution control and reductions of impacts of climate change casued

by measures for climate protection should be implemented. The assessment should be made following the impact pathway

approach described in Section 8.2. The impact pathway approach uses the methods and data from all the sections of this paper,

i.e. emission modelling, atmospheric modelling, exposure modelling and health impact modelling, thus to address the

challenges described in this paper would help to reduce uncertainties and improve efficiency in the scientific recommendations

for setting up inetgrated environmental policy plans. Within an integrated air pollution control and climate protection

assessment, a particularly important new development would be to use the individual exposure (the concentration of a pollutant

where it is inhaled by an individual averaged over a year) instead of some outdoor concentration asindicator for health impacts,

i.e., as input for the exposure-response relationship. In this case, the indoor concentration of air pollutants and thus indoor



source emission rates and ventilation air exchange rates would be important elements in the assessment along with contributions from outdoor sources when planning air pollution control strategies.


**9.7 Key recommendations**

Below in Table 1 we present a synthesis of key recommendations for scientific research and the importance for air quality policy that have emerged from this review. The table also provides an indication of the confidence in the scientific knowledge in each of the areas, the urgency to complete the science gaps in our knowledge and the importance of each of the listed areas

for supporting policy. It should be noted that our approach provides more of an overview and does not consider the needs of specific areas or of national needs which may differ from the regional status of knowledge. For example, in the case of emission inventories for Europe and North America, there is generally high confidence but that may not be the case in other regions of the world or for specific countries or sub-regions.

Table 1. A synthesis of key recommendations for scientific research and the importance for air quality policy. A three-level scale is used to indicate the current confidence in the scientific knowledge and understanding and a measure of the urgency to fill the science gaps where they exist. Similarly, a three-level scale is used to indicate the importance of the specific issues for policy support.

Scientific confidence – h: high (progress is useful but may not require significant specific research effort); m: medium (some

further research is required), l: low (concerted research effort is required).

Scientific urgency to meet gaps in knowledge – v: very urgent need to fill science gap, u: urgent need to fill scientific gap, w: widely accepted with less urgency to fill the science gap

Importance for policy support - H: high (is highly important for developments of new policies); M: medium (can lead to refinements of current policies), L: low (progress is useful but may not require significant effort in the short or medium term).

| Research theme | **Scientific confidence** in current knowledge and understanding | **Scientific urgency** to meet key gaps in knowledge | **Importance for policy** support e.g. compliance or health |
|---|---|---|---|
| Air pollution emissions | Annual totals in inventories for previous years in Europe and North America (h) and developing regions (l) | Source characterisation and attributions approaches (w), for developing regions (v) Emission totals and temporal profiles for highly intermittent sources (e.g. ammonia from agriculture, PM from wood burning) for all regions (v) | More information about particles including particle number from all types of abrasion (H-M) |
| | Vehicle emission factors for traditional pollutants under | Regional air pollution emission models (w) Need for higher resolution emission models to capture the | Better information about emissions of non-methane organic gases (NMOG) that lead to secondary organic aerosol formation (M-H) |



| | standard conditions (h) | effects of lane layouts, vehicle interactions and driving behaviour (u) | |
|---|---|---|---|
| | Total shipping emissions of SOx (h), $CO_2$ (h) and NOx (m) | VOC emissions including speciation (u), in particular for low sulphur fossil fuels (v) | Emission factors for upcoming synthetic fuels incl. ammonia (H) |
| | Source apportionment receptor Models (h) Source-oriented models (m) Inverse models for improvement of emission inventories (l) | Receptor models approaches generally (w) but for higher temporal resolution source apportionment for all regions (u) Emission composition profiles for air pollution sources (v) | Development of emission composition profiles for new technologies and fuels (H). Source apportionment of PM including the secondary PM (H) |
| | Characterisation of indoor air pollutants (m), Quantification of the indoor air pollution source fluxes (m) Chemical and physical processes affecting indoor air pollutants (l) | Development of indoor air quality models with accurate description of the key chemical and physical processes (v). | Systematic quantification of different indoor air pollution sources (H). Role of indoor sources in total exposure to air pollutants (H). Interactions between outdoor and indoor air pollution (H) |
| Urban air quality | Air quality at street level to devise mitigation options and define contribution from local and regional sources (l) Interactions between urban climate and air quality in cities (l) | Developing ratified emissions, air quality and meteorological datasets for cities in developing regions (v) Modification of urban heat island and heat stress due to changing climate (v) | Air quality assessment tools for developing countries to devise local and regional air quality management plans and test policy options to reduce health impacts (H) Quantification of how levels of $PM_{2.5}$, $O_3$, $NO_2$ will change in cities in the future to develop long term air quality and health policies (H) |
| Measurement of air quality | Personal air pollution exposure (h) Data assimilation of three-dimensional observations and modelling (h) Measurements by citizen science (m) | Three-dimensional high-resolution observation and modelling (v) Data assimilation of comprehensive data sources (v) Improve data quality from low-cost sensor networks (u) | Acceptance of data from crowdsourcing, big data analysis and data assimilation (H) Availability of air pollution data for personal health protection (H) Develop guidance for the deployment of low-cost sensors for air quality management purposes (M) |



| | Measurement of particulate matter speciation (l) | Extend measurements of particulate matter speciation in and around cities (u) | Examine control of particulate matter in relation to its species and local to regional scale contributions (H) |
|---|---|---|---|
| Air quality modelling | Spatial & temporal variability of air quality at the street-scale (m for developed regions, l for developing regions)<br><br>Representation of the reactivity of pollutants (aerosols mainly) at the fine spatial scales (l-m)<br><br>Understanding of episodic conditions driven my meteorology and chemistry (m)<br><br>Regional scale air quality modelling (m)<br>Obstacle resolving urban modelling (m) | Earth Systems Modelling for higher resolutions with atmospheric composition capability (v)<br><br>Developments in subgrid & multiscale modelling: emission counting, species transfer, model coupling, use of reliable fine-scale input data (u)<br><br>Methodology for the assessment of fine-scale model performances (v)<br><br>Fine-scale chemistry with aerosols notably in local models and links with larger scale chemistry (v)<br><br>Quantifying changes in the prevalence of episodic conditions in the future e.g., anticyclonic conditions, dry and wet periods (v)<br><br>Model scales matching and integration (u)<br>High time/space resolution atmospheric chemistry and meteorology coupling (u) | Integrated scenarios (activity / micro-scale traffic / urban management / exposure) for policy support (M-H)<br><br>Merging of model & observation (e.g., portable sensor information) for improved mapping and management of individual air quality (H)<br><br>Improved future climate-meteorological-composition interactions to develop more robust future are quality policies (H)<br><br>Integrated multiscale air quality impact assessment resolving urban scale and hotspots (H) |
| Air quality and health and exposure | Health effects (short-term and long-term) related to advanced PM properties [1] (l-m) | Quantification of health impacts related to particulate matter properties (v) | Health impact assessments including advanced PM properties [1] to support refined emission and air quality regulations (M-H) |
| | Concentration-response functions, including their potential thresholds (l-m) | More accurate determination of concentration-response functions representing the full concentration range (v) | Improved health impact assessments representing both higher and lower concentration regions (M) |
| | Combined effects of air pollution, extreme temperatures (heat waves and cold | Evaluation of the combined and synergetic effects of air pollution, extreme temperatures, allergenic pollen and viruses (u) | Impact assessments of the combined effects to support, e.g., mitigation strategies (H) |





| | | | |
|---|---|---|---|
| | spells), allergenic pollen and viruses (l) | | |
| | Dynamic exposure assessment [2] including indoor air quality (l-m) | Modelling and evaluation of dynamic exposure [2] including indoor air quality (u) | Improved policies based on more realistic information on exposures, and on both outdoor and indoor air quality (M-H) |
| | Use of spatially and temporally high-resolution multi-decadal concentration and meteorological datasets for extensive regions (l-m) | Global and regional assessments of air quality and health based on population exposure taking into account of confounding factors (u especially for developing regions) | Improved assessments of air quality and health to support the analysis of environmental inequalities (H) |
| Interactions of air quality, meteorology and climate | Regional scale BVOC emissions and dry deposition modelling (m) | Vegetation species dependent BVOC emissions and dry deposition of gas and aerosol in urban environment (v). | Improvement of air quality impact assessment of nature-based solutions (H) |
| Air quality management and policy | Health benefits of reducing emissions for $PM_{2.5}$ (m) and for $NO_2$ (l); reduction of biodiversity losses caused by reducing emissions (l); monetary valuation of damage endpoints (m) | Advanced multiscale and process based multiscale modelling approaches to support air quality management (u) | Estimation of personal exposure (H); improved exposure-response relationships (H); emission rates for indoor sources (H) and diffuse emissions, e.g from abrasion processes (H); integrating air quality and climate responses (H), estimation of marginal avoidance costs per t of CO2,eq (H) |


(1) PM properties refer here to, e.g., particulate matter size distributions, particle number and chemical composition.
(2) Dynamic exposure assessment refers to exposure studies, which treat the pollutant concentrations in different micro-environments as well as the infiltration of outdoor air to indoors. In dynamic exposure assessment, one can treat also pollution sources and sinks in indoor air.


**10. Conclusions and future direction**

This review has mainly examined research developments that have emerged over the last decade. As part of the review we have provided a short historical survey, before assessing the current status of the research field and then highlighting emerging challenges. We have had to be selective in the key areas of air quality research that have been examined. While the concept of
this review emerged from the 12th International Conference on Air Quality (held virtually during 18-26 May 2020), each of the sections not only provided an air quality research community perspective, but also included a wider literature examination of the areas.





**10.1 Emissions of air pollution**

The emphasis has been on air pollution emissions of major concern for health effects, namely exhaust and non-exhaust emissions from road traffic and shipping, and other anthropogenic emissions, e.g., those from agriculture and wood burning. Developments are continuing to improve global and regional emission inventories and integrating local emissions data into the larger scale inventories. With increasing demand for cleaner vehicles, there is still the need to assess if electric and hybrid vehicles actually reduce total $PM_{2.5}$ and $PM_{10}$ emissions, as emissions from non-exhaust PM from tyre, brake, and road wear

are still present. Developments in onboard monitoring to help improve estimation of real-world emission estimation is another growing area. Understanding the effects of non-exhaust emission will be important to design robust air quality management strategies in the context of other emissions, including wind-blown dust.

Uncertainties still exist in estimating emissions from diffuse processes, such as, abrasion processes in industry, households, agriculture, and traffic, where large variabilities are still present. Other sources, which are not well characterised, include

residential wood combustion as well as the spatial representation of these emissions across regions. While progress in source apportionment models has continued, inverse modelling used for improvement of emission inventories has the potential to reduce their uncertainties.

In terms of chemical speciation, while some improvements have taken place in estimating temporal profiles of agricultural emissions, the amount of $NH_3$ and PM emissions originating from agriculture are still uncertain for many regions. The impact

of new fuels on the chemical composition of NMVOC emissions from combustion processes remains highly uncertain (e.g., low sulphur residual fuels in shipping and new exhaust gas cleaning technologies).

Bringing together air pollution emission inventories with those of greenhouse gases will facilitate integrated assessment measures and policies benefitting from co-benefits. On the urban and street scales, emission models need to be able to simulate the spatial and temporal variations of emissions on a higher resolution from road traffic, taking account of traffic and driving

conditions.

The importance of shipping emissions is growing, as there is a shift to carbon-neutral or zero-carbon fuels. Emission factors for VOC from shipping are generally less certain, and hence little is known about their contribution to particle and ozone formation. To estimate the total environmental impact of shipping, integrated approaches are needed that bring together (i)

impacts from atmospheric emissions on air quality and health, deposition of pollutants to the sea, and (ii) impacts of discharges to the sea on the marine environments and biota, and (iii) climatic forcing.

The greater emphasis on reducing exposure to air pollution requires consideration of both emissions from outdoor and indoor sources, as well as their exchange between indoor and outdoor environments. Emissions of VOC for example, from transportation and the use of volatile chemical products, such as, e.g., pesticides, coatings, inks, personal care products and

cleaning agents are becoming more important, as are combustion gas appliances such as stoves and boilers, smoking, heating and cooking, which are important sources of $PM_{2.5}$, NO, $NO_2$ and PAHs. The complexity of integrated exposure models is





expected to increase, as they have to include both indoor and outdoor emissions of air pollution, accurate description of the key chemical and physical processes as well as treatment of dispersion of air pollution inside, outside and exchange between buildings and the ambient environment (Liu et al, 2013; Bartzis et al, 2015).


## 10.2 Observations to support air quality research

Regarding observation of air quality, this review has focussed on low-cost sensor (LCS) networks, crowdsourcing and citizen science and on the development of modern satellite and remote sensing technics. Connecting observational data with small-scale air quality model simulations to provide personal air pollution exposure has also been discussed.

Remote sensing measurements including satellite observations have a significant role in air quality management because of their spatial coverage, improving spatial resolution and their use in combination with modelling tasks (Hirtl et al, 2020), even for urban areas (Letheren, 2016). Machine learning algorithms are increasingly being used with remote sensing applications (e.g., Foken, 2021) and recent advances have highlighted the potential of statistical analysis tools (e.g., neural learning algorithms) for predicting air quality at the city scale based on data generated by stationary and mobile sensors (Mihaita et al.,

2019). Geostatistical data fusion is allowing fine spatial mapping by combining sensor data with modelled spatial distribution of air pollutant concentrations (Johansson et al., 2015, Ahangar et al., 2019, Schneider et al., 2017).

Applications of LCS as well as networks based on such sensors have increased over the past decade (e.g., Thompson 2016, Karagulian et al., 2019, Barmpas et al., 2020, Schäfer et al., 2021). These applications have also highlighted the need for proper evaluation, quality control and calibration of these sensors. The analysis of LCS data should take account of cross-sensitivities with other air pollutants, effects of aging, and the dependence of the sensor responses on temperatures and humidity in ambient

air (e.g., Brattich et al., 2020).

## 10.3 Air quality modelling

Air quality research, including approaches to manage air pollution, has relied heavily on the continuing developments,

applications and evaluation of air quality models. Air quality models span a wide range of modelling approaches including, e.g., CFD and RANS models used for very high-resolution dispersion applications (e.g. Nuterman et al., 2012; Andronopoulos et al., 2019), and Lagrangian plume models to Eulerian grid CTMs used for urban to regional scales. An interesting development is that of the implementation of multiply nested LES and coupling of urban-scale deterministic models with local probabilistic models (see e.g., Hellsten et al., 2021), although complexities arise because of the different parameterisations and

the treatment of boundary conditions. A limitation that needs addressing with CFD, including LES models, is that they are currently suited mainly for dispersion of tracer contaminants or where only simple tropospheric chemistry is relevant. Lack of more sophisticated or realistic description of, e.g., NOx/VOC chemistry can cause significant bias in the concentration gradients at very fine scales.

Over the last decade new developments have focussed in improving scale interactions and model resolution to resolve the

spatial variability and heterogeneity of air pollution (e.g. Jensen et al., 2017, Singh et al., 2014, 2020a) at street scales in a city





area. New approaches of artificial neural network models and machine learning have shown a more detailed representation of air quality in complex built-up areas (e.g., Wang et al., 2015b, Zhan et al., 2017, Just et al., 2020, Alimissis et al., 2018). CTMs have also been developed to improve spatial resolution, for example, through downscaling approaches for predicting air quality in urban areas, forecasting air quality and simulation of exposure at the street scale (Berrocal et al., 2020, Elessa Etuman et al.,

2020, Jensen et al., 2017). Ensemble simulations have proven to be successful to provide more reliable air quality prediction and forecasting (e.g., Galmarini et al., 2012, Hu et al., 2017) and complementary hybrid approaches have been explored for multi-scale applications (Galmarini et al., 2018).

The strong interaction between local and regional contributions, especially to secondary air pollutants ($PM_{2.5}$ and $O_3$) has motivated the coupling of urban and regional scale models (e.g., Singh et al., 2014; Kukkonen et al., 2018). With the

importance of exposure assessment increasing, the incorporation of finer spatial scales within a larger spatial domain is required, which introduces the challenging issue of representing multiscale dynamical and chemical processes, while maintaining realistic computational constraints (e.g. Tsegas et al., 2015). Similarly, machine learning approaches offer possibilities to use observational data to improve fine scale air quality and personal exposure predictions (Shaddick et al., 2021).


### 10.4 Interactions between air quality, meteorology and climate

Our review has highlighted the need to integrate predictions of weather, air quality and climate where Earth System Modelling (ESM) approaches is playing an increasing role (WWRP, 2015; WMO, 2016). There are also continued improvements from higher spatial resolution modelling and interconnected multiscale processes, while maintaining realistic computational times.

Many advances have taken place in the development and use of coupled regional scale meteorology-chemistry models for air quality prediction and forecasting applications (e.g., Kong et al 2017; Baklanov et al., 2014, 2018a). These advances contribute to assess complex interactions between meteorology, emission and chemistry, for example, relating to dust intrusion and wild-land fires (e.g. Kong et al., 2015). Data assimilation of chemical species data into CTM systems is still an evolving field of research; it has the potential to better constrain emissions in forecast applications. An example would be data assimilation of

urban observations (including meteorological, chemical and aerosol species) to investigate multiscale effects of the impacts of aerosols on weather and climate (Nguyen and Soulhac, 2021).

Urban and finer scale (e.g., built environment) studies are showing that improvements in the treatment of albedo, the anthropogenic heat flux, and the feedbacks between urban pollutants and radiation, can influence urban air quality significantly (e.g. González-Aparicio et al., 2014; Fallmann et al., 2016; Molina, 2021). These considerations can be very important for

urban air quality forecasting, as temporal variations in air pollutant concentrations on a short term are largely due to variabilities in meteorology. Understanding and parameterising multiscale and non-linear interactions, for example, evolution and dynamics of urban heat island circulation, and aerosol forcing and urban aerosol interactions with clouds and radiation remains an ongoing atmospheric science challenge. Another remaining research challenge that involves multiscale interactions includes



the formation of secondary air pollutants (e.g. ozone, and secondary organic and inorganic aerosols), especially to describe air
quality over urban, sub-urban and rural environments.

Development and evaluation of nature-based solutions to improve air quality demands an improved understanding of the role
of biogenic emissions (Cremona et al., 2020) as a function of vegetation species and characteristics. Interactions are influenced
by several factors, such as vegetation drag, pollutant absorption as well as biogenic emissions. These factors will determine
the impact on air quality, be it positive or negative (Karttunen et al., 2020; San Josè et al., 2020; Santiago et al., 2019).
Advanced approaches are needed to describe biogenic emissions together with gas and particle deposition over vegetation
surfaces to further assess the effectiveness of nature-based solutions to improve air quality in cities.

**10.5 Air quality exposure and health**

Air quality related observations to support air quality health impact studies are heterogeneous; for many developing regions,
such as Africa, ground-based monitoring is sparse or non-existent (Rees at al., 2019). The motivation is growing for an inter-
disciplinary approach to assess exposure and the burden of disease from air pollution (Shaddick et al., 2021); this could benefit
from the combined use of ground and remote sensing measurements, including satellite data, with atmospheric chemical
transport and urban scale dispersion modelling.

Air quality impact on health can occur on short and long timescales. PM, which is one of the most health relevant air pollutants,
is associated with many health effects, such as, all cause- cardiovascular, respiratory mortality and childhood asthma (e.g., Dai
et al., 2014; Samoli et al., 2013; Stafoggia et al., 2013 and Weinmayr et al., 2010). There have been significant advances that
reveal new evidence of the health impact of PM components, such as $SO_4$, EC, OC and metals (Wang et al., 2014; Adams et
al., 2015; Hampel et al., 2015 and Hime et al., 2018). Challenges remain to elucidate the relative role of PM components and
measures in determining the total health impact. These include particle number concentrations (PNC), secondary organic PM,
primary PM, various chemical components, suspended dust, the content of metals, and toxic or hazardous pollutants.

Improved knowledge on the health impacts of PM components have also stimulated further debate on the optimal
concentration-response functions and on the necessity of threshold or lower limit values, below which health impacts might
not manifest (Burnett et al. 2018). These challenges will feed into health impact studies, such as EEA (2019) that estimated
that more than 340.000 premature deaths per year in Europe were linked to exposure to $PM_{2.5}$. However, another study by
Lelieveld et al. (2019) indicated that health impacts from $PM_{2.5}$ exposure may have been considerably underestimated.

The worldwide impact from the COVID-19 pandemic caused by the SARS-CoV-2 virus has raised global interest in the links
between air quality and the spread of viruses (van Doremalen et al., 2020). However, the exact role and mechanisms are not
yet clear and require concerted effort (e.g., Pisoni and Van Dingenen 2020). There is also evidence that poor air quality can
exacerbate health effects from other environmental stressors, including heat waves, cold spells, and allergenic pollen (e.g.,
Klein et al., 2012, Horne et al., 2018; Xie et al. 2019; Phosri et al. 2019).

The link between population activity and actual exposure is also becoming clearer, where dynamic diurnal activity patterns
provide more accurate representation of exposures to air pollution (Soares et al., 2014, Kukkonen et al., 2016b, Smith et al.,



2016; Singh et al., 2020a). Recent work by Picornell et al. (2019) and Ramacher et al. (2019) has also demonstrated the importance of the movements of people to assess exposure.

For evaluating the relative significance of various PM properties and measures on human health, a denser measurement network on advanced PM properties would be needed, both on regional and urban scales. The required PM properties would include, in particular, size distributions and chemical composition. Clearly, such a network would be especially valuable in cities and regions, which include high-quality population cohorts. The most important requirement in terms of PM modelling would be improved emission inventories, which would also include sufficiently accurate information on particle size

distributions and chemical composition.

### 10.6 Air quality management and policy

Integrated assessment of air pollution control policies has progressively developed over the last two decades and has been widely used as a tool for air quality management (e.g., EC, 2021). Relatively recently, integrated assessment for air pollution

control in research projects has started to take account of climate change. Correspondingly, integrated assessment activities for climate protection have started to include impacts of air pollution into the assessment (Friedrich et al., 2020). Some national authorities, such as, e.g., the German Federal Environmental Agency or the UK Department for Business, Energy and Industrial Strategy have also recommended an integrated assessment, combining the assessment of climate and air pollution impacts (Matthey and Bünger 2019, DBEIS 2019). Impact pathway approaches are also currently increasingly incorporating exposure

to air pollutants as an indicator of health impacts, instead of the previously applied concentration of air pollutants at fixed outdoor locations (Li and Friedrich, 2019). This has an implication for epidemiological studies, which usually are based on correlation between modelled or measured concentrations at outdoor locations and health risks (e.g., Singh et al., 2020a). Interdependence of air pollution and climatically active species allows co-benefits to be optimised. This approach also shows that costs of meeting policy obligations for climate protection (e.g., for the Paris Agreement) can be reduced or offset by the

benefits of reduced health impacts from improved air quality (Markandya et al., 2020). On the other hand, for some climate protection measures, the benefits of the reduced climate change are much smaller than the impacts caused by the increased air pollution. This has been demonstrated for wood combustion, which while being more climate friendly than fossil fuels, will give rise to $PM_{2.5}$ and NOx emissions (Huang et al., 2016; Friedrich and Vogt 2020, Kukkonen et al., 2020b). Some recent studies (e.g., Schmid, 2019) have provided evidence on the advantages of using costs and benefits for both climate and air

pollution abatement measures into integrated assessments.

Air quality management must adapt to the tightening of policy driven regulations. Recently, the sulphur content of the fuel for ships has been reduced to 0.5% worldwide (IMO, 2019). The EURO 6d norm has led to a significant reduction of NOx in the exhaust gas of diesel cars, whereas the EURO 7 norm planned to be implemented 2025 will further reduce PM and NOx emissions from vehicle engines. The European Council has recently (in September 2020) agreed to reduce EU's greenhouse

gas emissions in 2030 by at least 55% compared to the corresponding emissions in 1990. Together with national reduction plans for GHG this will significantly reduce emissions of air pollutants from the combustion of fossil fuels. However, there is



one exception: small wood and pellet firings (< 500 kW)i, where still further measures should be developed for reducing the PM and NOx emissions (e.g., Kukkonen, 2020b).

While direct combustion emissions are expected to decrease, a particular challenge will be to control diffuse emissions, e.g., from abrasion processes, bulk handling, demolition of buildings and use of paints and cleaning agents. Despite cleaner vehicles, emissions from tyre and brake wear and road abrasion remain an important challenge. Other areas that pose challenges for air quality management are the need to reduce agricultural emissions, especially of ammonia, which can lead to the production of secondary aerosols (especially ammonium nitrates).

Using personal exposure instead of outdoor concentration as an indicator in health impact assessments offers the opportunity to assess the impacts of indoor air pollution control. Possibilities to reduce emissions from indoor sources should be assessed, such as smoking, frying and cooking, candles and incense sticks, open chimneys and wood stoves, and cleaning agents. Furthermore, using HEPA filters in vacuum cleaners, air filters and cooker bonnets and using mechanical ventilation with heat recovery should be analysed. In addition, possibilities for reducing PM concentrations in underground rail stations should be

explored.

Finally, we consider cross cutting needs as a synthesis of our finding and suggest recommendations for further research. Specifically, we indicate the confidence in the scientific knowledge, the urgency to complete the science gaps and the importance of each area for supporting policy.

**Acknowledgments**

The support of the following institutions and enterprises is gratefully acknowledged:

University of Hertfordshire, Aristotle University Thessaloniki, TITAN Cement S.A., TSI GmbH

APHH UK-India Programme on Air Pollution and Human Health (funded by NERC, MOES, DBT, MRC, Newton Fund), American Meteorological Society (AMS) Air & Waste Management Association (A&WMA)

We especially acknowledge the tireless effort of Ioannis Pipilis, Afedo Koukounaris, Eva Angelidou.

World Meteorological Organisation (WMO) GAW Urban Research Meteorology and Environment (GURME) Programme for supporting and contributing to this review.

KS is grateful for funding within the frame of the project Smart Air Quality Network by the German Federal Ministry of Transport and Digital Infrastructure - Bundesministerium für Verkehr und digitale Infrastruktur (BMVI) under grant no.

19F2003A-F.

TH is grateful for funding within the activity PROGRES Q16 by the Charles University, Prague.

Vikas Singh is thanked for providing Figure 10.

This activity has received funding from the European Union's Horizon 2020 research and innovation program under grant agreements No 603946 (HEALS), No 690105 (ICARUS), No 814893 (SCIPPER), 820655 (EXHAUSTION), No 874990

(EMERGE).



The activity has been supported by the EU LIFE financial program through the project VEG-GAP "Vegetation for Urban Green Air Quality Plans" (LIFE18 PRE IT003).

This work reflects only the authors' view and the Innovation and Networks Executive Agency is not responsible for any use that may be made of the information it contains.

We would also like to thank for the funding of Nordforsk under the Nordic Programme on Health and Welfare (Project #75,007: NordicWelfAir - Understanding the link between Air pollution and Distribution of related Health Impacts and Welfare in the Nordic countries).

We wish to thank Dr. Antti Hellsten (FMI) for his useful comments on CFD modelling.

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
