# Peer review of "Advances in Air Quality Research – Current and Emerging Challenges"

_Atmospheric Chemistry and Physics, 2021_

## Author Response (AR1)

**Authors Responses to RC1 Comments**

This manuscript presents a deep and extensive review of the research and developments that have emerged during approximately the last decade with respect to 'Advances in Air Quality Research'.

All topics considered are appropriate, timely and well targeted.

The document addresses relevant scientific questions, its content represents a successful review of the key elements of the subject under consideration.

The manuscript correctly presents the review of ideas, tools and data.

The conclusions are timely and substantial.

The analysis carried out clearly supports supporting the interpretations and conclusions.

The authors give appropriate credit to the related work.

The title clearly reflects the content of the article.

The abstract presented is a concise and complete summary.

The language is precise and correct, although there are sections that are difficult to read.

The number and quality of the references, in general, are adequate and reflect well the review carried out.

AC - We are grateful to the Reviewer for his/her kind words and appreciation of quality of the review manuscript.

The document really represents an in-depth review of the state of the art, especially those developed in Europe. But it is too long, there are certain topics that are treated, and this is indicated in the text, several times. It is considered that a revision of the text would be necessary to make it more compact. Despite its interest and the good structure of the different sections, its length makes its reading really hard and heavy, resulting in its loss of interest.

AC – We thank the Reviewer for his/her reflections on the length of the manuscript. Firstly, given the depth and breadth of the review we are of the view that the length is appropriate and it will be difficult to reduce the length of the paper without curtailing the depth of the review. Secondly, we have very carefully selected only the most relevant and upcoming subtopics of air quality research and omitting any one of these or further reducing the critical analysis, will inevitably lead to diluted conclusions or will weaken the evidence that support the final recommendations. Thirdly, the review has been structured so that each section is self-contained. This has the advantage that readers have the option to select sections that are most relevant to their interests. Reviewer 2 also has commented that there not 'many repetitions that can be reduced to make the manuscript shorter'. In conclusion, we ask the editors to keep the length of the manuscript as it is for the reasons given above. If, however, the editors wish us to reduce the length we will abide by their decision with the proviso listed above.

line 80. The manuscript should be updated in reference to the new WHO '2021 guideline values.

AC – We have made reference to the updated WHO 2021 air quality guidelines in lines 88-90.

line 225, what is indicated is totally opportune, it could be complemented by the following references.

- WHO, 2006. Health risks of particulate matter from long-range transboundary air pollution. 113 pp. World Health Organization http://www.euro.who.int/__data/assets/pdf_file / 0006/78657 / E88189.pdf.
- Lenschow, P., Abraham, H.-J., Kutzner, K., Lutz, M., PreuB, J.-D., Reichenbfficher, W., 2001. Some ideas about the sources of PM10. Atmos. Environ. 35 (1), S23-S33. doi.org/10.1016/S1352-2310(01)00122-4 (2001).
- Baldasano J.M. (2020) COVID-19 lockdown effects on air quality by NO2 in the cities of Barcelona and Madrid (Spain). Science of the Total Environment, STOTEN-140353 doi.org/10.1016/j.scitotenv.2020.140353

AC – We welcome the suggestions from the reviewer however, we feel that the first two are very old and not representative of the current state of the art. We have added the reference of EEA 2020c which describes the long-range air pollution contributions and is up to date (see line 225 and Reference list). We have also included the third reference in line 2687 which is more relevant to support the discussion on air quality and covid in Section 9.5) on challenges related to pandemics.

Line 595. The reference on HERMES should be complemented with:

- Baldasano JM, LP Güereca, E. López, S. Gassó, P. Jiménez-Guerrero (2008) Development of a high resolution (1 km x 1 km, 1 h) emission model for Spain: the High-Elective Resolution Modeling Emission System (HERMES). Atmospheric Environment, 42: 7215-7233 doi: 10.1016 / j.atmosenv.2008.07.026
- Guevara M., F. Martínez, G. Arévalo, S. Gassó, J.M. Baldasano (2013) Improved system for modeling Spanish emissions: HERMESv2.0. Atmospheric Environment 81: 209-221 doi: 10.1016 / j.atmosenv.2013.08.053

AC – We are grateful to the reviewer for alerting us to these publications. These references have been included in line 584 along with more up to date work.

Subfigures a) and b) of figure 1 should be in the absolute values of the pollutants emitted, not in relative%, since this would more clearly show the reduction effort made. Subfigure c) already relativizes the contribution of the different emission sources.

AC – As suggested by the Reviewer, we have made the changes to Figure 2 (a, b and c) as described below.

Figure 2. EU-28 emission trends in absolute and relative numbers for a) the main gaseous air pollutants, and b) particulate matter. Panel c) shows the share of EU emissions of the main pollutants by sector in 2018 (EEA, 2020c)

The entire References section and the corresponding citations in the text should be reviewed in detail. There are citations that are not included (eg Niemeyer 1960), others are not ordered correctly.

AC – We thank the Reviewer for alerting us to the issues on References. We have carefully examined all references and checked that they are cited and referenced properly.

**Authors Responses to RC2 Comments**

This manuscript presents an extensive and detailed review of the air pollution research and the advances in the last decades. It highlights and focuses on particular challenges and improvements in our understanding and implementation of air pollution science and provides recommendations for future research.

AC - We are grateful to the Reviewer for his/her kind words and appreciation of quality of the review manuscript.

The structure of the manuscript is very well-designed, and the flow is very smooth. One concern is the length of the paper, although I do not see many repetitions that can be reduced to make the manuscript shorter.

AC – We thank the Reviewer for his/her reflections on the length of the manuscript. In our reply to the comments from Referee 1, we feel that given the depth and breadth of the review the length of the manuscript is appropriate. As the Reviewer agrees there are very few areas where we can reduce the length of the article without sacrificing its quality and possibly diluting conclusions or weakening the supporting evidence for the final recommendations. In conclusion, we ask the editors to keep the length of the manuscript as it is for the reasons given above. If, however, the editors wish us to reduce the length we will abide by their decision with the proviso listed above.

I find the manuscript suitable for publishing in ACP, however with some minor comments and additions that I have listed below.

Line 53: Is it not "low cost sensors"? is there a difference between "low" and "lower"? - DONE

AC – The usual and well-established term is "low-cost sensor". However, terms like "very low cost sensor", "ultra low cost sensor" and "lower cost sensor" are used to show that differences in the complexity and quality of technology exists and these are often expressed in costs of the units. Here, the term "lower cost sensor" was used to cover low-cost sensors (LCS) and medium-cost sensors (MCS). In response to the reviewers' comment, however, we have changed all terms "lower cost sensor" into "low-cost sensor".

In Figure 6 caption, we have also changed the term "Ultra-Low-Cost Measurements" into "Low-Cost Sensor Measurements".

Line 86: Please also refer to more recent studies such as Lelieveld et al. (2019) estimating more than 8 million premature deaths globally.

AC – We are grateful to the Reviewer for alerting is to the article. We have added and discussed the reference in line 1919 and 2871 and the references has also been included in the Reference Section.

Line 175: Please provide a time period for this decrease.

AC – As requested by the Reviewer, this now indicated in line 177.

Line 299: Remove "now" or "nowadays" to avoid replication in meaning.

AC – As requested by the Reviewer, the word 'now' has been deleted (see line 301).

Section 1.7: Here, other emerging approaches than the mass-based PM2.5 can be briefly introduced, such as the oxidative potential. In addition, it is important to refer also to the global burden of disease (GDB) studies, as well as linear vs non-linear risk functions, fx Lehtomaki et al. (2020).

Lehtomäki, H., Geels, C., Brandt, J., Rao, S., Yaramenka, K., Åström, S., Andersen, M.S., Frohn, L.M., Im, U., & Hänninen, O. (2020). Deaths attributable to air pollution in Nordic countries: disparities in the estimates. Atmosphere, 11(5), 467. doi:10.3390/atmos11050467.

AC – We are grateful to the Reviewer for his/her suggestions which have all be incorporated in the manuscript.

Oxidative potential of PM has been mentioned in line 371-372 and further elaborated in lines 2150 – 2154.
The following references have also been added.
Gao, D., Godri Pollitt, K. J., Mulholland, J. A., Russell, A. G., and Weber, R. J.: Characterization and comparison of $PM_{2.5}$ oxidative potential assessed by two acellular assays, Atmos. Chem. Phys., 20, 5197–5210, https://doi.org/10.5194/acp-20-5197-2020, 2020.

He L., Norris C., Cui X., Li Z., Barkjohn K. K., Brehmer C., Teng Y., Fang L., Lin L., Wang Q., Zhou X., Hong J., Li F., Zhang Y., Schauer J. J., Black M., Bergin M. H., Zhang J. J.: Personal Exposure to $PM_{2.5}$ Oxidative Potential in Association with Pulmonary Pathophysiologic Outcomes in Children with Asthma. Environ Sci Technol. 2021 Mar 2;55(5):3101-3111. doi: 10.1021/acs.est.0c06114, 2021.

The GDB has been mentioned in lines 343-345.

Linear vs non-linear is discussed in lines 2120 - 2124

The reference of Lehtomäki et al., (2020) has been included in lines 1939, 2122 and 2375 and has been added to the list of references.

Line 466: Please explain why BC decreased more than other PM components.

AC – An explanation is given in lines 470-471.

Section 3.2.4: Opening up new routes in the Arctic ocean due to loss of sea-ice can also be mentioned here, i.e. Geels et al., 2021.

Geels, C., Winther, M., Andersson, C., Jalkanen, J.-P., Brandt, J., Frohn, L. M., Im, U., Leung, W., and Christensen, J. H.: Projections of shipping emissions and the related impact on air pollution and human health in the Nordic region, Atmos. Chem. Phys., 21, 12495–12519, https://doi.org/10.5194/acp-21-12495-2021, 2021.

AC – As suggested by the Reviewer, this reference has been included in Section 3.2.4, lines 682. We added a sentence on the construction of future scenarios with ship emission model systems and also added the references Matthias et al., 2016 , Karl et al., 2019, and Geels et al., 2021 at this place.

Line 790: Consider adding Im et al., 2019 as a recent example:

Im, U., Christensen, J. H., Nielsen, O.-K., Sand, M., Makkonen, R., Geels, C., Anderson, C., Kukkonen, J., Lopez-Aparicio, S., and Brandt, J.: Contributions of Nordic anthropogenic emissions on air pollution and premature mortality over the Nordic region and the Arctic, Atmos. Chem. Phys., 19, 12975–12992, https://doi.org/10.5194/acp-19-12975-2019, 2019.

AC – This reference has been included in Section 3.2.6, lines 781.

Section 3.3: Although explicitly stated which sectors will be focused and why, I think it is also important to mention changes in natural emissions, especially their link to climate change. Some emerging ones include high latitude dust, marine organic aerosols and precursor gases such as marine VOCs, etc.

AC – In response to the Reviewers' suggestion, we have made the changes as suggested in lines 844-846.

Section 5.2.6. I recommend to also mention the later phases of AQMEII, ie AQMEII2 (online-coupled models only: Im et al., 2015a,b), and AQMEII3 (Galmarini et al., 2018, Im et al., 2018a,b).

Im, U., Bianconi, R., Solazzo, E., Kioutsioukis, I., Badia, A.,Balzarini, A., Baro, R., Bellasio, R., Brunner, D., Chemel, C.,Curci, G., Denier van der Gon, H., Flemming, J., Forkel, R.,Giordano, L., Jimenez-Guerrero, P., Hirtl, M., Hodzic, A.,Honzak,L., Jorba, O., Knote, C., Makar, P. A., Manders-Groot, A.,Neal, L., PeÌ• rez, J. L., Pirovano, G., Pouliot, G., San Jose, R., Savage,N., Schroder,W., Sokhi, R. S., Syrakov, D., Torian, A., Tuccella,P., Wang, K., Werhahn, J., Wolke, R., Zabkar, R., Zhang,Y., Zhang, J., Hogrefe, C., and Galmarini, S.: Evaluation of operational online coupled regional air quality models over Europe and North America in the context of AQMEII phase 2, Part II: particulate matter, Atmos. Environ., 115, 421–441, 2015a.

Im, U., Bianconi, R., Solazzo, E., Kioutsioukis, I., Badia, A.,Balzarini, A., Baro, R., Bellasio, R., Brunner, D., Chemel, C., Curci, G., Flemming, J., Forkel, R., Giordano, L., Jimenez- Guerrero, P., Hirtl, M., Hodzic, A., Honzak, L., Jorba, O., Knote,C., Kuenen, J. J. P., Makar, P. A., Manders-Groot, A., Neal, L., PeÌ• rez, J. L., Pirovano, G., Pouliot, G., San Jose, R., Sav- age, N.,Schroder, W., Sokhi, R. S., Syrakov, D., Torian, A., Tuc- cella, P.,Werhahn, J.,Wolke, R., Yahya, K., Zabkar, R., Zhang, Y., Zhang,J., Hogrefe, C., and Galmarini, S.: Evaluation of op- erational online-coupled regional air quality models over Europe and NorthAmerica in the context of AQMEII phase 2, Part I: ozone, Atmos. Environ., 115, 404–420, 2015b.

Galmarini, S., Kioutsioukis, I., Solazzo, E., Alyuz, U., Balzarini, A., Bellasio, R., Benedictow, A. M. K., Bianconi, R., Bieser, J., Brandt, J., Christensen, J. H., Colette, A., Curci, G., Davila, Y., Dong, X., Flemming, J., Francis, X., Fraser, A., Fu, J., Henze, D. K., Hogrefe, C., Im, U., Garcia Vivanco, M., Jiménez-Guerrero, P., Jonson, J. E., Kitwiroon, N., Manders, A., Mathur, R., Palacios-Peña, L., Pirovano, G., Pozzoli, L., Prank, M., Schultz, M., Sokhi, R. S., Sudo, K., Tuccella, P., Takemura, T., Sekiya, T., and Unal, A.: Two-scale multi-model ensemble: is a hybrid ensemble of opportunity telling us more?, Atmos. Chem. Phys., 18, 8727–8744, https://doi.org/10.5194/acp-18-8727-2018, 2018.

Im, U., Brandt, J., Geels, C., Hansen, K. M., Christensen, J. H., Andersen, M. S., Solazzo, E., Kioutsioukis, I., Alyuz, U., Balzarini, A., Baro, R., Bellasio, R., Bianconi, R., Bieser, J., Colette, A., Curci, G., Farrow, A., Flemming, J., Fraser, A., Jimenez-Guerrero, P., Kitwiroon, N., Liang, C.-K., Nopmongcol, U., Pirovano, G., Pozzoli, L., Prank, M., Rose, R., Sokhi, R., Tuccella, P., Unal, A., Vivanco, M. G., West, J., Yarwood, G., Hogrefe, C., and Galmarini, S.: Assessment and economic valuation of air pollution impacts on human health over Europe and the United States as calculated by a multi-model ensemble in the framework of AQMEII3, Atmos. Chem. Phys., 18, 5967–5989, https://doi.org/10.5194/acp-18-5967-2018, 2018.

Im, U., Christensen, J. H., Geels, C., Hansen, K. M., Brandt, J., Solazzo, E., Alyuz, U., Balzarini, A., Baro, R., Bellasio, R., Bianconi, R., Bieser, J., Colette, A., Curci, G., Farrow, A., Flemming, J., Fraser, A., Jimenez-Guerrero, P., Kitwiroon, N., Liu, P., Nopmongcol, U., Palacios-Peña, L., Pirovano, G., Pozzoli, L., Prank, M., Rose, R., Sokhi, R., Tuccella, P., Unal, A., Vivanco, M. G., Yarwood, G., Hogrefe, C., and Galmarini, S.: Influence of anthropogenic emissions and boundary conditions on multi-model simulations of major air pollutants over Europe and North America in the framework of AQMEII3, Atmos. Chem. Phys., 18, 8929–8952, https://doi.org/10.5194/acp-18-8929-2018, 2018.

AC - We thank the reviewer for these insightful remarks on the references from later AQMEII phases. The Galmarini et al., (2018) reference is already cited in the paragraph (line 1435).

The first two proposed references are more about ensemble evaluation approaches and not ensemble techniques and the section is not dedicated to evaluation. However, we have cited another paper (line 1430) by the authors, Kioutsioukis et al., (2016), which elaborates on the AQMEII approach for ensemble modelling.

The last two references concern the use of an ensemble for economic evaluations and the conduct of sensitivity studies with an ensemble, respectively, and do not fall directly under this section and hence have not been cited here.

Section 6.1: Earth system model (ESM) terminology generally refers to current state-of the-art climate models where the different components of the Earth system are taken into account in a coupled way. However, ESMs refereed in this section focus on online-coupled meteorology and chemistry/transport models. I recommend not using ESM in this section, but something like online-coupled CTMs or something like that. If this terminology will be used, then I recommend to refer also to CMIP experiments that serve as input to the IPCC assessment reports.

AC – We thank the Reviewer for noticing that the distinction between ESM and coupled chemistry-meteorology models which was not clear.

The text of Section 6.1 has been modified (lines 1536-1541) to clarify that the development of EMS (models that take into account the different components of the Earth system in a coupled way) is the future perspective. Atmospheric meteorology-chemistry coupled models on their own (can be an element of ESM) are still taken into account in our review paper dealing with air quality related research.

Line 1569: Consider adding Im et al., 2015a,b

Im, U., Bianconi, R., Solazzo, E., Kioutsioukis, I., Badia, A.,Balzarini, A., Baro, R., Bellasio, R., Brunner, D., Chemel, C.,Curci, G., Denier van der Gon, H., Flemming, J., Forkel, R.,Giordano, L., Jimenez-Guerrero, P., Hirtl, M., Hodzic, A.,Honzak,L., Jorba, O., Knote, C., Makar, P. A., Manders-Groot, A., Neal, L., PeÌ• rez, J. L., Pirovano, G., Pouliot, G., San Jose, R., Savage,N., Schroder,W., Sokhi, R. S., Syrakov, D., Torian, A., Tuccella,P., Wang, K., Werhahn, J., Wolke, R., Zabkar, R., Zhang,Y., Zhang, J., Hogrefe, C., and Galmarini, S.: Evaluation of operational online coupled regional air quality models over Europe and North America in the context of AQMEII phase 2, Part II: particulate matter, Atmos. Environ., 115, 421–441, 2015a.

Im, U., Bianconi, R., Solazzo, E., Kioutsioukis, I., Badia, A.,Balzarini, A., Baro, R., Bellasio, R., Brunner, D., Chemel, C., Curci, G., Flemming, J., Forkel, R., Giordano, L., Jimenez- Guerrero, P., Hirtl, M., Hodzic, A., Honzak, L., Jorba, O., Knote,C., Kuenen, J. J. P., Makar, P. A., Manders-Groot, A., Neal, L., PeÌ€rez, J. L., Pirovano, G., Pouliot, G., San Jose, R., Sav- age, N.,Schroder, W., Sokhi, R. S., Syrakov, D., Torian, A., Tuc- cella, P.,Werhahn, J.,Wolke, R., Yahya, K., Zabkar, R., Zhang, Y., Zhang,J., Hogrefe, C., and Galmarini, S.: Evaluation of operational online-coupled regional air quality models over Europe and North America in the context of AQMEII phase 2, Part I: ozone, Atmos. Environ., 115, 404–420, 2015b.

AC – We are grateful to the Reviewer for bringing these excellent papers to our attention. We have cite the work done inside the AQMEII Phase2 project by adding a citation of the AQMEII Atmospheric Environment special issue (see lines 1570 – 1571 and 1578-1579) including the suggested papers by Im et al. 2015a,b as well as other 25 relevant papers of the special issue. The new references have been added in Section 6.2.1.

Line 1812: add also Bauer et al., 2019.

Bauer, S.E., U. Im, K. Mezuman, and C.Y. Gao, 2019: Desert dust, industrialization and agricultural fires: Health impacts of outdoor air pollution in Africa. J. Geophys. Res. Atmos., 124, no. 7, 4104-4120, doi:10.1029/2018JD029336

AC – As recommended this excellent reference has been added to line 1817.

Line 1822: I recommend referring to two multi-model ensemble based health impact studies in frame of HTAP2 and AQMEII3.

Im, U., Brandt, J., Geels, C., Hansen, K. M., Christensen, J. H., Andersen, M. S., Solazzo, E., Kioutsioukis, I., Alyuz, U., Balzarini, A., Baro, R., Bellasio, R., Bianconi, R., Bieser, J., Colette, A., Curci, G., Farrow, A., Flemming, J., Fraser, A., Jimenez-Guerrero, P., Kitwiroon, N., Liang, C.-K., Nopmongcol, U., Pirovano, G., Pozzoli, L., Prank, M., Rose, R., Sokhi, R., Tuccella, P., Unal, A., Vivanco, M. G., West, J., Yarwood, G., Hogrefe, C., and Galmarini, S.: Assessment and economic valuation of air pollution impacts on human health over Europe and the United States as calculated by a multi-model ensemble in the framework of AQMEII3, Atmos. Chem. Phys., 18, 5967–5989, https://doi.org/10.5194/acp-18-5967-2018, 2018.

Liang, C.-K., West, J. J., Silva, R. A., Bian, H., Chin, M., Davila, Y., Dentener, F. J., Emmons, L., Flemming, J., Folberth, G., Henze, D., Im, U., Jonson, J. E., Keating, T. J., Kucsera, T., Lenzen, A., Lin, M., Lund, M. T., Pan, X., Park, R. J., Pierce, R. B., Sekiya, T., Sudo, K., and Takemura, T.: HTAP2 multi-model estimates of premature human mortality due to intercontinental transport of air pollution and emission sectors, Atmos. Chem. Phys., 18, 10497–10520, https://doi.org/10.5194/acp-18-10497-2018, 2018.

AC – We are grateful to the reviewer for bringing these excellent papers to our attention. These have been included in line 1827.

Section 7.3.1: I recommend addressing the linear and non-linear exposure response functions and threshold values used in the assessment of health impacts due to exposure to PM2.5. The impact of these approaches can be substantial (e.g. Lehtomaki et al., 2020; EEA, 2019).

Air quality in Europe — 2019 report, No 10/2019. ISSN 1977-8449.

AC – We have included the suggested discussion in Section 7.3.1, in lines 2120 – 2124.